# Local field potentials in the parietal reach region reveal mechanisms of bimanual coordination

Eric Mooshagian [1,3 ✉], Charles D. Holmes[2] & Lawrence H. Snyder [1,2 ✉]

Primates use their arms in complex ways that frequently require coordination between the two arms. Yet the planning of bimanual movements has not been well-studied. We recorded spikes and local field potentials (LFP) from the parietal reach region (PRR) in both hemispheres simultaneously while monkeys planned and executed unimanual and bimanual reaches. From analyses of interhemispheric LFP-LFP and spike-LFP coherence, we found that task-specific information is shared across hemispheres in a frequency-specific manner. This shared information could arise from common input or from direct communication. The population average unit activity in PRR, representing PRR output, encodes only planned contralateral arm movements while beta-band LFP power, a putative PRR input, reflects the pattern of planned bimanual movement. A parsimonious interpretation of these data is that PRR integrates information about the movement of the left and right limbs, perhaps in service of bimanual coordination.

[1] Department of Neuroscience, Washington University School of Medicine, St. Louis, MO, USA. [2] Department of Biomedical Engineering, Washington University in St. Louis, St. Louis, MO, USA. [3] Present address: Department of Cognitive Science, University of California San Diego, La Jolla, CA, USA. ✉email: ericm@eye-hand.wustl.edu; larry@eye-hand.wustl.edu

Primates, including humans, are uniquely skilled at using their limbs in complex ways to achieve a variety of goals. Their behavioral repertoire includes manipulating objects, brachiating, defensive, and offensive maneuvers. How the brain coordinates the movements of multiple body parts to produce coherent motor behavior remains a fundamental question in systems neuroscience, applicable across a broad range of species and motor outputs.

Each hemisphere primarily controls the limbs on the opposite side of the body[1,2]. To make a coordinated bimanual reach, some regions must have information about both arms. Bimanual responses have been characterized in cortical motor and premotor areas[3–7]. Suggestive evidence also exists for bimanual representations in the parietal cortex[2,5,6,8], but those studies did not address bilateral contributions to the movement.

The posterior parietal cortex contains sensorimotor signals involved in spatial visuomotor transformations, with different cortical regions specialized for different effector systems[9]. In macaques, reach planning and execution drives systematic patterns of activity in the parietal reach region (PRR)[10–13]. In humans, functional neuroimaging shows reach planning activity in possible homologs of PRR[14–17]. In both species, the primary organization is contralateral. Temporary inactivation of PRR selectively interferes with contralateral arm movements[18], and the population average unit activity codes only contralateral arm movements. Chang et al.[19] showed that some individual cells in PRR reflect plans for ipsilateral limb movements, but Mooshagian et al.[2] showed that this response could more parsimoniously be attributed to the presence of a behaviorally relevant stimulus in the cells' response field, and not to an ipsilateral limb movement plan per se. The putative human homolog of PRR, though similarly contralaterally biased, also shows some degree of bilateral activation[15,16]. We hypothesize that information about the reach plan for each arm is exchanged across hemispheres in the parietal cortex, and particularly in PRR so that each PRR has information about what the other arm will do and can adjust its own plans accordingly.

We found that local field potentials (LFP) are coherent across hemispheres, with the extent of the coherence depending on the type of reach being planned. This is consistent with information about reach plans being exchanged across the hemispheres. Since spiking activity does not contain ipsimanual arm information, but beta-band LFP power does, we suggest that this information originates from PRR in the opposite hemisphere. This is supported by our finding of interhemispheric spike-LFP coherence in the beta band that, like LFP–LFP coherence, is modulated by the type of reach being planned. Interhemispheric spike-LFP coherence is maximized when the LFP is lagged compared to spikes by about 15 ms, consistent with the input to one hemisphere (LFP) being driven by the output (spikes) from the opposite hemisphere. Altogether, our results suggest that bimanual reach planning is achieved in part by the interhemispheric transfer of information at the level of the parietal cortex.

## Results

We recorded single units and LFP in each hemisphere of two monkeys to look for evidence of interhemispheric exchange of information during the planning of bimanual limb movements. Animals performed saccade, unimanual, and bimanual reaches on interleaved trials. On each trial, animals were instructed to prepare a movement to one or two spatial targets and then cued to initiate that movement after a variable delay of 1250–1750 ms. Unimanual reaches were made with either the right or left arm, and electrophysiological responses were sorted based on whether the arm was ipsilateral or contralateral to the recording site. On

bimanual reach trials, both arms reached the same target (bimanual-together) or each arm reached a different target at diametrically opposed positions (bimanual-apart) (Fig. 1a, b and Supplementary Fig. 1). For bimanual-apart reaches, the arms could be uncrossed or crossed (Fig. 1b). Reaches were accompanied by saccades. A fifth interleaved condition consisted of a saccade without a reach, as a control to isolate any confounding effects of saccadic eye movements by themselves. We recorded LFP from 312 sites in PRR (133 from MkT and 179 from MkZ) and recorded single units from 113 of those sites (43 from MkT and 70 from MkZ) (Fig. 1c). There were no clear differences in the population average single-unit activity of the cells recorded in each anatomical area, with respect to the hypotheses or conclusions of this study (Supplementary Fig. 2). Similar results were obtained in an independent set of recordings that were performed and published several years earlier from the same animals[2]. Overall behavioral performance was good (Supplementary Table 1 and Supplementary Table 2). The median movement times were 170 and 211 ms for MkT and MkZ, respectively. The timing was consistent from trial to trial; the median standard deviations were 22 and 32 ms, or 13 and 15% of the mean. For bimanual movements, the start and end times of reaches with the two arms were similar. In 80% of bimanual-together trials, the two arms began moving within 66 and 67 ms of each other (MkT and MkZ, respectively) and ended within 78 and 75 ms of one another. For bimanual-apart trials, these values were 81 and 91 ms, and 90 and 156 ms, respectively. For a more complete treatment of behavioral performance in these tasks, see[20].

**Interhemispheric LFP–LFP coherence is frequency- and task-specific**. We first tested the hypothesis that there is interhemispheric sharing of information between the two PRRs that depends on the pattern of planned arm movement. Shared information can be identified by an increase in shared variance. Coherence is a measure of shared variance that, unlike correlation, is robust to small-time lags produced by conduction delays.

We measured interhemispheric LFP–LFP coherence as a function of frequency and task during the planning/preparatory period 650–1150 ms after target onset. We made the following predictions. (1) When only one arm will move, then bimanual interactions do not occur, and no exchange of information is necessary. Therefore, coherence should remain at or near the baseline. (2) When both arms move towards the same target and must reach that target at a similar time (bimanual-together task), both spatial and temporal coordination of the two limbs is required. This bimanual interaction will require an exchange of information across hemispheres. If this exchange involves PRR, then we expect that interhemispheric coherence will increase, relative to a unimanual reach. (3) When each arm moves toward a different target, that is, if the reaches are spatially dissociated (bimanual-apart task), it is useful to functionally decouple the activity of the two hemispheres. One way that could happen, or one marker of that happening, would be a decrease in interhemispheric coherence. We, therefore, predict that the bimanual-apart task will not result in increased coherence relative to a unimanual reach, and the requirement for independent spatial control of each arm might even lead to a decrease in interhemispheric coherence.

Prior to the delivery of a task instruction, interhemispheric (across hemispheres) LFP–LFP coherence is frequency-dependent, ranging from 0.17 to 0.24 on a scale from completely independent (0.00) to completely coherent (1.00) (Fig. 2, black trace). During saccade-only and unimanual reach trials, LFP–LFP coherence remained close to baseline across a wide range of frequencies (black, yellow, and gray traces, respectively),

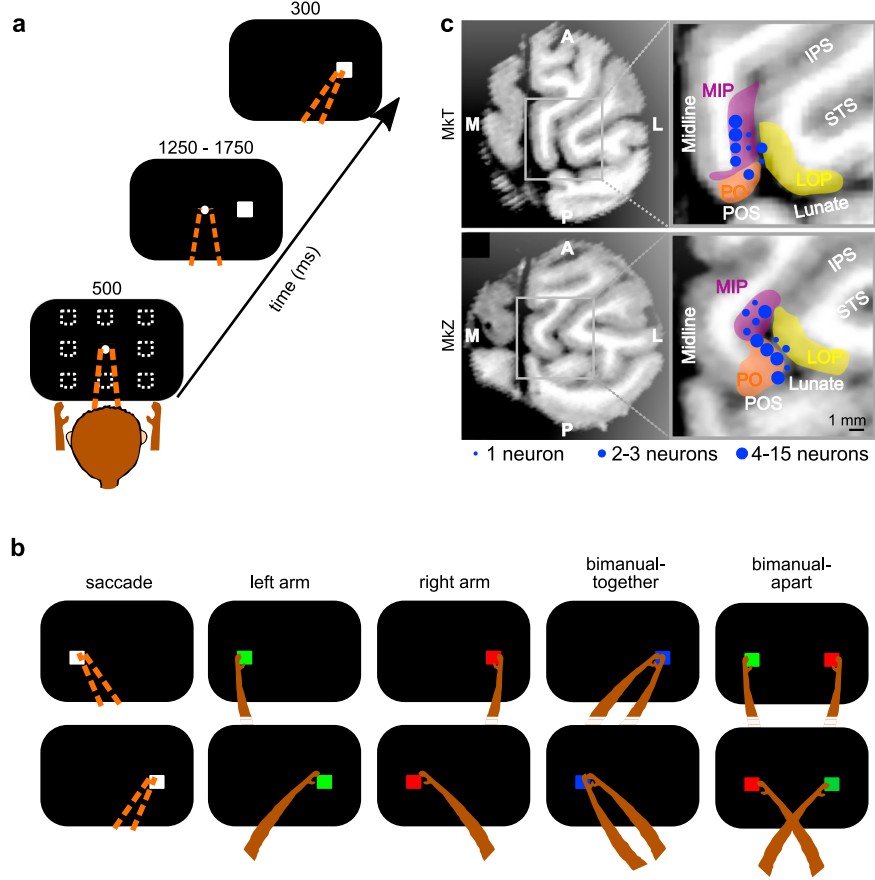

**Fig. 1 Delayed movement tasks and recording sites. a** On each trial, after an initial fixation, a peripheral target appeared and instructed the spatial location and effector to be used (eyes or arm[s]) for the subsequent movement(s). The stimulus remained visible during a variable delay period. Throughout saccade and unimanual reach trials, the hand(s) that were not instructed to move were required to remain on the home pad(s). After the go cue (fixation offset), animals made the instructed movements to the target location(s). On single-target trials, eye movements to the target were required. Movements were either into or 180 degrees out of the RF. On bimanual-apart reach trials, eye movements were unconstrained once the animals were cued to initiate the movement. One arm moved into the RF and the other arm moved out of the RF on each trial. Movement directions and movement types were randomly interleaved. A saccade-only trial (white stimulus) is depicted. **b** Unimanual left or right arm reaches were instructed with a single green or red peripheral target, respectively. Reaches with both arms to a single target (bimanual-together) were instructed with a blue stimulus. Reaches with each arm to a different target (bimanual-apart) were instructed with one red and one green stimulus separated by 180 degrees across the central fixation. Unimanual reaches were either to targets on the same side of the body (upper row) or crossed to the opposite side of the body (lower row). Bimanual-apart reaches were made with the arms either uncrossed (upper row) and or crossed (lower row). Note that only one of the 4 possible target pairs is illustrated. **c** Recording sites from the right hemisphere of each monkey. Coordinates of recorded cells in MkT (upper row) and MkZ (lower row) are shown projected to a single MRI section perpendicular to the path of the recording electrode, with zoomed-in views on the right. IPS, intraparietal sulcus; Midline, longitudinal fissure; POS, parieto-occipital sulcus; STS, superior temporal sulcus. The colored regions are from[89]: LIP, lateral intraparietal area; LOP, lateral occipital-parietal area; MIP, medial intraparietal area; PO, parietal-occipital area. The medial, lateral, anterior, and posterior directions are labeled as M, L, A, and P, respectively. The size of each circle indicates the number of cells recorded along that track. LFP recordings were obtained at these locations and other sites within 2 mm. Left hemisphere sites (not shown) are similar.

supporting prediction (1) above. During bimanual-together trials, LFP–LFP coherence was significantly elevated compared to unimanual trials at 28–32 Hz, supporting prediction (2) (blue). During bimanual-apart trials, LFP–LFP coherence was significantly depressed compared to bimanual-together trials at 22–28 Hz (purple), supporting prediction (3). Similar effects were found in both animals (Supplementary Fig. 3). The effect first appears ~250 ms after the reach instruction is delivered (Supplementary Fig. 4).

**Local power is frequency- and movement type-specific.** Next, we tested whether the amplitudes of within-hemisphere LFP oscillations depend on the type of reach that will be made. LFP power (amplitude squared), computed during the preparatory

period as a function of frequency on each trial and averaged across trials, varies with planned movement type and with LFP frequency (Supplementary Fig. 5). Task-specific modulation is most prominent in the frequency range from 12 to 30 Hz. To isolate the effects of movement type independent of frequency, we normalized by dividing by baseline power at each frequency, subtracting 1, and then plotted the result as a function of time.

This analysis revealed that LFP power in the beta range (20–30 Hz) contains substantial information about bimanual and ipsilateral arm movements (Fig. 3a; see also Supplementary Results). Different movement plans were associated with differences in the power of 10–20% (all $P < 0.05$ even after correction for the 10 possible comparisons). Beta power was not tuned to the directional preferences of nearby spikes (Supplementary Fig. 6). This contrasts with single units, whose firing

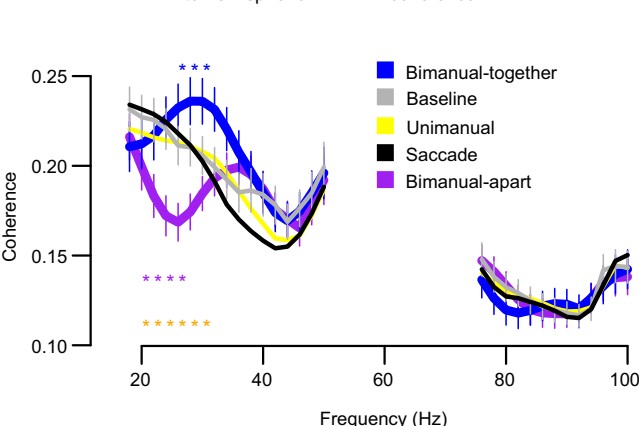

**Fig. 2 Beta-band LFP–LFP coherence between PRR in the left and right hemispheres distinguishes bimanual-together and bimanual-apart movements from baseline and from unimanual movements.** Coherence in the beta band (~20–30 Hz) is elevated for bimanual-together movements (blue) and decreased for bimanual-apart movements (purple) compared to unimanual reaches (yellow) or to the baseline period (gray). Coherence during saccade trials (black) resembled that seen during unimanual reach. Data are averaged from the 113 pairs of sites (43 from MkT, 70 from MkZ) recorded simultaneously in the two hemispheres, with coherence measured in the −500–0 ms interval before the cue to move. Blue and purple asterisks indicate significant differences of bimanual-together and bimanual-apart, respectively, versus unimanual. Gold asterisks denote comparison of bimanual-together versus bimanual-apart (two-sided t-tests, large asterisks = $P < 0.05$ after Bonferroni correction for testing at each of the 39 different frequency values plotted in the figure). Coherence outside the pictured range (from 16 to 20 Hz and 100 to 240 Hz) was unaffected (corrected $P > 0.05$). Error bars = ±1 s.e.m. Source data are provided as a Source Data file.

rates distinguish only target location and the pattern of contralateral arm movement at the population level, and even at the single-unit level show only sporadic and non-systematic effects of bimanual movements (Fig. 3c; Supplementary Fig. 7; see Supplementary Fig. 8a for individual animals; see also ref. [2] for a detailed analysis of these effects).

Beta-band power is strongly suppressed even before the target appears and then slowly recovers over the course of the delay period. The recovery is fastest and most complete for saccades, followed in order by ipsimanual reaches, contramanual, bimanual-together, and bimanual-apart. This order is consistent across recording sites (Supplementary Fig. 9). Interestingly, task-specific effects appear only ~250 ms after target onset in the LFP data, ~100 ms later than in the spiking data (compare divergences in Fig. 3a, c). This observation implies that modulations of firing rate by the task are not driven by modulations in beta power. Instead, the modulation in firing rate may drive the modulation in LFP. We return to this point in the next section.

Beta power drops after the go cue, reaching a nadir ~100 ms prior to reach onset (Supplementary Fig. 10, middle). Subsequently, the power for trial types that include a contralateral arm movement (contramanual, bimanual-together, and bimanual-apart) rebounds and has a positive slope, up to the time of reach onset. The pre-movement increase in LFP beta power is less clear when the traces are aligned on either the go cue or saccade onset (Supplementary Fig. 10, left and right). This suggests that LFP power is closely related to contralateral limb reach initiation. Indeed, there is no late increase in LFP power with ipsilateral reaches or saccades.

In contrast to beta-band power, we found that normalized gamma power (>70 Hz) is similar to unit activity (Fig. 3b, c and Supplementary Fig. 8b, c), consistent with previous work suggesting that gamma-band activity is a direct correlate of local spiking activity[21]. For both spikes and LFP, there is an anticipatory increase in power prior to target onset. Preferred and null direction responses diverge after target appearance. The LFP divergence appears to be synchronous with target onset, but power is computed over a ±100 ms window so the actual divergence occurs within 100 ms of target appearance, similar to what is seen in single units. Also similar to single units, there is a second divergence at ~150 ms. Trials in which the contralateral arm will move in the preferred direction (contralateral only, bimanual-together, and bimanual-apart trials) result in the highest response, while ipsilateral only and saccade trials result in intermediate responses. Within the intermediate and high response groups, the early responses to different upcoming movement types are not statistically different from one another (50–350 ms after target appearance, paired t-tests of adjacent traces from each site with an isolated single unit, 113 sites, all $P > 0.1$ after multiple comparison correction [and all but one $P > 0.1$ prior to correction]). Adding an ipsilateral arm movement to a contralateral arm movement has little effect on gamma power. That is, the bimanual-apart and bimanual-together responses resemble the contramanual only response. A parsimonious explanation of the activity on ipsimanual trials is that it is a response to a target appearing in the response field. This explanation is consistent with the fact that activity on ipsimanual trials is very similar to that on saccade trials, and mirrors the conclusions drawn concerning single-unit responses[2]. Thus, while beta power carries information about the movements of each arm, gamma LFP power, like unit activity, carries relatively little information about ipsilateral arm movement.

In summary, beta power differs dramatically from the unit activity and from gamma power in at least four ways. Most importantly, units and gamma power show just two levels of activity for preferred direction movements based on the type of movement to be performed: high levels for the contralateral arm and low levels for both the ipsilateral arm and the eye. In contrast, beta power shows a distinct level of activity for each of the five-movement types. Second, beta power, unlike gamma, is not directionally tuned. Third, the ordering of response magnitudes is inverted for beta power compared to gamma power and units. For example, saccades and ipsilateral arm movements are associated with the lowest level of spiking and gamma power, but the highest level of beta power. Finally, movement plans affect beta power only ~100 ms after they affect gamma power and single-unit activity. While temporal differences must be interpreted cautiously given the temporal smoothing in the beta LFP, this smoothing will mainly shorten the LFP latencies and so cannot be responsible for the divergences in LFP power lagging the divergences in the single units.

We have shown LFP power at 20–30 and 70–120 Hz, but task effects are continuous across frequencies, with at least two distinct regimes separated by a transition zone (Fig. 4 and Supplementary Fig. 11). Above 50 Hz, responses to preferred direction movements depend strongly on movement type and only weakly on frequency; similar power levels are attained for a given movement type at 70 and 170 Hz. Power is elevated ~16% above baseline for any task that includes a contramanual reach, is at baseline for ipsimanual reaches (−1%), and is slightly suppressed for saccades (−9%). The pattern for null direction responses is more complex. Power is elevated for bimanual-apart, near the baseline for bimanual-together, and suppressed for contramanual reach, ipsimanual reach, and saccades.

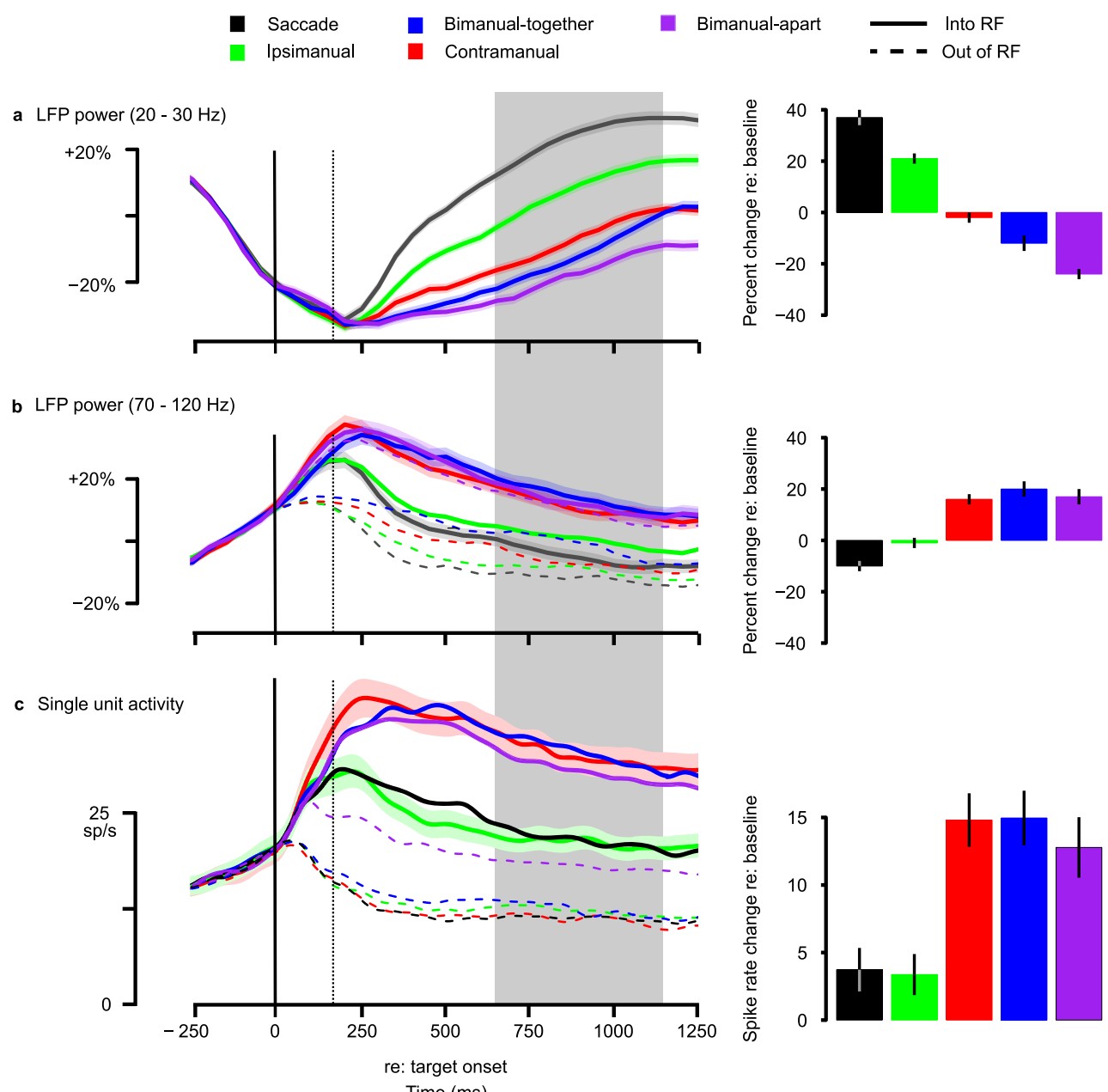

**Fig. 3 A comparison of neural activity during 5 movement conditions.** Data on the left are aligned to target onset (solid vertical line) and truncated at the time of the earliest cue to initiate movement. Data on the right show the mean activity for preferred direction movements, relative to baseline, in the interval from 650 to 1150 ms (gray rectangle). **a** Beta-band LFP power (20–30 Hz) contains information about movement type. Power is computed in ±200 ms intervals every 100 ms. Colored shading indicates ±1 s.e.m. Responses to preferred and null directions showed no difference and so are merged across $n = 312$ sites. **b** Gamma-band LFP power (70–120 Hz) is similar but not identical to unit activity and unlike the beta band contains little additional information. (See text for additional detail.) Format as in **a** except that preferred and null directions are shown separately. Power is computed in ±100 ms intervals every 50 ms. **c** Single-unit activity is high when the animal prepares a contramanual reach in the preferred direction, whether alone (contramanual, solid red line), with an ipsimanual reach in the same direction (bimanual-together, blue), or with an ipsimanual reach in the opposite direction (bimanual-apart, purple). Firing is intermediate for preferred direction saccades and for preferred direction ipsimanual reaches (solid green and black), and for bimanual reaches in which the ipsilateral arm moves in the preferred direction and the contralateral arm moves out (bimanual-apart, dashed purple). (Bimanual-apart reaches are labeled based on the direction of the contralateral arm.) Activity is low for single-target movements in the null direction (dashed lines) during the delay period, independent of the movement type. Note that the divergence of firing rate associated with movement type (dotted vertical line) occurs ~100 ms sooner than the divergence in beta LFP power associated with movement type. Data from $n = 113$ single units. Colored shading indicates ±1 s.e.m. Source data are provided as a Source Data file.

A second regime occurs at lower frequencies, from about 16 to 32 Hz. Here LFP power depends strongly on both movement type and frequency, but only weakly on the direction. Responses are not clustered but instead are ordered by movement type. The point of minimum power occurs at a low frequency for saccades (≤8 Hz), at an intermediate frequency for ipsimanual reaches (~8–16 Hz), and at still higher frequencies for contramanual, bimanual-together, and bimanual-apart reaches (12–20 Hz).

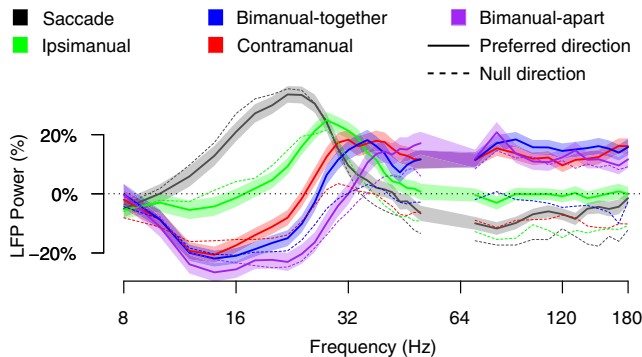

**Fig. 4 LFP power as a function of movement type and frequency.** LFP power is computed over the delay period for movements in the preferred and null directions. In the 16–32 Hz range power depends on task and frequency. In the 70–170 Hz range power depends on the task and is relatively stable across frequency. Data are for the 113 sites from which a tuned unit was simultaneously obtained. Source data are provided as a Source Data file.

From 32 to 50 Hz there is a transition zone in which the effects of task on power change rapidly with frequency. Sensitivity of LFP power with task disappears almost entirely at ~36 Hz and also at ~8–10 Hz. Below 8 Hz, the measurement interval (500 ms) is too short compared to the wavelength being measured to obtain reliable estimates of power (see "Methods" section).

**Interhemispheric spike-LFP coherence is frequency- and task-specific.** We have shown that beta power encodes substantial information about both contralateral and ipsilateral arm movement plans, while spikes primarily encode information about the contralateral arm. Where does the LFP information about the ipsilateral arm come from? There are prominent connections between homotopic cortical areas in the right and left hemispheres[22]. Thus, one possible source of left arm information in left PRR is signals from right PRR, and vice versa. The fact that task effects in the beta power lag effects in the spikes (Fig. 3a, c) is consistent with this assumption. We investigate this possibility further using lagged interhemispheric spike-LFP coherence.

LFP is thought to be generated primarily by dendritic currents[23–26]. Therefore it reflects not only local recurrent signals but also distal input, while spiking activity carries local recurrent signals plus local output[27,28]. If some axons from the left PRR travel to the right PRR, then some of the spikes recorded in the left PRR may help generate dendritic currents in the right PRR (Fig. 5a). These dendritic currents might help drive LFP in left PRR. If so, then we might find that spikes from left PRR are coherent with LFP from right PRR, particularly in the beta band (Fig. 5b, black). Since the PRR spike rate depends on what type of movement is being planned, movement plans that evoke higher spike rates might drive differing amounts of spike-LFP coherence. Alternatively, if spikes from PRR in one hemisphere do not drive LFP in the other PRR, then spike-LFP coherence should be minimal.

Interhemispheric spike-LFP coherence is elevated above chance levels during the delay interval for some tasks and at some frequencies (Fig. 5c). In particular, spike-LFP coherence at ~20–50 Hz is elevated for bimanual-together movements ($P <$ 0.01, permutation test) and elevated somewhat less for unimanual movements ($P < 0.05$ permutation test). Coherence on bimanual-apart trials was not significantly different from chance, and in fact was less than that seen with unimanual movements, although the suppression was not significant. Spike-LFP coherence for bimanual-together movements was significantly higher than for

bimanual-apart movements over most of the beta range (two-tailed $t$-test; asterisks in Fig. 5c). Similar results were obtained using the pairwise-phase consistency (Supplementary Fig. 12). This difference could not be attributed solely to strongly oscillating LFP signals during bimanual-together compared to bimanual-apart trials, since the elevation also occurs in phase-locking value. (Phase-locking values take into account only spike timing and LFP phase, not LFP magnitude; see "Methods" section). An alternate scenario is that common input to the two hemispheres is responsible for interhemispheric coherence. In this scenario, common input drives similar LFP on each side, and the similar LFP drives similar patterns of spikes (Fig. 5d). Thus spikes on one side reflect LFP on the other side, even though there is no direct (causal) link and the resulting spike-LFP relationship is the same as with direct interhemispheric communication (Fig. 5b, black versus gray).

If LFP reflects synaptic input, and if synaptic input in PRR is driven in part by spikes from PRR in the opposite hemisphere, then we might expect that the greatest effect of contralateral spikes on ipsilateral PRR activity would occur only after a short conduction delay (Fig. 5e, black). In the common input scenario, however, since LFP drives spikes, the greatest coherence should occur when spikes are lagged relative to LFP, rather than vice versa as in the first scenario (Fig. 5e, gray).

We computed interhemispheric lagged spike-LFP coherence and found that peak spike-LFP coherence occurs during bimanual-together trials when LFP is lagged by ~10 ms relative to the spikes (Fig. 5f). During bimanual-apart trials, spike-LFP coherence reaches a nadir when LFP is lagged by ~15 ms relative to the spikes. These results are consistent with spikes from one PRR driving LFP in the other PRR, with a conduction delay occurring as spikes travel from one hemisphere to the other.

**An algebraic model of interhemispheric signal transfer explains the beta frequency LFP responses.** The spike-LFP coherence results suggest that task effects in beta LFP could be driven by a combination of local signals—spikes from the ipsilateral PRR—plus signals from other areas, including spikes from the contralateral PRR (Fig. 5). An alternative hypothesis is that beta LFP arises exclusively from local (ipsilateral) signals. We compared these two alternatives using a simple algebraic model. A model based on 80% local input with the remainder coming from contralateral PRR (Supplementary Fig. 13, red) fits the observed data (dashed black) much more closely than did the model based on 100% local input (blue). See Supplementary Information for details.

## Discussion

This study reveals mechanisms of bimanual coordination in the posterior parietal cortex. Our findings support the hypothesis that the task-specific changes observed in beta-band LFP (PRR input) are driven, in part, by spikes (PRR output) from the opposite hemisphere. First, interhemispheric spike-LFP coherence in the beta band rises while planning bimanual-together movements and falls while planning bimanual-apart movements. Second, there is a reliable relationship between spikes in one hemisphere and LFP in the other (Fig. 5c), consistent with direct communication between the left and right PRR. Third, this relationship is strongest when spikes are compared with LFP that occurs 10–15 ms later in time (Fig. 5h). Our findings also demonstrate that beta-band LFP power contains information about planned movements of either arm that is not present in the population average spiking output (Fig. 3). Taken together, the data indicate that signals describing how the contralateral arm will move are sent from one PRR to the other and that these signals are

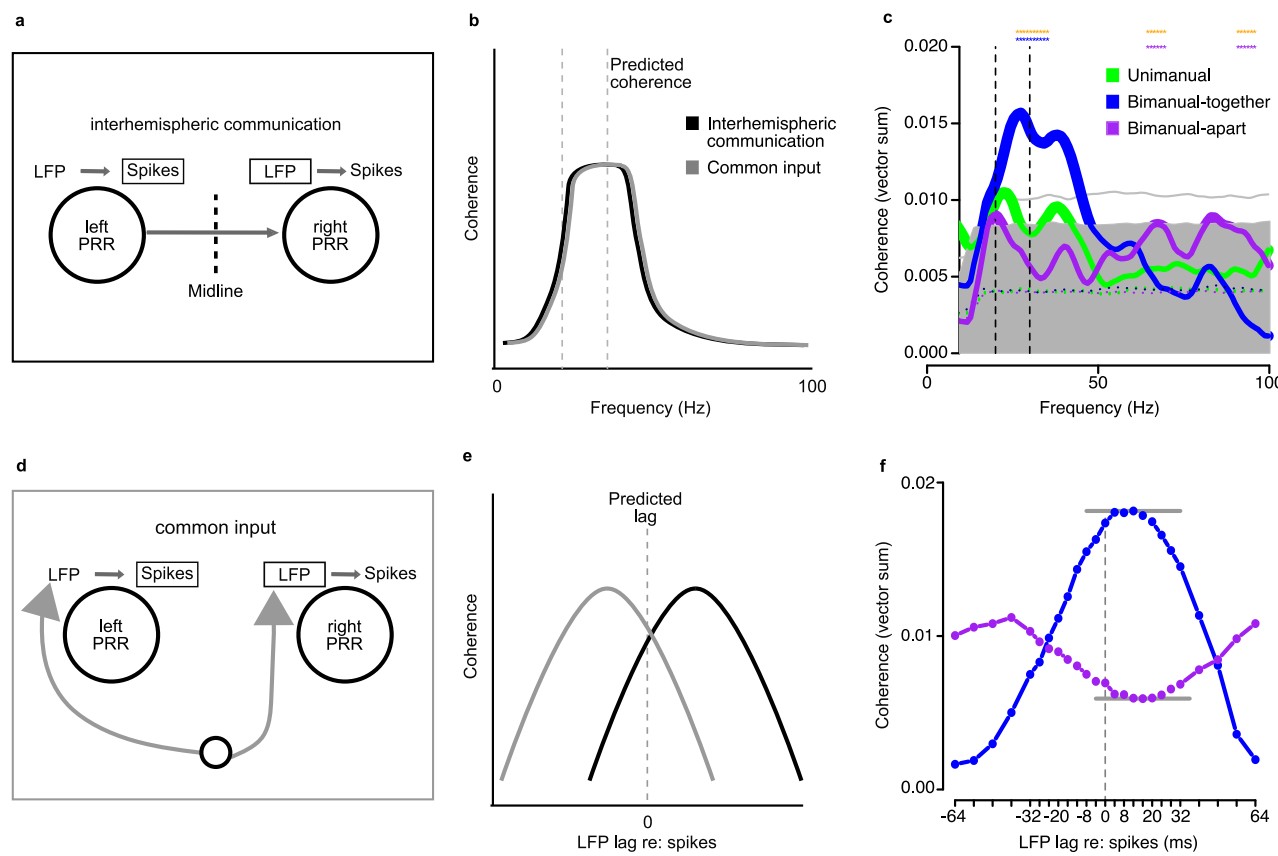

**Fig. 5 Interhemispheric beta-band spike-LFP coherence distinguishes between bimanual-together and bimanual-apart movements. a** Schematic model depicting local and distal inputs and outputs to and from PRR. Mass input and output of PRR is shown as arrows. The axon terminals of most neurons contact other neurons locally (light gray arrows), but some portion project distally, including to the homotopic area in the opposite hemisphere (dark gray arrows). Connections with non-homotopic areas are omitted for clarity. **b** Identical spike-LFP coherence predictions under interhemispheric communication scenario (black curve) and common input scenario (gray curve). **c** From 20 to 50 Hz, spike-LFP coherence is consistently high for bimanual-together movements (solid blue), intermediate for unimanual movements (solid green), and low for bimanual-apart movements (solid purple). The distributions of coherence expected by chance were computed by shuffling interspike intervals. The medians of these distributions are shown as dashed traces. The gray shaded region covers 95% of the values expected by chance; values that exceed this are marked by thickened lines ($P < 0.05$). Values exceeding the light gray line are significant at $P < 0.01$. Note that there is no effect of movement type on the shuffled coherences. For this reason, data are pooled across movement types for the $P < 0.05$ and $P < 0.01$ thresholds. Coherences that are significantly larger for bimanual-together compared to bimanual-apart are indicated by asterisks (two-tailed $t$-test, $P < 0.05$). Dashed vertical lines indicate the beta range (20–30 Hz). Coherence was measured during the 800 ms before the go cue. Data are averaged from $n = 42$ pairs of sites (24 from MkT, 18 from MkZ) recorded simultaneously in the two hemispheres. Only sites with at least 500 spikes are shown. **d** Schematic model depicting common input to PRR in each hemisphere. Spike-LFP coherence predictions for common input model. **e** Lagged spike-LFP coherence predictions for direct communication and common input models, respectively. **f** Extremes of spike-LFP coherence (24–38 Hz) occur when spikes lead LFP by 10–15 ms (gray lines). Positive and negative $x$ axis values indicate the relative temporal relationship between spike times and the LFP in the original data. Source data are provided as a Source Data file.

manifest in the beta-band LFP of the receiving hemisphere. These signals are likely to be used by PRR to adjust planned movements of the contralateral limb in the service of bimanual coordination.

Beta oscillations are prevalent throughout the motor system and are modulated in reach tasks[29,30]. In many areas, beta oscillations are enhanced during the hold and instructed delay periods and during movement preparation, are suppressed around the time of movement, and rebound in the post-movement and reward periods[31–33]. Enhanced beta power during delay periods, prior to overt movement, has been attributed to coordination[34], attention[31], movement preparation[35], and maintaining a motor plan[32], but effects vary by task and by area[33,36–40]. In PRR, task-specific effects previously observed include an increase in beta power for planning reaches compared to saccades[38], more pronounced changes in power for reaches performed with an accompanying eye movement compared to without one[41], and both frequency- and monkey-specific differences in responses to memory-guided compared to visually-guided reaches[42].

Like these studies, we find that LFP power is strongly and differentially modulated by task and by frequency, especially at frequencies in and around the beta range (Fig. 4). Beta power initially decreases, falling nearly in half over the period from 350 ms before to 200 ms after target onset, and then rises to a plateau whose level depends on the task that has been instructed (Fig. 3a). Whether beta power is described as enhanced or suppressed in any particular task depends on the choice of a baseline time interval. This choice may have had a similar influence in previous studies[38,41]. Therefore, rather than characterize the observed responses as enhancements or suppressions relative to baseline, we focus instead on their rank order. Delay period power was most suppressed for movements of both arms to two different targets, followed in order by movements of both arms together to the same target, contralateral arm movements, ipsilateral arm movements, and saccades (Fig. 4). Dean et al.[43], also observed beta power that was suppressed more for reaches plus saccades than for saccades alone. In contrast, Scherberger et al[38].

observed the reverse effect: beta power was greater with contralateral reaches than saccades. This difference may reflect differences in task designs. We used a random-delay visually-guided task, whereas Scherberger et al. used a fixed-delay memory task. Power in visual versus memory tasks has been observed to vary by task and even by monkey[42]. In addition, we allowed coordinated saccades with our reaches while Scherberger et al. did not, and we used 5 interleaved trial types while Scherberger et al. used 2.

The initial drop in beta power, prior to the first stimulus, could reflect the fact that the animal is about to receive a task instruction. The drop may be more pronounced than that seen in previous studies because of the larger number of possible tasks that could be instructed. One could argue that this is consistent with beta power reflecting the maintenance of a motor plan: when a new plan is expected, beta power drops[44,45]. Note, however, that beta power also drops shortly after the go cue and rebounds as movement is initiated (Supplementary Fig. 10). The first drop in power, at the time of target instruction, occurs while the movement plan is changing and motor output is held constant. The second drop in power, at the time of the go cue, occurs while the movement plan is held constant but the motor output is changing. We conclude that elevated beta power has no single correlate that applies across all areas, conditions, and times.

Beta power differed from spikes in its response to movement type and direction. Unlike the population-averaged firing rate, beta power was not tuned to the type of movement that was planned. The dendrites from which beta LFP most likely arise are driven by a mixture of inputs from local and distal sources. In contrast, although we record spikes exclusively from local neurons and their proximal axons, those axons may project to either local or distal targets. Thus, beta LFP reflects local processing plus distal inputs, while spikes reflect local processing and distal outputs. In addition to this structural constraint, beta rhythms in particular may be biased to reflect distal input. This is because oscillations at beta frequencies are more likely to occur in conjunction with total axonal loop delay times greater than 20 ms, which will occur more often for axons coming from distal rather than local sources[46]. Incoming signals from distal sources drive dendritic currents but only indirectly influence spiking outputs. This could explain why ipsimanual reach signals are explicitly represented in beta LFP but not in the local population-averaged firing rate. Since we sample only a fraction of the ~100,000 cells per cubic millimeter found in the cortex, and the fraction that we sample is likely biased toward large pyramidal cells[44,47], it is possible that signals related to ipsimanual movements may be prominent in the spiking output of cells that are too small for us to isolate.

Although not present in the population average, individual cells do show idiosyncratic effects of ipsilateral arm movement plans. We speculate that aspects of a contralateral movement plan must be modified when an ipsimanual component is added. For example, proximal musculature related to posture may depend on whether the ipsilateral arm will remain at rest or move in the same or opposite direction as the contralateral arm. The contralateral movement trajectory and hand posture might also be subtly altered, depending on whether both arms are approaching the same target[45]. These subtle changes would not be systematic across different patterns of coordination or movement direction, and so would tend to disappear in the population average.

The fact that the ipsimanual arm only minimally affects the spike rate is partly the result of our task design. We intentionally designed tasks to minimize interaction effects between the two arms so that any difference in activity between conditions could be distinguished as reflecting either the ipsilateral arm movement or a higher-order bimanual reach coordination signal. A task with

substantial interactions between the two arms, e.g., tying a knot using both hands, would almost certainly find a substantial influence of ipsimanual movements on PRR firing, but such an influence could be an effect of bimanual coordination, rather than its cause.

Beta power was not tuned to the directional preferences of nearby spikes (beta power: Supplementary Fig. 6; spikes: Fig. 3c; compare this with gamma-band power, which shows clear spatial tuning). These results confirm and extend Dean et al. (2012) who showed little or no spatial tuning between 20 and 30 Hz and strong tuning, congruent to spikes, at 35 Hz and above for contramanual reaching. The lack of spatial tuning from 20–30 Hz could reflect widespread pooling of signals from many individual cells with diverse directional preferences. Dean et al. also saw a significant reversal of spatial tuning below ~18 Hz, in the lower beta range which was the focus of their study. We also see reversed spatial tuning in the 10–16 Hz range, but the effect is not significant even before multiple comparisons testing. The pooling of individual cell responses does not, however, fully explain beta power. Population-averaged spiking activity is insensitive to trial type, while beta power differentiates each of the five types of movement plans (Fig. 3a versus 3c). (Spiking activity can be sensitive to trial type in individual cells, but the effects are not systematic from one cell to the next and are therefore lost in the population average[2].)

Local pooling of effects across cells and cell processes can explain why beta power does not depend on movement direction, but pooling cannot explain the observed tuning for movement type. We found no topography or even coarse clustering of movement type sensitivity in single units (unpublished observations). Even if clustering exists, then local pooling would produce different patterns of beta power sensitivity at different recording sites, a pattern that we did not see (Supplementary Fig. 9). It is therefore noteworthy that in our model, in which beta-band LFP is driven by spikes from the opposite hemisphere, both observations are accounted for: the model-derived beta power depends on movement type but not on movement direction (Supplementary Fig. 13 and Supplementary Table 3).

Our model could be implemented using inputs from areas other than contralateral PRR such as the contralateral premotor cortex. Given that many different hemispheric inputs to PRR exist[22,48], it is highly likely that at least some of the ipsilateral information originates from these areas. However, the finding that interhemispheric LFP coherence reflects task type suggests that either contralateral PRR is a major source of input, or that areas such as the premotor cortex that provide ipsilateral input project to both ipsilateral and contralateral PRR.

The coherence results support the idea that interhemispheric interactions are responsible for the pattern of beta LFP results. Interhemispheric beta LFP–LFP coherence (Fig. 2) and spike-LFP coherence (Fig. 5) are movement-specific. A plan to move both arms to the same target increases beta-band coherence while planning to move each arm to different target results in a decrease in beta-band coherence, compared to baseline and unimanual reach levels. There were no modulations of LFP–LFP coherence outside the beta band. Similarly, a plan to move both arms together increased spike-LFP coherence compared to a plan to move each arm to a different target. Again, this was most prominent in the beta band. These results cannot be explained by task difficulty, since coherence did not track the presumed effort required to make each movement. The saccade-only condition is likely easy to perform yet has intermediate coherence, while the bimanual-apart condition is difficult yet has the lowest coherence.

It appears that information exchange between the hemispheres is effectively facilitated with bimanual reaches in the same direction (bimanual-together) and effectively inhibited with

bimanual reaches in opposite directions (bimanual-apart). Increases or decreases in information exchange are likely to facilitate yoked or independent movement of the two arms, respectively. Similar results have been found in the motor cortex during the movement period itself[49]. Temporal coordination is required in both our bimanual tasks, but our results indicate that it is the spatial aspects of the task that are responsible for differences in the interhemispheric exchange of information. If temporal coordination were the key aspect then we would expect increased exchange in both bimanual-together and bimanual-apart tasks[50,51].

Two pieces of evidence suggest that spikes drive LFP instead of the reverse. First, task coding emerges sooner after target onset in the spikes (~125 ms after target onset) compared to the fields (~250 ms). Second, peak spike-LFP coherence occurs when LFP lags spikes by 10–15 ms. This lag is consistent with a spike originating in one hemisphere affecting LFP in the other hemisphere only after a short delay. Notably, the ~125 ms difference in task coding onset between spike and beta power measurements is an order of magnitude greater than the 10–15 ms peak lag in spike-LFP coherence. This apparent discrepancy can be explained by the fact that the two measurements capture different phases of the task. Spike and LFP power reflect the onset of task coding, e.g., the presence of a visual stimulus in the receptive field. Spike-LFP coherence, on the other hand, reflects the emergence of a later, stable communication between the hemispheres.

Each cerebral hemisphere primarily controls the limbs on the contralateral side of the body. Single-unit recordings and reversible inactivation indicate that this lateralized limb control is implemented as early as PRR[2,18]. Yet, human functional magnetic resonance imaging studies report bilateral parietal blood-oxygen-level-dependent (BOLD) responses for reaching[17,52,53], and arm movements can be decoded from electrocorticogram (ECoG) signals from the ipsilateral cortex[54]. The BOLD signal, like LFP and ECoG, likely reflects areal input and intracortical processing[55]. Thus, the difference between the single-unit activity and functional imaging results fits with our proposal. Measures that are sensitive to outputs, like single-unit spiking, capture lateralized motor activity in the early reach pathway, whereas measures that are sensitive to inputs, like LFP, capture information about movements on both sides of the body.

Dorsal premotor and M1 neurons are active during contralateral and ipsilateral arm movements[52,53,56]. Bimanual movements are not coded as the linear sum of the activations of the left and right arm in the SMA[4], M1[3,4], or parietal cortex[2,5]. This arrangement may allow lateralized cortex to command movement plans of the contralateral arm dependent on the state of the ipsilateral arm. Controlling bimanual movements is a particular challenge in the development of brain-machine interfaces for the restoration of motor function after paralysis[57]. Simultaneous recording of spikes from both hemispheres shows promise for controlling two limb prostheses at once[58]. LFPs offer advantages over spikes in terms of signal degradation[59,60] and the combination of spikes and low-frequency LFP improves performance beyond spikes alone[61]. Our results indicate that, in order to control both arms from signals from just one hemisphere, it is critical to record beta band LFP power, either with or without spikes.

Competing theories of callosal function focus on either its excitatory[62] or inhibitory[63] role in interhemispheric processing and there is evidence for both in the human literature[64]. Our data indicate that interhemispheric connections can be functionally either inhibitory or excitatory, depending on the particular task being planned. These data do not speak to whether interhemispheric transmitters themselves are excitatory or inhibitory[65].

Chronic hemiparesis after unilateral stroke has been attributed to an imbalance of interhemispheric inhibitory interactions[66]. This hemispheric competition hypothesis posits that tonic inhibition between the left and right cortex is disrupted by unilateral lesions[63]. This hypothesis is supported by tonic pre-movement interhemispheric inhibition (IHI) from the intact to the lesioned hemisphere in chronic stroke as measured by transcranial magnetic stimulation[67]. As a result, downregulation of excitability of the intact hemisphere to restore interhemispheric balance has become a target of stroke rehabilitation[68]. However, our results do not support this hypothesis. We find that interhemispheric interactions in intact subjects do no change shortly before movement onset—in particular, LFP–LFP coherence was unchanged prior to a unimanual reach compared to baseline (Fig. 2 and Supplementary Fig. 3). Maladaptive recruitment of the intact hemisphere after unilateral lesions has been recently challenged, however, on the basis that release of IHI prior to unimanual movement onset is normal in acute stroke and only becomes abnormal in chronic stroke[69,70]. Such findings raise questions about the efficacy of targeting IHI to restore function in the weakened limb. If the goal of rehabilitation is to improve the function of the paretic limb, our results underscore bimanual coordination training as another potential target for stroke rehabilitation[71–73].

Additional factors may contribute to the observed coherence effects, including visual effects of one versus two targets, spatial effects of one versus two-movement goals, or the effects of having motor plans that are congruent or incongruent. In our analyses we do not consider the absolute side of the spike-field pairs, and hence the directionality of interhemispheric communication, e.g., the laterality of the target in each trial, or whether or not reaches cross the midline. There could be an asymmetry in interhemispheric coherence depending on the specific task conditions (e.g., a left-arm reach toward a right visual field target versus a left-arm reach to a left visual field target). As for an influence of the associated eye movements, we showed previously that the direction of the initial saccade on bimanual-apart trials does not affect PRR unit activity either during planning or movement[74]. Saccade direction also does not affect LFP power in PRR (data not shown). Finally, while the lagged spike-LFP results support the hypothesis that there is direct communication between PRR in each hemisphere, we cannot rule out either common input or indirect communication driving the coherence effects. This will require an interventional experiment.

We only recorded from PRR. Other pairs of homotopic cortical motor areas in the two hemispheres may exhibit task-specific modulations in their interhemispheric coherence, and these may facilitate yoked or independent movements of the two limbs. While it is by no means clear that interhemispheric circuit mechanisms should be common across brain regions, our proposal that local beta power in PRR reflects a weighted sum of local (ipsilateral) and distal (contralateral) spike inputs is consistent with recent findings from SMA and pre-SMA. In these areas, gamma power was observed to be stronger before a contralateral compared to an ipsilateral arm movement, whereas beta power showed the reverse effect[56].

We found that information about the two limbs is shared across hemispheres in a frequency- and movement-type-specific manner. In PRR, both interhemispheric LFP–LFP and spike-LFP coherence were modulated during bimanual movement plans. Population average spiking and gamma-band LFP power encode only contralateral arm movement plans, while beta-band LFP power contains a rich representation of the pattern of bimanual coordination. We conclude that the information about ipsilateral arm movements encoded in the beta-band LFP is driven by spikes from the opposite hemisphere, and that increased information

transfer facilitates spatially coordinated movements while decreased transfer facilitates disjunctive movements. More generally, beta LFP power reveals one site at which information about the movements of each arm is shared across the hemispheres.

## Methods

**Contact for reagent and resources sharing.** Further information and requests for resources and reagents should be directed to and will be fulfilled by the Lead Contact, Eric Mooshagian (ericm@eye-hand.wustl.edu).

**Experimental model and subject details.** All procedures conformed to the Guide for the Care and Use of Laboratory Animals and were approved by the Washington University Institutional Animal Care and Use Committee. Two male rhesus macaques (Macaca mulatta), MkT (16-year-old male, 9.0 kg) and MkZ (14-year-old male, 10 kg), were used in the study.

### Methods details

*Apparatus.* Head-fixed animals sat in a custom-designed monkey chair (Crist Instrument, Hagerstown, Maryland) with an open front to allow unimpaired reaching movements with both arms. Visual stimuli were back-projected by an LCD projector onto a translucent plexiglass screen mounted vertically, approximately 40 cm in front of the animal. Supplementary Fig. 1 shows the touch screen in schematic form. The eight target positions on the screen were organized in a rectangle centered on the fixation point, each target ~ 8 cm (11 deg of visual angle) from the center fixation point. A small piece of plexiglass (2 in × 3/8 in) oriented in the sagittal plane was mounted on the front of the projection screen to bisect the touching surface at each target location (one plexiglass piece per target). Touches were monitored every 2 ms using 16 capacitive sensors, mounted on the back of the screen, with one sensor on each side of each of the 8 possible target locations (to sense reach endpoints) and one sensor at each of the two home pads (to sense reach starting positions). The animals were trained to reach with the left hand to the left side of the plexiglass divider and with the right hand to the right side of the divider so that each hand activated a unique capacitive sensor, even when both hands reached to the same target (Supplementary Fig. 1). Eye position was monitored using the 120 Hz ISCAN eye-tracking laboratory (ETL-400). Animals were monitored in the testing room at all times using an infrared camera equipped with an infrared illuminator.

*Behavioral tasks.* The task design and the movement conditions are shown in Fig. 1b. The animals performed delayed saccades or reaches with the left, right, or both arms[20]. Animals first fixated on a circular white stimulus (1.5° x 1.5°) centered on the screen in front of them. Left and right hands touched "home" pads situated at waist height and 20 cm in front of each shoulder. After holding fixation (± 3°) and initial arm positions, for a fixed duration of 500 ms, either one or two peripheral target(s) (5° × 5°) appeared on the screen. Fixation was required throughout the instructed delay period. After 1250–1750 ms, the central eye fixation target shrank in size to a single pixel, cueing the animal to move to the peripheral target(s) in accordance with a specifically trained code conveyed by target color. Reach trials could be unimanual or bimanual. A green target instructed a left-arm reach, and a red target instructed a right-arm reach. Bimanual trials could be to one or two targets. A blue target instructed a combined reach with both arms (bimanual-together). When two targets appeared (red and green, "bimanual-apart"), they were separated by 180° relative to the central fixation point, i.e., on the left and right, top and bottom, or at opposed diagonal locations. For bimanual-apart reaches, the arms could be uncrossed or crossed (Fig. 1b). A white target instructed a saccade (no reach).

All trial types were randomly interleaved within sets of 10 or 40 trials (one each per condition and direction; see below). Throughout saccade and unimanual reach trials, hand(s) not instructed to move were required to remain on the home button(s). On unimanual reach trials, eye movements were constrained to move to the target. On bimanual reach trials, fixation was required up until receipt of the go cue, after which time eye movements were unconstrained. On bimanual trials, the left and right hands were required to hit their target(s) within 500 ms of one another. For single-target trials, the animals were required to maintain their gaze on the final target for 300 ms. For all reach trials, animals were required to maintain their hand(s) on the final target(s) for 300 ms. Spatial tolerances were ±3° for reaches and ±2° for saccades. When an error occurred (a failure to achieve or maintain the required eye or hand positions), the trial was aborted and a short (1500 ms) time-out ensued. Aborted trials were excluded from further analyses. Successful trials were rewarded with a drop of water or juice.

### Electrophysiological recordings.

Recordings were made from the left and right hemispheres of two adult male rhesus monkeys. Recording chambers were centered at ~11 mm posterior to the ear canals and 8 mm lateral of the midline and placed flush to the skull. Anatomical magnetic resonance images were used to localize the medial bank of the intraparietal sulcus. Extracellular recordings were made using glass-coated tungsten electrodes (Alpha Omega, Alpharetta, GA; electrode impedance 0.5–3.0 M ohms at 1 kHz). Neuronal activity was referenced to the signal

recorded from a steel guide tube in the same recording well. Neural signals were acquired using the Plexon MAP system (Plexon, Inc.). The continuous signals were passed through a preamplifier and then separated into two signal paths. The LFP channel was band-pass filtered between 0.7 to 300 Hz and digitized at 1 kHz. The spike channel was band-pass filtered between 100 Hz and 8 kHz and digitized at 25 kHz. Single units were isolated online via manually-set waveform triggers. During each recording session, one electrode was placed in PRR in the right hemisphere and one in the left hemisphere. While searching for cells, animals performed saccade and contralateral arm reach trials (with respect to the side of the isolated cell) as described above. Online, the preferred direction for a cell was defined as the target location that resulted in the largest sustained firing during the delay period. The null direction was defined as the target location 180° from the preferred direction across the fixation point. Data were then collected for all trial types (Fig. 1b). We recorded LFP from 312 sites (133 from MkT and 179 from MkZ) and single units from 113 of those sites (43 from MkT and 70 from MkZ). In each case, we recorded LFP from both hemispheres simultaneously, along with a single unit from either one or both hemispheres. We obtained data either for targets in all 8 directions (29 neurons, 7, in MkT, and 22 in MkZ) or for the preferred and null directions only (84 neurons, 36 in MkT, and 48 in MkZ). We obtained an average of 15 repetitions for each of the 10 trial types for each of the 113 neurons.

*Definition of PRR.* PRR does not fit neatly into any single anatomical area, but instead lies at the boundary of MIP and PO/V6A, though it also extends slightly towards the lateral bank, towards LOP[2,10]. We therefore functionally identify PRR as a region containing many neurons with a transient response to the presentation of a visual stimulus, especially one instructing a reach; sustained activity during a memory or delayed reach task that is greater than the sustained activity during memory or delayed saccade task; and another (often small) transient at the time of reach onset. (Not all neurons within the region show all of these properties).

*Recording during the preparatory period.* We focused on the preparatory period for three reasons. First, there is considerable evidence that PRR is primarily involved in movement planning rather than movement execution. Second, the preparatory period in an instructed delay paradigm provides a long stable period in which to measure responses. Within ~400 ms surrounding movement onset, multiple events occur in rapid succession, including the go cue, a saccade to the target, the change in visual input associated with that saccade, and the initiation and completion of the reach. Each of these events could modulate neural responses[75]. Spikes have high enough temporal resolution to follow changes occurring in 10 ms or less. In contrast, beta frequency LFP is an inherently slow signal, with accurate estimates of 20 Hz power requiring at least 100 ms and accurate estimates of coherence requiring 500 ms[76]. Thus, a stable preparatory period of 1 s or more is better suited to analyzing the role of LFP signals than during the movement period. Third, once movement begins, the parietal cortex receives proprioceptive feedback about postural changes of the limb[77–80]. This feedback has a short latency and can be recorded even before frank movement begins[81–83], making it impossible to distinguish effects that drive a movement from effects that are driven by that movement.

### Quantification and statistical analysis.

All data analyses were carried out using custom code written in C[84], R (version 4.0.3, https://www.R-project.org/[85]), and Matlab 2018a (Mathworks). Matlab code included the Chronux toolbox (http://chronux.org/[86]).

*LFP methods.* For analysis, an interval of interest was identified. The continuous LFP signal was windowed over the interval of interest for each trial. Here, we call $x_n(t)$ the windowed LFP signal for trial $n$ at time $t$.

*LFP power:* LFP power spectral density was estimated with a multitaper method. In brief, for each trial, the LFP signal was windowed with each of a number of orthogonal Slepian tapers, and Fourier transforms were estimated. The Fourier transform of the LFP signal $x_n(t)$ with the $k$th taper, $d_k(t)$ was estimated according to Eq. (1)

$$X_{n,k}(f) = \sum_{t=1}^{T} d_k(t) x_n(t) e^{-2\pi j f t}, \tag{1}$$

where $T$ is the length of $x_n(t)$, $f$ is the frequency, and $j$ is the imaginary unit (i.e., $\sqrt{-1}$). The power spectral density for a single trial $n$, $S_{xx,n}(f)$, was then estimated as a weighted average of auto-spectra across tapers according to Eq. (2)

$$S_{xx,n}(f) = \frac{1}{f_s K} \sum_{k=1}^{K} w_k |X_{n,k}(f)|^2, \tag{2}$$

where $f_s$ is the sampling frequency, $K$ is the number of tapers, and $w_k$ are weights determined by an adaptive algorithm[87]. The power spectral density was then averaged across trials to produce a single estimate of the power spectral density according to Eq. (3)

$$S_{xx}(f) = \frac{1}{N} \sum_{n=1}^{N} S_{xx,n}(f), \tag{3}$$

where $N$ is the number of trials.

We used a time-half-bandwidth product of 2.5, affording us 4 Slepian tapers. We used either 400 ms or 200 ms windows, affording us frequency resolutions of ±6.25 or ±12.5 Hz, respectively. Band-limited power was estimated by summing the power spectral density estimate over the band of interest. Power time signals were estimated by stepping the time window by either 100 ms (400 ms windows) or 50 ms (200 ms windows) and estimating band-limited power centered at each time step. In most cases, we present power time signals and power spectral density as a percentage of baseline power or power spectral density, respectively. Baseline values are estimated as the average value over 500 ms before the target presentation. Power was computed at each LFP recording site individually before averaging across the population.

The bands of interest, 20–30 Hz and 70–120 Hz, were not selected a priori. Instead, these bands were selected empirically early in our study to capture general trends in the power density spectra and then maintained as we collected more data. Note that with a frequency resolution of ±6.25 Hz (400 ms time windows), the band labeled 20–30 Hz actually included information from frequencies from 13.75 to 36.25 Hz. The same is true for measures of synchronization described below. The entire analysis was repeated using the Chronux toolbox (http://chronux.org/[86]). Although estimates were performed at slightly different frequencies, the results were essentially identical.

*LFP–LFP synchronization*: The degree of synchronization between two LFP signals was quantified with coherence. Like variance, coherence is a measurement across trials. Coherence between two LFP signals, $x$ and $y$, was estimated according to Eq. (4)

$$C_{xy}(f) = \frac{S_{xy}(f)}{\sqrt{S_{xx}(f)S_{yy}(f)}},\qquad(4)$$

where $S_{xx}(f)$ and $S_{yy}(f)$ are the mean power density spectra across trials for LFP signals $x$ and $y$, respectively, and $S_{xy}(f)$ is the mean cross-spectrum across trials for LFP signals $x$ and $y$. Power spectral densities were estimated with the same multitaper method described above. Cross spectra were computed in a manner similar to power density spectra. The cross-spectrum for a single trial $n$, $S_{xy,n}(f)$ was then estimated as a weighted average of the cross spectra across tapers according to Eq. (5)

$$S_{xy,n}(f) = \frac{1}{f_s K}\sum_1^K w_k X_{n,k}(f) Y^*_{n,k}(f)\qquad(5)$$

where $X_{n,k}(f)$ and $Y_{n,k}(f)$ are the Fourier transforms of time series $x(t)$ and $y(t)$, respectively, and $Y^*_{n,k}(f)$ is the complex conjugate of $Y_{n,k}(f)$. The mean cross-spectrum across trials is then estimated according to Eq. (6)

$$S_{xy}(f) = \frac{1}{N}\sum_{n=1}^N S_{xy,n}(f),\qquad(6)$$

where $N$ is the number of trials.

*Spike-LFP synchronization*: Synchronization between spikes and LFPs was quantified with three different measures: spike-LFP coherence, phase-locking value, and pairwise phase consistency. All three measures were computed over an 800 ms time window just before the go cue. Each measure was computed separately for each movement type. To protect against differences in mean spike rate driving spike-LFP coherence, all analyses were restricted to neurons with at least 500 spikes in each movement condition. Restricting the analysis to exactly 500 (randomly sampled) spikes from each cell for each condition resulted in similar results.

Spike-LFP coherence was estimated using the Chronux toolbox (http://chronux.org/[86]). The methods of the toolbox are briefly summarized here. For each trial, Fourier transforms were computed separately for spike and LFP signals using a multitaper method. For each taper, the LFP and spike signals were windowed by the taper and a Fourier transform was computed using a fast Fourier transform (FFT) algorithm (http://fftw.org[88]). Before computing the FFT of the spikes, the mean spike rate of the trial was subtracted away to remove the DC component. The Fourier transforms were then used to compute coherence values as with LFP–LFP coherence above. We elected to use a time-half-bandwidth product of 12, which afforded the use of 23 Slepian tapers and yielded a frequency resolution of ±15 Hz. Analyses using narrower frequency resolutions revealed the same effects but included additional narrow-band noise. This narrow-band noise likely reflects the fact that we interleaved 10–80 trial types (5 task types and 2 to 8 directions) and therefore had limited numbers of trials per trial type.

Phase-locking values (PLV) were computed for a range of frequencies. For a given frequency, the LFP phase at the time of each spike was estimated with a wavelet transform. Phases were then pooled across trials. PLV was estimated according to Eq. (7)

$$PLV = \frac{1}{S}\sum_{s=0}^S \exp(j\theta_s)\qquad(7)$$

where $\theta_s$ is the phase at the time of spike $s$ and $S$ is the number of spikes. Significance was assessed with a Rayleigh test.

Pairwise-phase consistency (PPC) was assessed in a similar manner to PLV. Phases at the time of spikes were obtained using a wavelet transform. However,

phases were not pooled across trials. PPC was estimated according to Eq. (8)

$$PPC = \frac{1}{N(N-1)}\left(\left|\sum_{n=1}^N\left(\frac{1}{S_n}\sum_{s=0}^{S_n}\exp(j\theta_{n,s})\right)\right|^2 - \sum_{n=1}^N\left|\frac{1}{S_n}\sum_{s=0}^{S_n}\exp(j\theta_{n,s})\right|^2\right)\qquad(8)$$

where $N$ is the total number of trials, and $S_n$ is the number of spikes in trial $n$, and $\theta_{n,s}$ is the phase at the time of spike $s$ in trial $n$.

Many neurons have well-defined preferred directions for reach and saccade trials. To test whether LFP also shows preferred directions, one can find the movement direction at each site that produces the strongest modulation, average those strongest responses together, and contrast them with the response obtained for movements in the opposite direction. While such a procedure will capture a tuning if it exists, it is also highly likely to produce a statistically significant differential effect even in the absence of tuning. Appropriate analyses can exclude such artifacts, but we instead ask a simpler question. We restrict the data to sites at which a well-tuned single unit was recorded on the same electrode (113 sites, 43 from MkT and 70 from MkZ) and ask what information is coded by LFP power when each cell's preferred direction is considered.

*Statistics.* No statistical methods were used to pre-determine the sample size, but the numbers of monkeys used for these experiments are comparable to those used in the field and in previous studies. Data collection and analysis were not performed in a manner blind to the conditions of the experiments. Both animals performed all tasks and were not randomly assigned to a specific experiment group. All testing occurred during the light cycle. All trial types were randomly interleaved for each cell or site in a recording session.

Statistical analyses used trial-averaged data. Unless specified otherwise, data were pooled across sessions and monkeys. We used pooled *t*-tests unless specified. All tests were two-sided. The criterion for all tests was alpha = 0.05. Values are reported as the mean ± standard error of the mean (s.e.m.). We use a conservative approach (Bonferroni) for multiple comparison corrections. When reporting significant differences, we include the correction, but when reporting no difference, we do not.

Significant spike-LFP coherence at each frequency was tested against the null hypothesis that there was no spike-field coherence using a permutation test. We generated the null distribution for no spike-field coherence by permuting the interspike intervals on each trial (188 permutations). The permuted significance thresholds for each movement condition are similar so the mean thresholds are displayed. Spike-field coherence was tested during the delay period in the 800 ms immediately before the go cue.

**Reporting summary**. Further information on research design is available in the Nature Research Reporting Summary linked to this article.

## Data availability

All the relevant data are that support the findings of this study are available at http://eye-hand.wustl.edu/supplemental/Mooshagian2021. A reporting summary for this Article is available as a Supplementary Information file. Source data are provided with this paper.

## Code availability

The code used for the analyses that support the findings of this study are available from http://eye-hand.wustl.edu/supplemental/Mooshagian2021.

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

## Acknowledgements

This work was supported by the National Eye Institute Grant EY-012135 (LHS), and the National Institute of Neurological Disorders and Stroke grant NS-076206 (EM). We thank Dr. Ilya E. Monosov, Dr. Pablo M. Blazquez, and Dr. David M. Kaplan for useful discussion and comments on an early draft of the manuscript. We also acknowledge the considerable technical assistance and wisdom of Chuck Crist.

## Author contributions

E.M. and L.H.S. designed the experiment. E.M. performed the experiment. E.M, C.D.H., and L.H.S analyzed the data. E.M. and L.H.S. wrote the manuscript.

## Competing interests

The authors declare no competing interests.
