## [Peer Review File · Nature Communications]

Reviewers' Comments:

Reviewer #1:

Remarks to the Author:

This is very interesting and important study from the lab of one of best experts in the field. I noticed the rebuttal letter addressing the previous submission, and I agree with most points made by the authors in their reply, especially on the suitable novelty of the research question and the approach. It is however quite complex work (although it might appear fairly straightforward on the surface), and several methodological and interpretational aspects need clarification or justification, to be more convincing. Partially this might be due to the fact that the methodology has only recently been adopted by this group. The presentation is generally quite clear but also fairly rough at times and the manuscript needs a bit more polishing.

The major claim of the study is that that local field potential (LFP) coherence is increased in the beta range during motor preparation of bimanual reaches to the same spatial target. This increase is interpreted as a sign of enhanced interhemispheric communication supporting bimanual coordination. This is a novel (perhaps not too surprising) and important finding that hasn't been demonstrated for the parietal cortex, and as such, is of general enough interest to the community. If put into a more global context, it is also of interest for the wider audience (but the authors are advised to work on a more encompassing framework for their findings, please see comments on the Discussion below). I am convinced that this work will stimulate further studies on the neuronal basis of interhemispheric communication. I think this work merits the publication in the Nature Communications, after a revision.

Major comments

1)

One obvious question is why the saccades were allowed (or forced in unimanual conditions) during reaching. I believe that most work in this lab (and many other labs) used reach-only or saccade-only conditions, unless the goal was to study explicitly the eye-hand coordination. I fully understand that it is more difficult for the monkeys to perform unimanual and especially bimanual reaches without accompanying saccades, and that saccade-reach is a more natural behavior. But many PRR cells are responsive to visual stimuli denoting upcoming saccade targets, and are active during the delay period prior to saccades (even if less so than for reaches). Therefore it is not entirely clear how the saccade preparation, which might take place in both hemispheres -- in part because PRR is typically not very contralaterally tuned for space -- might affect the results. It would be beneficial for the narrative if the authors explain why they believe this does not represent a confound, or suggest ways to isolate potentially confounding effects of saccade preparation in their analysis. They should also explain how very likely divergent patterns of saccadic behavior -- e.g. saccading first to one target, then to another in the bimanual-apart reaches, as compared to reaches to the same target -- might have influenced the findings (e.g. the decreased LFP-LFP coherence for the bimanual-apart condition).

2)

The results on spatial tuning of beta power are somewhat puzzling. Beta power in the parietal cortex has been reported to be tuned not only for the task/effector, but also spatially, with inRF more suppressed than outRF, relative to the fixation baseline (e.g. Dean et al., 2012). One notable exception is the study of Scherberger and colleagues (Scherberger et al., 2005), but these results are perplexing (as the authors of the present study note, Scherberger et al. also reported higher beta for reaches than for saccades). The present study shows that the delay beta power is highest for saccades and lowest for contramanual and bimanual movements. This makes total sense, but the reversed effect in inRF (higher beta) vs. outRF (lower beta) does not. To re-phrase it in a simpler way, the first finding is that the "most relevant" (or preferred) for the PRR condition (i.e. contramanual reaches) show less beta. But the second finding is that the "most relevant" (i.e. inRF) direction shows more beta. It might be that the apparent discrepancy is due to the sorting of in/out (preferred direction) in accordance to

the spatial tuning of single unit spiking. Since PRR neurons could have both contralateral and ipsilateral tuning, and the tuning of a single neuron might not be representative of the overall population reflected in the LFP, my suggestion would be to sort the data in Figure S4 to contralateral/ipsilateral space, rather than to preferred/anti-preferred direction in respect to the associated unit.

3)

Lines 200-206: it is surprising that the authors talk about pre-movement increase in beta power (for the contramanual reaches). The common finding is that beta is disrupted by the transient events such as visual cue and saccade/reach in frontoparietal areas (e.g. Dean et al., 2012; Sendhilnathan et al., 2017). Please show population power time-frequency plots for all conditions, aligned to cue and movement onset.

4)

Line 236: "Finally, movement plans affect beta power only ~100 ms after they affect gamma power and single unit activity." – I would be cautious with interpreting temporal differences in this range given the considerable temporal smoothing. Please comment on this.

5)

Lines 260-264: "A second regime occurs at lower frequencies, from about 16 to 32 Hz. Here LFP power depends strongly on both movement type and frequency, but only weakly on direction. Responses are not clustered but instead are ordered by movement type. Peak power occurs at a low frequency for saccades (~24 Hz), at an intermediate frequency for ipsimanual reaches (~30 Hz), and at still higher frequencies for contramanual, bimanual-together and bimanual-apart reaches." – The reasoning behind these statements is not clear. Why the authors refer here to the positive peak of the power if their consistent message is that it is the decrease of beta power (and hence, the negative peak) relative to the baseline that is relevant for contramanual reach conditions?

6)

Lines 303-306: "Spike-LFP coherence for bimanual-together movements was significantly higher than for bimanual-apart movements over most of the beta range (two-tailed t test; asterisks in Fig. 6a). This difference could not be attributed solely to strongly oscillating LFP signals during bimanual-together compared to bimanual-apart trials, since the elevation also occurs in coherence measures that are unaffected by LFP magnitude (spike-LFP synchrony)." – I admit to not following the reasoning here. What's "spike-field synchrony" – isn't this what is being estimated by the spike-LFP coherence (but it sounds like those are considered two different things, from the excerpt above)? I am not entirely certain at the moment if there is a full consensus on whether the power of a noisy LFP signal in a certain band is affecting spike-field coherence level in this band – I understand that coherence is normalized to the LFP power, but if there is very little power in a certain frequency then the noise will probably dominate. But let's assume that the authors meant here that the coherence is not affected by the LFP power.

What I am a bit more concerned about is the usage of the coherence, rather than less biased pairwise-phase consistency (PPC) measure for assessing the spike-LFP coupling (Vinck et al., 2012). The authors however addressed this issue by fixing the amount of spikes to 500, and the coherence has been extensively used before. Nevertheless, given that PPC = gains more and more popularity, it would be very interesting to see if the coherence findings also hold for the PPC (but it is a lot of work so I leave it at the discretion of the authors whether to add this analysis).

7)

What is however majorly missing from the analysis is the further separation of the data into two conditions – spikes one hemisphere – LFP another hemisphere, and vice versa. For unimanual reaches, this separation can be done in respect to the hemisphere contralateral to the acting hand. For bimanual-together reaches, in respect to the hemisphere contralateral to the target (for bimanual-apart reaches, the meaningful separation is not possible, unless both "crossed" and "uncrossed"

conditions were present, in which case I would separate to crossed vs uncrossed conditions – see the question on Methods below in Minor comments). The important question here is whether there is a “directionality” (or asymmetry) of the interactions depending on specific task conditions – for instance, is there more coherence when spikes from the hemisphere that is contralateral to the target are considered, as compared to the spikes from the ipsilateral hemisphere?

8)

Figure 6B: Can this finding be also expressed in terms of spike-LFP phase relationship analysis (cf. Hawellek et al., 2016), rather than, or in addition to the lag?

9)

The methodology of the local field potential would benefit from some clarifications:

How LFPs were recorded/preprocessed? There is not enough information to understand some details of the methodology. First, were LFPs in the two hemispheres recorded using common ground/reference? Were the means of the two LFPs subtracted from each channel, to perform a sort of re-referencing, or the analysis was performed on raw LFPs?

Second, the relationship between time windows, sliding window steps, and frequency resolution is not clear. In particular, the reported frequency resolution is very low for the LFP analysis (± 6.25 , ± 12.5 and ± 5 Hz, derived from the relationship between window length and the time-half-bandwidth: time-half-bandwidth $TW = NF/2$). But it seems that the plots in Figure 2 and 3 show much higher frequency resolution? For the spike-field coherence, $TW = 0.8 \cdot 15 = 12$, but how this results in only 4 Slepian tapers, given number of tapers $L = 2TW - 1$? I am sure I am missing something. Incidentally, has Chronux also been used for the LFP power and coherence, or custom code, or native MATLAB functions (e.g. `pmtm.m`), as opposed to the Chronux usage for the spike-LFP coherence?

What were the reasons for using the multitaper approach for the low frequencies? It has been suggested that while multitaper works well for gamma frequencies, it introduces too much smoothing into the lower frequency (beta) range: (cf. <http://www.fieldtriptoolbox.org/tutorial/timefrequencyanalysis/>)

Third, the patterns for gamma power and spiking are largely very similar (besides the bimanual apart condition). It might be expected, but given that the separation into the two streams -- spiking and LFP -- was done using the Plexon online filtering (and not via recorded broadband data that provides more flexibility for offline analyses, e.g. noncausal median filtering and removal of spiking waveforms in time domain, and/or and high frequency power from LFPs), there are potential concerns that spikes might have leaked into the high frequency LFP. Since the gamma frequency is not the main point of the study, this is not a critical issue, but perhaps the authors could comment on this.

10)

Discussion is very mechanistic-oriented and lacks the big picture for a wider audience, and some important points for experts in the field. For example, I expected the discussion of the interhemispheric competition / rivalry / push-pull models, and the construal of the increased coherence for bimanual-together condition as facilitatory exchange of information between hemispheres, as compared to potentially inhibitory effects during the bimanual-apart condition with decreased coherence (as one potential, perhaps overly simplistic, interpretation). At the very least, the authors should attempt to relate their findings to a lively debate on the compensatory/beneficial vs. maladaptive recruitment of the intact hemisphere in visuospatial and motor functions during unilateral brain damage, in neurological literature (Murase et al., 2004; Umarova et al., 2011; Bartolomeo and Thiebaut de Schotten, 2016), in noninvasive brain stimulation studies (Johansen-Berg, 2007; O’Shea et al., 2007; Sparing et al., 2009; Koch et al., 2011), and in reversible parietal lesion experiments in monkeys (Wilke et al., 2012).

On a more detailed circuit-oriented level, however, the transcallosal connectivity is mainly mediated

via axonal projections to inhibitory interneurons (Bloom and Hynd, 2005; Palmer et al., 2012). How the known interhemispheric circuitry constrains the schematic model of information transfer the authors have in mind?

Another aspect that needs to be considered is the translational value of these experiments to humans, and its limitations. For example, effects of training are important. Previous work demonstrated differences in motor performance in over-trained monkeys as compared to naïve humans: while in humans bimanual movements took longer than unimanual ones, movement times of bimanual movements in monkeys were shorter than unimanual ones (Gribova et al., 2002).

Minor comments

Introduction

“Temporary inactivation of PRR selectively interferes with contralateral arm movements, and at a population level, unit activity codes only contralateral arm movements.” – I am not sure how well this statement agrees with previous work from the same lab that shows a substantial number of ipsilateral and bilateral arm neurons (Chang et al., 2008).

“We found that local field potentials (LFP) are coherent across hemispheres, with the extent of the coherence depending on the type of reach being planned. This is consistent with information about reach plans being exchanged across the hemispheres. In fact, beta-band LFP carries task-specific plans for ipsilateral arm movements.” – The last sentence does not appear the logical continuation of the previous two – how ipsilateral arm encoding is mapped onto interhemispheric exchange?

Results

“Reaches were made with just the arm ipsilateral to the recording site (ipsimanual), with just the arm contralateral to the recording site (contramanual), or with both arms (bimanual).” – This terminology does not make much sense for simultaneous bihemispheric recordings. I guess the idea is to specify conditions for LFP power and spiking activity from each hemisphere, e.g. in the Figure 3. Please consider rephrasing.

Figure 2 legend – what are the “30 frequency bands”? What is the frequency resolution in this plot? It seems that it’s 2Hz, but please see the question about LFP methodology above.

“To isolate the effects of movement type independent of frequency, we normalized by dividing by baseline power at each frequency and then plotted the result as a function of time.” – It seems that %signal change from the baseline is plotted, not a purely divisive normalization. Please clarify.

Supplementary Figure 5 – there are no “Out of RF” traces here.

Discussion

Lines 450-453: “With this in mind, delay period power was greatest for saccades, followed in order by ipsilateral arm movements, contralateral arm movements, movements of both arms together to the same target, and movements of both arms to two different targets. The peak frequency of these effects varied in the same order, from ~21 Hz for saccades up to ~35 Hz for bimanual-apart movements (Fig. 4).” – same problem as in Results, the positive peaks for saccade and ipsimanual are equated with rebound from negative peaks in the other conditions (and in bimanual-apart, there is really no peak in this range – the power just continues to raise with the frequency). I believe that this is misleading description and it does not lead anywhere.

Lines 543-545: “The substantial baseline level of beta-band LFP-LFP coherence suggests that

information is shared between left and right PRR even when the arms do not move, e.g., in a saccade task or during an intertrial interval." – Or it is there just because there is shared noise, volume conduction, etc. What kind of information the authors are referring to – postural control, awareness of limbs, small spontaneous movements? In general, I would refrain from equating the apparent LFP-LFP coherence (functionally significant or not) with the "information".

Methods

It is indeed commendable that the authors managed to train the monkeys to do bimanual-apart reaches. Was it always that the left hand went to the left target, and the right to the right, or they were also doing even more demanding "crossed" reaches (left hand to right space and vice versa) or only "uncrossed" ones?

At any rate, the bimanual-apart reaches are important because this condition can be used to argue that the main finding cannot be explained by the task difficulty/effort. The saccade is the easiest condition, the bimanual-apart is the most difficult, the bimanual-together is arguably easier than apart, but still, the highest LFP-LFP coherence corresponds to the bimanual-together condition. If the authors agree with this reasoning, they might want to mention in the text.

"(<http://chronux.org/>; {Mitra;qIMLon5c})." –proper reference?

"Mean values were subtracted from the spike signals before the FFT to remove DC components." – This is not clear. What are mean values of the "spike signals"? Mean firing rate across all trials? Please explain.

References

"Percival, D. B., Multitaper, A. W. S. A. F. P. A.1993. Spectral analysis for physical applications: Multitaper and conventional univariate techniques. Cambridge Univ. Press, New York." -- the second author is not multitaper, it's Walden :)

Percival, D. B., and A. T. Walden. Spectral Analysis for Physical Applications. Cambridge, UK: Cambridge University Press, 1993.

Literature cited in the review

Bartolomeo P, Thiebaut de Schotten M (2016) Let thy left brain know what thy right brain doeth: Inter-hemispheric compensation of functional deficits after brain damage. *Neuropsychologia* Available at: <http://linkinghub.elsevier.com/retrieve/pii/S0028393216302159> [Accessed July 4, 2016].

Bloom JS, Hynd GW (2005) The role of the corpus callosum in interhemispheric transfer of information: Excitation or inhibition? *Neuropsychology Review* 15:59–71.

Chang SW, Dickinson AR, Snyder LH (2008) Limb-specific representation for reaching in the posterior parietal cortex. *J Neurosci* 28:6128–6140.

Dean HL, Hagan MA, Pesaran B (2012) Only Coherent Spiking in Posterior Parietal Cortex Coordinates Looking and Reaching. *Neuron* 73:829–841.

Gribova A, Donchin O, Bergman H, Vaadia E, de Oliveira SC (2002) Timing of bimanual movements in

human and non-human primates in relation to neuronal activity in primary motor cortex and supplementary motor area. *Exp Brain Res* 146:322–335.

Hawellek DJ, Wong YT, Pesaran B (2016) Temporal coding of reward-guided choice in the posterior parietal cortex. *Proceedings of the National Academy of Sciences* 113:13492–13497.

Johansen-Berg H (2007) Functional imaging of stroke recovery: what have we learnt and where do we go from here? *Int J Stroke* 2:7–16.

Koch G, Cercignani M, Bonni S, Giacobbe V, Bucchi G, Versace V, Caltagirone C, Bozzali M (2011) Asymmetry of Parietal Interhemispheric Connections in Humans. *J Neurosci* 31:8967–8975.

Murase N, Duque J, Mazzocchio R, Cohen LG (2004) Influence of interhemispheric interactions on motor function in chronic stroke. *Annals of neurology* 55:400–409.

O'Shea J, Johansen-Berg H, Trief D, Gobel S, Rushworth MF (2007) Functionally specific reorganization in human premotor cortex. *Neuron* 54:479–490.

Palmer LM, Schulz JM, Murphy SC, Ledergerber D, Murayama M, Larkum ME (2012) The Cellular Basis of GABAB-Mediated Interhemispheric Inhibition. *Science* 335:989–993.

Scherberger H, Jarvis MR, Andersen RA (2005) Cortical Local Field Potential Encodes Movement Intentions in the Posterior Parietal Cortex. *Neuron* 46:347–354.

Sendhilnathan N, Basu D, Murthy A (2017) Simultaneous analysis of the LFP and spiking activity reveals essential components of a visuomotor transformation in the frontal eye field. *PNAS* 114:6370–6375.

Sparing R, Thimm M, Hesse MD, Kust J, Karbe H, Fink GR (2009) Bidirectional alterations of interhemispheric parietal balance by non-invasive cortical stimulation. *Brain* 132:3011–3020.

Umarova RM, Saur D, Kaller CP, Vry M-S, Glauche V, Mader I, Hennig J, Weiller C (2011) Acute visual neglect and extinction: distinct functional state of the visuospatial attention system. *Brain* 134:3310–3325.

Vinck M, Battaglia FP, Womelsdorf T, Pennartz C (2012) Improved measures of phase-coupling between spikes and the Local Field Potential. *J Comput Neurosci* 33:53–75.

Wilke M, Kagan I, Andersen RA (2012) Functional imaging reveals rapid reorganization of cortical activity after parietal inactivation in monkeys. *PNAS* 109:8274–8279.

Reviewer #2:

Remarks to the Author:

This study reports that beta-band LFP power in PRR changes systematically with different patterns of uni-manual and bi-manual reaching tasks. In contrast, neural spiking activity in PRR reflects only what the contralateral arm is doing (or not doing) in different task conditions. Gamma-band LFP power likewise reflects the actions of the contralateral arm. The findings imply that PRR is implicated in bimanual coordination and the underlying neural processes cause alterations in LFP power in the beta band that may result from task-dependent synaptic inputs from PRR in the hemisphere contralateral to the recording sites.

The main findings of this study seem to be solid and are a significant contribution to the field.

However, there are a number of points that need clarification.

1) One of my great frustrations over the years is how exactly PRR is defined and to what degree it does or does not correspond to cortical regions identified by other criteria and nomenclatures. In fact, the source of my frustration is that it has never been clearly defined in any study that I have read about "PRR". It just seems to be wherever they record arm-related activity across a large swath of cortex rostral and medial to the intraparietal sulcus. For instance, Fig 1C shows that many neural data were collected in MIP and in PO. Are both of these cortical regions lumped into "PRR" and are all of those data pooled into the "PRR" data set in this study? If the lump them together, that implies that there are no functionally meaningful differences in activity of reach-related neurons in those two cortical areas, at least with respect to this study, but other studies have treated them as functionally different. Please try to be a bit more specific about what "PRR" means in this study.

2) There are a number of technical details about the configuration of reach targets and the nature of the unimanual and bimanual reaches that confuse me, because the description of the task is too cryptic. A more detailed figure of the configuration of targets on the monitor would be very helpful:

2A) Ln 831-832: "Touch positions on the screen were organized in a virtual 3 X 3 grid centered on the fixation point." Does that mean that there were two 3 X 3 grids of potential target locations on either side of the fixation point and so on either side of the body, or was there only one 3 X 3 grid centered on the fixation point and body midline, or something else? If the latter, does that mean that the monkeys sometimes reached to targets aligned to their body midline?

2B) Ln 833-834: "Plexiglass dividers were mounted on the front of the screen at the middle of each target location." The first 5 times I read that paragraph, they sounded like barriers that would prevent the monkeys from actually touching each target. I think I finally worked it out but a figure would likely make this much easier to understand.

2C) Ln 845-847: "When two targets appeared...at opposed diagonal locations." Since I'm still not sure if there were two 3X3 target grids on either side of the midline or only one 3X3 grid centered on the midline, I don't understand what this means, especially "opposed diagonal locations". I'm going to assume that this means that the two targets were not chosen randomly, but were always yoked by some simple rule. I'm going to assume further that there were 2 grids and that the two targets were always mirror-symmetrically located in the two grids. I hope I was right. Please clarify. Again, a figure showing how target locations might yoked would be useful.

2D) IF there were TWO grids, did the monkeys only reach to the grid on the same side of the body during the unimanual reaches or did they also reach across their body to the grid on the other side of the body? If they did not do the latter, then the only time they made reaches across their body would have been during bimanual-together trials. Is that correct, or not? If it is correct, then they have no unimanual control for the coherence or power associated with reaches of the arm across the body midline into the contralateral hemi-field during bimanual-together movements.

3) Ln 120-140, Fig 2: The prediction that LFP-LFP coherence should increase for the bimanual-together condition compared to the unimanual condition seems reasonable, but why should the coherence decrease for the bimanual-apart condition when the movements are aimed at different targets rather than converging on the same one? They are still temporally coordinated reaches (but see the next comment), and if anything seem to be more demanding since the monkeys must split their attention between the two target locations to ensure accurate reaching for both arms. Why do spatially convergent reaches necessarily mean higher spatial coordination than spatially divergent reaches? They are both spatially-directed actions (c.f., "where versus when", In 129) that require similar degrees of precision, just not to the same target. At one level, the latter is more spatially coordinated because the biomechanics of the two reaching movements are probably nearly mirror-symmetrically similar, unlike the reaches that converge on the same target. Why should this require

less coherence than the unimanual reaches when the other arm is not moving at all? I think that the point that they are trying to make is that when the two arm movements are spatially dissociated, it is useful to functionally decouple the activity in the two hemispheres even beyond that observed during unimanual tasks and one way to do that is to reduce the coherence level further. This would provide a stronger argument than making some vague statements about spatial coordination. It sounds too ad hoc and post hoc.

4) Ln 859-860: "On bimanual trials, the left and right hands were required to hit their target(s) within 500ms of one another." 500ms is nearly forever for monkeys in this kind of task. That could give them enough time to make the two reaches completely sequentially, not simultaneously at all. That statement points to critical data that are missing from this report. This is a study of bimanual coordination and yet there are no data about the degree of coordination of the two arm movements. Did they start simultaneously or did one arm tend to lead the other? Likewise, did the two hands touch the targets simultaneously or in sequence? Did this differ for bimanual-together and bimanual-apart actions? I know that this study focusses on preparatory activity, but it would be important to see these execution-related data to give some insight into how the monkeys viewed these task conditions, e.g., coordinated bimanual actions or sequential unimanual actions?

5) With that last two comments in mind, maybe it might be better to use "bimanual-convergent" and bimanual-divergent" rather than "bimanual-together" and bimanual-apart". I find the former more explicit (and maybe even a little bit more esthetically pleasing!) than the latter. Just a suggestion.

6) A major finding is that the LFP coherence reflects the different uni/bimanual reach conditions, but the spiking activity only reflects the action of the contralateral arm. Is there a disconnect in those findings? If the coherence reflects synaptic inputs from the contralateral hemisphere and elsewhere and plays an important role in bimanual coordination, and if spiking activity represents the outputs that generate the movements, how do changes in bilateral LFP coherence influence bimanual control without having a measurable effect on spiking activity? I'm not suggesting that this disproves their hypothesis, but I would like to know how that can accommodate those two findings.

7) This article focusses on PRR, and implicates PRR in bimanual coordination. But is it not possible that these same findings could also generalize to other cortical motor areas that receive callosal projections from their homologous regions in the other hemisphere? The article almost implies that this function is unique to PRR.

8) Ln 877-878: "...an electrode was placed in PRR in one or both hemispheres." How many sessions involved simultaneous recordings in both hemispheres and how many were only unilateral? Please state such information clearly in one place, such as in the Methods. I can see some numbers buried in figure legends, but it is very frustrating to have to look around for such basic information at random locations throughout the article.

9) Ln 369-388, 1115-1178, Fig 7: This material purports to show the results of an "algebraic model" that they present to support their proposal that the beta-band LFP modulations across task conditions could be explained by a simple weighted linear sum of mean spike rates in the two hemispheres across task conditions. They show some spike-rate numbers in the Supplementary Table 1 and the results of the "model" in Fig 7, but they don't show the "model". My guess is that it is nothing more than a simple two-term algebraic equation like:

$$\text{LFP change} = b_1 \cdot \text{Contra spike rate change} + b_2 \cdot \text{Ipsi spike rate change},$$

where b_1 and b_2 are the relative weights for the two spike rates (e.g., 100:0 or 80:20 from the table). Is that all it is? Please provide the equation. This seems to assume a simple linear relationship between spike rates and LFP power, despite all the disclaimers they listed in Ln 1127-1138, including that "there is no reason to assume that the degree of suppression will be linear with the rate of

spiking input." This whole exercise is too simplistic to be very convincing.

Reviewer #3:

Remarks to the Author:

The paper by Mooshagian and colleagues investigates PRR's role in planning unimanual and bimanual reaches. I think the question the authors address is interesting and quite understudied. Their task design is appropriate: it allows them to separate the temporal (during the instructed delay) and spatial aspects of planning bimanual reaches (move the arms to the same target or to two different targets), with some additional unimanual reaching conditions. Some results are quite intriguing, including the differences in LFP modulation across bands and how they relate to aspects of the upcoming behavior, as well as the spike-field coherence analysis.

The paper is in general well-written except for a few sentences that I mention in my comments; the Discussion could be shortened (it's 40 % of the paper). My only concern is the lack of raw data, and the need to include more intermediary results and quantify some effects (see below). The authors should also give more precise details about their analyses. I'd also like to see some new analyses and controls:

Comments:

1. The authors have long-standing experience with these experiments but I'd like them to show raw data and "raw results," not only grand averages. For example, 1) How consistent are movement kinematics? (important for trial-averaging); 2) Are the LFPs in PRR modulated by trial type and target location? (e.g., show PSTHs of the beta and gamma powers, similar to their Suppl Fig 4); 3) Show LFP-LFP coherence traces between a few pairs of example channels for the different task conditions (like Fig. 2); 4) Similar to #2, show PSTHs of a few example single neurons (across trial types and target); 5) Show the averaged results by monkey instead of pooling both monkeys together.

2. I couldn't find whether the authors used trial-averaged data or single trials for their analyses, and how they did the pooling across days, sessions and monkeys. This should be clarified, and the authors should provide evidence of consistent behavior (and more details about the monkeys' behavior)

3. I'm probably missing something, but why does beta power start decreasing *before* target presentation? (lines 189-190, Suppl Fig 3). Could the monkeys predict that a trial was about to start? --I couldn't find in the Methods if the inter-trial interval was variable or not.

4. Lines 225-226: I'm puzzled by the following statement: "Thus, like unit activity but unlike beta LFP power, gamma LFP power carries relatively little information about ipsilateral arm movement". Do the authors simply mean that gamma LFP power and firing rates aren't more modulated by arm movement than saccades are? Is that their "baseline" for this statement? I am not saying that I disagree about there being a difference between, but I do see some modulation. Since this result is the basis for the model, it should be clarified (and perhaps quantified). The same for Lines 272-273, Lines 505-506, and Lines 645-647.

5. Fig. 6: the chance coherence values may be this low because of how they were generated. The authors should take firing rates from trials from other task conditions and compute their coherence with LFPs from the condition being analyzed; this shuffling should give a more informative lower bound as it will preserve some key features of the neural data. I'd also like to see the same analysis performed "within hemisphere" (i.e., intra-hemisphere LFP-firing rate coherence). It'd be especially interesting to see whether the LFP-firing rate lag that gives maximum coherence is shorter than for the inter-hemispheric analysis.

6. Fig. 7 and “Algebraic model”: As it stands, this section is not very useful. I’d lean towards expanding the explanation (like they do at the beginning of the Discussion in Lines 407-408, and perhaps add a schematic) and leaving it in the paper, or alternatively leave it all in the Supplementary material. Personally, I’d be more convinced if the comparison between inter-hemispheric and intra-hemispheric LFP-firing rate coherence described above is consistent with the authors’ hypothesis than by this model, although I can see its merit.

7. In the Discussion the authors propose “that reach plans are first produced in PRR contralateral to the limb that will move. We propose further that plans for movement of the ipsilateral limb, which are prominently encoded by beta LFP power and appear only sporadically in spiking activity, likely originate from distal areas that include PRR in the opposite hemisphere” (Lines 494->). I think it’s a very interesting idea, and part of it could be tested with the current data. For example, the authors could test whether spikes or at least LFPs in the contralateral hemisphere lead those in the ipsilateral hemisphere, to complement their Fig. 6. I’m not a huge fan of asking for lots of additional analyses when reviewing a paper, but this one would be nice.

8. What happens at the higher gamma frequencies? Some groups have shown that there’s task-relevant information even at 200-400 Hz in M1 (Bansai et al J Neurophysiol 107, 2012). Is this also true for PPR?

9. Ames & Churchland (bioRxiv 2018) have recently studied M1 activity during ipsilateral and contralateral movements. Similar to previous studies (e.g., Cisek et al J Neurophysiol 89, 2003), many single motor cortical neurons were active during both ipsilateral and contralateral movements. But Ames & Churchland present a very interesting novel result: neurons covaried among them in a fully condition-dependent manner (contralateral vs. ipsilateral), meaning that they occupy orthogonal subspaces or “neural manifolds.”

I think that applying this method to the authors’ data would complement nicely their LFP-firing rate coherence analysis at the neural population level, making the paper stronger. For example, I’d expect neural populations to plan “bimanual-apart” and “bimanual-together” reaches in orthogonal subspaces.

Minor comments:

a. The authors should report how many trials for each condition and monkey they’ve averaged. Also, did they use concatenated single trials or trial-averaged data for the different analyses? In my opinion, this information should be upfront in the Results --I also couldn’t find it on the Methods. This also ties up to my request to see more raw data.

b. The authors repeatedly say “spikes” but they are analyzing “neural firing rates”; perhaps the results would be completely different if they assumed a “precise spike timing” perspective. I’m not suggesting the authors redo the whole paper, but rather that they use a more precise language.

c. I don’t know what I’m supposed to take out of Suppl Fig 2 on top of the known log-scale of the LFP power. Perhaps highlight some features of interest for your tasks?

d. Lines 166-168: What does the following statement mean? “This contrasts with single units, whose firing rates distinguish only target location and the pattern of contralateral arm movement at the population level, and even at the single unit level show only sporadic and non-systematic effects of bimanual movements.” Given the increasing emphasis on population level analyses (see Cunningham & Yu Nature Neurosci 2014) the authors should clarify what they mean by “population level”. Also, I’d be surprised if they couldn’t find specific neurons that had different responses across task conditions. Again, it would be nice to see more raw data.

e. Lines 196-197: "The fact that spikes lead LFP power leads to the surprising conclusion that modulations of firing rate by task are not driven by modulations in beta power." -> refer to where this is shown in the paper -I understand it's Fig. 3?- and quantify more rigorously.

f. Lines 208-210: Quantify the similarity of gamma power and neural firing rate to substantiate the statement.

g. The authors mention in several places that beta LFP power lags changes in neural firing rate. In the Methods they state that they calculate LFP PSDs using 200 or 400 ms windows? Did these windows end at the represented time t, or were they centered around it? Specifically for Fig. 3, did they use 200 or 400 ms windows? I want to understand how reliable this lag estimate is.

h. I like the simple model that the authors propose. I'd recommend them to rework Fig 5 so it has a schematic of the two alternative models they are considering: the interhemispheric connectivity model and the shared common input model. Adding a cartoon of the expected coherence and coherence vs. lag results for each model would also help lay out the logic for Fig 6.

i. The authors based many of their claims and analyses on LFP oscillations at the beta frequency largely reflecting inputs to the neural population, as widely-accepted by the community. They should substantiate this with the appropriate references, and perhaps discuss an alternative interpretation if there's an appropriate one

j. I found the first paragraph of the Discussion a bit hard to follow. It may be me, but I advise the authors to revise it. Also, the Discussion is a bit too long; it should be more focused

Minor clarifications/suggestions:

- Also consider clarifying how the baseline LFP spectra were computed; it's on the methods but a quick clarification will make the paper easier to follow

- Line 133: clarify that LFP-LFP coherence is across hemispheres

The authors use a pretty consistent color scheme across the figures for the bimanual tasks; I'd suggest they try not to alternate between black meaning saccade or baseline

We thank the reviewers for the constructive comments and suggestions on our previous submission (MS# NCOMMS-18-30170A-Z). We believe our new submission addresses all of the reviewers' concerns. The specific changes to the paper and where they occur are listed below. The reviewer comments are indicated by number and *italicized font*. Our responses are in sans serif font and quoted manuscript text is in serif font.

Reviewer 1:

1.1. One obvious question is why the saccades were allowed (or forced in unimanual conditions) during reaching. I believe that most work in this lab (and many other labs) used reach-only or saccade-only conditions, unless the goal was to study explicitly the eye-hand coordination. I fully understand that it is more difficult for the monkeys to perform unimanual and especially bimanual reaches without accompanying saccades, and that saccade-reach is a more natural behavior. But many PRR cells are responsive to visual stimuli denoting upcoming saccade targets, and are active during the delay period prior to saccades (even if less so than for reaches). Therefore it is not entirely clear how the saccade preparation, which might take place in both hemispheres -- in part because PRR is typically not very contralaterally tuned for space -- might affect the results. It would be beneficial for the narrative if the authors explain why they believe this does not represent a confound, or suggest ways to isolate potentially confounding effects of saccade preparation in their analysis. They should also explain how very likely divergent patterns of saccadic behavior -- e.g. saccading first to one target, then to another in the bimanual-apart reaches, as compared to reaches to the same target -- might have influenced the findings (e.g. the decreased LFP-LFP coherence for the bimanual-apart condition).

We now address this point in multiple ways. Allowing animals to make eye movements is, as the referee notes, a more natural behavior. By default, the animals made spontaneous saccades to the reach targets in all of the tasks. By making this saccade a requirement we standardized the behavior, ensuring that reaches for all but the bimanual conditions were accompanied by very similar saccades. (See below for treatment of the bimanual-apart condition.) We use a saccade-only trial in order to isolate any confounding effects of saccade preparation. We have added the following text to the first paragraph of the results to make the role of the saccade-only trial explicit (Page 3, line 88):

A fifth interleaved condition consisted of a saccade without a reach, as a control to isolate any confounding effects of saccadic eye movements by themselves.

Next, we added the saccade-only trace to the LFP-LFP coherence figure in order to demonstrate that there is no effect of saccades on LFP-LFP coherence compared to baseline (Fig. 2; LFP-LFP coherence during the saccade task [black] closely follows the baseline [gray trace] level of LFP-LFP coherence). We also now show this lack of an effect of the saccade task on coherence in each individual animal's data as well (new Supplementary Fig. 3).

Revised Figure 2:

Figure 2. Beta-band LFP-LFP coherence between PRR in the left and right hemispheres distinguishes bimanual-together and bimanual-apart movements from baseline and from unimanual movements. Coherence in the beta-band (~20-30 Hz) is elevated for bimanual-together movements (blue) and decreased for bimanual-apart movements (purple) compared to unimanual reaches (yellow), saccade only trials (black), or to the baseline period (gray). Data are averaged from the 113 pairs of sites (47 from M1, 66 from M2) recorded simultaneously in the two hemispheres, with coherence measured in the 650 - 1150 ms interval after target onset. Blue and purple asterisks indicate significant differences of bimanual-together and bimanual-apart, respectively, versus unimanual. Gold asterisks denote comparison of bimanual-together versus bimanual-apart ($P < 0.05$ after Bonferroni correction for testing at 30 different frequency bands). Coherence outside the pictured range (from 16 to 20 Hz and 100 to 240 Hz) was unaffected (corrected $P > 0.05$).

Supplementary Figure 3:

Supplementary Figure 3 (related to Figure 2). Beta-band LFP-LFP coherence between PRR in the left and right hemispheres distinguishes bimanual-together and bimanual-apart movements from baseline and from unimanual

movements. Coherence in the beta-band (~20-30 Hz) is elevated for bimanual-together movements (blue) and decreased for bimanual-apart movements (purple) compared to unimanual reaches (gray) or to the baseline period (black). Coherence during saccade trials (black) resembled that seen during unimanual reach (yellow). Data are averaged from the 113 pairs of sites (55 from M1, 64 from M2) recorded simultaneously in the two hemispheres, with coherence measured in the 650 - 1150 ms interval after target onset. Blue and purple asterisks indicate significant differences of bimanual-together and bimanual-apart, respectively, versus unimanual. Gold asterisks denote comparison of bimanual-together versus bimanual-apart ($P < 0.05$ after Bonferroni correction for testing at 30 different frequency bands). Coherence outside the pictured range (from 16 to 20 Hz and 100 to 240 Hz) was unaffected (corrected $P > 0.05$).

We have revised the relevant Results text to state the lack of a saccadic effect explicitly (Page 4, beginning line 150):

During saccade-only and unimanual reach trials, LFP-LFP coherence remained close to baseline across a wide range of frequencies (black, yellow, and gray traces, respectively), supporting prediction (1) above.

We have made sure to include the saccade control condition in virtually every other figure, including supplementary figures, and table. In each case, the saccade control condition serves to “isolate potentially confounding effects of saccade preparation in [our] analysis”, as requested by the referee.

Finally, with regard to the bimanual-apart condition, it is in indeed the case that there are divergent patterns of saccadic behavior. On these trials, the animal makes a saccade first to one target and then the other (Mooshagian et al., 2014; Mooshagian and Snyder, 2018). Importantly, we have previously demonstrated that initial saccade direction does not modulate the delay period activity in PRR (Mooshagian and Snyder, 2018). The same is true for LFP activity. We now address these points in the Discussion (Page 14, beginning line 531):

Additional factors may contribute to the observed coherence effects, including visual effects of one versus two targets, spatial effects of one versus two movement goals, or the effects of having motor plans that are congruent or incongruent. As for an influence of the associated eye movements, we showed previously that the direction of the initial saccade on bimanual-apart trials does not affect PRR unit activity either during planning or movement⁵². Saccade direction also does not affect LFP power in PRR (data not shown).

1.2. The results on spatial tuning of beta power are somewhat puzzling. Beta power in the parietal cortex has been reported to be tuned not only for the task/effector, but also spatially, with inRF more suppressed than outRF, relative to the fixation baseline (e.g. Dean et al., 2012). One notable exception is the study of Scherberger and colleagues (Scherberger et al., 2005), but these results are perplexing (as the authors of the present study note, Scherberger et al. also reported higher beta for reaches than for saccades). The present study shows that the delay beta power is highest for saccades and lowest for contramanual and bimanual movements. This makes total sense, but the reversed effect in inRF (higher beta) vs. outRF (lower beta) does not. To re-phrase it in a simpler way, the first finding is that the “most relevant” (or preferred) for the PRR condition (i.e. contramanual reaches) show less beta. But the second finding is that the “most relevant” (i.e. inRF) direction shows more beta. It might be that the apparent

discrepancy is due to the sorting of in/out (preferred direction) in accordance to the spatial tuning of single unit spiking. Since PRR neurons could have both contralateral and ipsilateral tuning, and the tuning of a single neuron might not be representative of the overall population reflected in the LFP, my suggestion would be to sort the data in Figure S4 to contralateral/ipsilateral space, rather than to preferred/anti-preferred direction in respect to the associated unit.

This criticism seems to misconstrue both our results and the results of Dean et al. 2012. We found **no** significance difference for inRF versus outRF power in the range from 20-30 Hz. (There was a small effect for contramanual reaches, inRF greater than outRF, but this did not survive multiple comparisons correction.) We describe this result in the text:

Legend of Fig. 3,

Responses to preferred and null directions showed no difference and so are merged across 312 sites.

Legend of Supplementary Fig. 9

There is no significant effect of direction (preferred versus null) for saccades and for ipsilateral and bimanual-apart reaches (mean power in the interval 650-1150 ms after target appearance, paired t-tests, $n = 113$ for each comparison, $P > 0.05$). For contralateral and bimanual-together reaches there is a significant effect of direction that does not survive correction for multiple comparisons (5 comparisons).

Results (Page 7, line 254):

... beta power, unlike gamma, is not directionally tuned.

Results (Page 8, line 282):

A second regime occurs at lower frequencies, from about 16 to 32 Hz. Here LFP power depends strongly on both movement type and frequency, but only weakly on direction.

Discussion (Page 12, line 430):

Beta power differed from spikes in its response to movement type and direction. Unlike spikes, beta power was not tuned to the type of movement that was planned.

This result is almost exactly the same as that of Dean et al. (2012). They show that for all frequencies above 20 Hz, the response to inRF is greater than the response to outRF (Figure 3E of Dean et al. 2012). At 20 Hz there is no significant difference, and from 20-30 Hz (the same interval that we report on) there is a small difference with the same polarity as what we report (inRF greater than outRF; in our data the significance disappears with multiple comparison correction). It is only below 20 Hz that Dean et al.

see outRF greater than inRF. (In fact, we also see higher power for outRF compared to inRF for contramanual reaches in the 10 to 16 Hz range, but the effect is not significant even before multiple comparisons testing.)

We now highlight the congruence between our study and Dean et al. (2012) in the Discussion (Page 13, beginning line 461):

Also unlike spikes, beta power was not tuned to the directional preferences of nearby spikes (beta power: **Supplementary Fig. 9**; spikes: **Fig. 3c**; compare this with gamma band power, which shows clear spatial tuning). These results confirm and extend Dean et al. (2012) who showed little or no spatial tuning between 20 and 30 Hz and strong tuning at 35 Hz and above for contramanual reaching. (Dean et al. also saw a significant reversal of spatial tuning below ~18 Hz, which we do not replicate.) The lack of spatial tuning from 20 – 30 Hz could reflect widespread pooling of signals from many individual cells with diverse directional preferences. Pooling of individual cell responses does not, however, fully explain beta power. Population averaged spiking activity is insensitive to trial type, while beta power differentiates each of the 5 types of movement plans (**Fig. 3a versus 3c**). (Spiking activity can be sensitive to trial type in individual cells, but the effects are not systematic from one cell to the next and are therefore lost in the population average ².)

We appreciate the suggestion to split the data by contralateral/ipsilateral space. We completely agree with the referee that this is an interesting and important topic, but it goes beyond the scope of the current paper (see also 1.7). A full treatment of spatial asymmetries should also include whether the reach crosses midline or not, and should consider factors including spike rate, spike timing, LFP power or LFP timing. In these analyses, each of our four reach conditions would need to be considered separately. To include all of this in the current paper would make the paper far too long and would detract from the central point of this paper. We are therefore currently preparing a separate paper to address these closely related but distinct spatial issues.

1.3. Lines 200-206: it is surprising that the authors talk about pre-movement increase in beta power (for the contramanual reaches). The common finding is that beta is disrupted by the transient events such as visual cue and saccade/reach in frontoparietal areas (e.g. Dean et al., 2012; Sendhilnathan et al., 2017). Please show population power time-frequency plots for all conditions, aligned to cue and movement onset.

Supplementary Figure 9 shows LFP power as a function of time aligned to saccade onset, reach onset and the go cue. The increase in beta power (20 – 30 Hz) is most prominent when the data are aligned on the onset of the reach rather than on the cue or on the saccade onset, and the increase is stronger in the bimanual compared to unimanual reach condition (Supplementary Figure 9). Figure 3 in Dean et al. (2012) is saccade-aligned and shows only a unimanual reaching movement, and therefore lacks the power that we have with reach-aligned data and bimanual movements.

Sendhilnathan et al. (2017) is a study of LFP responses to saccades in the *frontal eye fields*, not parietal cortex; perhaps the reviewer had a different study in mind?

As requested, we now present population time-frequency spectrograms for all conditions aligned on the target onset and movement onset (New Supplementary Fig. 11). In the zoomed-in movement-aligned figures (row d) it can be seen that while there is no increase in 20-30 Hz power for saccades or ipsimanual reaches, there is a weak effect for contramanual reaches and a strong effect for bimanual movements. The circles in the figure below (not included in the manuscript) highlight this increase, showing the increase in power -50 to +100 ms relative to reach onset (red circles around light green). The circles do not appear in the actual manuscript.

Supplementary Figure 11. Population spectrograms of the LFP. **a.** LFP power (0 – 100 Hz) as a function of time for each movement type during the delay period aligned to target onset. **b.** Zoomed in view of target-aligned data from 20-45 Hz. **c and d.** Same as **a and b** except data are

aligned to movement onset (saccade onset for the first column, reach onset for the remaining columns). Color code indicates the LFP power relative to the baseline period (500 ms prior to target onset). The color map (“viridis” package of R and matplotlib of python) is designed to be perceptually uniform (<http://cran.mtu.edu/web/packages/viridis/viridis.pdf>). In **c**, white corresponds to power greater than 150%.

{Red circles highlight, for the reviewer, the increase in beta power that begins just before reach onset. These red circles do not appear in the version of the figure in the manuscript. }

1.4. Line 236: “Finally, movement plans affect beta power only ~100 ms after they affect gamma power and single unit activity.” – I would be cautious with interpreting temporal differences in this range given the considerable temporal smoothing. Please comment on this.

We agree that one must be cautious. The critical point is that the divergence in LFP power occurs later than the divergence in single units. We use more temporal smoothing for LFP than for units, and so if anything, temporal smoothing will *reduce* the difference in latencies. To make this clear, we include for the reviewer figures with a range of smoothing. Units are on the left, LFP is on the right, and smoothing decreases as one moves down the page:

We have added the following text to address this point (Page 7, line 258 in **bold**):

Finally, movement plans affect beta power only ~100 ms after they affect gamma power and single unit activity. **While temporal differences must be interpreted cautiously given the temporal smoothing in the beta LFP, this smoothing will mainly shorten the LFP latencies**

and so cannot be responsible for the divergences in LFP power lagging the divergences in the single units.

1.5. Lines 260-264: “A second regime occurs at lower frequencies, from about 16 to 32 Hz. Here LFP power depends strongly on both movement type and frequency, but only weakly on direction. Responses are not clustered but instead are ordered by movement type. Peak power occurs at a low frequency for saccades (~24 Hz), at an intermediate frequency for ipsimanual reaches (~30 Hz), and at still higher frequencies for contramanual, bimanual-together and bimanual-apart reaches.” – The reasoning behind these statements is not clear. Why the authors refer here to the positive peak of the power if their consistent message is that it is the decrease of beta power (and hence, the negative peak) relative to the baseline that is relevant for contramanual reach conditions?

We have rephrased the referenced passage as follows (Page 8, beginning line 284):

The point of minimum power occurs at a low frequency for saccades (≤ 8 Hz), at an intermediate frequency for ipsimanual reaches (~8-16 Hz), and at still higher frequencies for contramanual, bimanual-together and bimanual-apart reaches (~12-20 Hz).

*1.6. Lines 303-306: “Spike-LFP coherence for bimanual-together movements was significantly higher than for bimanual-apart movements over most of the beta range (two-tailed *t* test; asterisks in Fig. 6a). This difference could not be attributed solely to strongly oscillating LFP signals during bimanual-together compared to bimanual-apart trials, since the elevation also occurs in coherence measures that are unaffected by LFP magnitude (spike-LFP synchrony).” – I admit to not following the reasoning here. What’s “spike-field synchrony” – isn’t this what is being estimated by the spike-LFP coherence (but it sounds like those are considered two different things, from the excerpt above)? I am not entirely certain at the moment if there is a full consensus on whether the power of a noisy LFP signal in a certain band is affecting spike-field coherence level in this band – I understand that coherence is normalized to the LFP power, but if there is very little power in a certain frequency then the noise will probably dominate. But let’s assume that the authors meant here that the coherence is not affected by the LFP power.*

We are indeed considering two different measures. Our first, more standard measure of coherence takes into account spike timing, LFP magnitude and LFP phase. Our second measure uses only spike timing and LFP magnitude. In the previous version we used “spike-field synchrony” as a label for the second measure, but are now changing this to the more commonly-used term, “phase locking value”. We have amended the cited passage as follows:

This difference could not be attributed solely to strongly oscillating LFP signals during bimanual-together compared to bimanual-apart trials, since the elevation also occurs in phase

locking value. (Phase locking values take into account only spike timing and LFP phase, not LFP magnitude; see Methods).”

We describe the computation of phase locking value in the Methods (Page 24, beginning line 937) as follows:

Phase-locking values (PLV) were computed for a range of frequencies. For a given frequency, the LFP phase at the time of each spike was estimated with a wavelet transform. Phases were then pooled across trials. PLV was estimated as:

$$PLV = \frac{1}{S} \sum_{s=0}^S \exp(j\theta_s),$$

where θ_s is the phase at the time of spike s and S is the number of spikes. Significance was assessed with a Rayleigh test.

We are sensitive to the noise argument — if LFP amplitude is low enough, the “real” signal will be buried in noise, corrupting our measure of phase. However, this would predict that apparent phase coherence would follow LFP power (Fig. 3), that is, it would be highest for saccades (the condition with the highest LFP power), intermediate for unimanual reaches, and lowest for both bimanual-together and bimanual-apart. This was not the case: it is high for bimanual-together, intermediate for unimanual and low for bimanual-apart (Fig. 6a). We prefer not adding this argument to the manuscript for reasons of clarity and bulk, and since our results using pairwise phase consistency make this argument unnecessary (see next response).

What I am a bit more concerned about is the usage of the coherence, rather than less biased pairwise-phase consistency (PPC) measure for assessing the spike-LFP coupling (Vinck et al., 2012). The authors however addressed this issue by fixing the amount of spikes to 500, and the coherence has been extensively used before. Nevertheless, given that PPC = gains more and more popularity, it would be very interesting to see if the coherence findings also hold for the PPC (but it is a lot of work so I leave it at the discretion of the authors whether to add this analysis).

We agree that this is worthwhile, and we agree that it was a lot of work, but we now include the estimate of spike-LFP coherence using the PPC method (new Supplementary Fig. 12). The results in the beta band are similar.

Supplementary Figure 12. Spike-LFP coherence computed using pairwise phase consistency (PPC). Format same as in Fig. 5c.

1.7. What is however majorly missing from the analysis is the further separation of the data into two conditions – spikes one hemisphere – LFP another hemisphere, and vice versa. For unimanual reaches, this separation can be done in respect to the hemisphere contralateral to the acting hand. For bimanual-together reaches, in respect to the hemisphere contralateral to the target (for bimanual-apart reaches, the meaningful separation is not possible, unless both “crossed” and “uncrossed” conditions were present, in which case I would separate to crossed vs uncrossed conditions – see the question on Methods below in Minor comments). The important question here is whether there is a “directionality” (or asymmetry) of the interactions depending on specific task conditions – for instance, is there more coherence when spikes from the hemisphere that is contralateral to the target are considered, as compared to the spikes from the ipsilateral hemisphere?

We agree that this is an interesting question, but a complex one and one that goes beyond the scope of the current manuscript. Since we have data from both crossed and uncrossed conditions, the questions become relatively complex: moving arm on the same or opposite side as the hemisphere in which the spikes are recorded, crossed or uncrossed reach, target in or out of the response field, etc. Splitting up the data in these many ways may require a larger data set and will certainly require many additional analyses and descriptions that are only peripherally related to the main point of the current manuscript.

1.8. Figure 6B: Can this finding be also expressed in terms of spike-LFP phase relationship analysis (cf. Hawellek et al., 2016), rather than, or in addition to the lag?

In Figure 6B (now Fig. 5f), we considered the issue of whether spike-LFP coherence is likely to arise from direct callosal connections or from common inputs into each hemisphere. To resolve this, we asked whether spikes are causal to LFP or vice versa. It is not clear how the phase relationships between spikes and LFP, with or without an imposed lag, would bear on this particular question. To be fair, the cited paper includes many different analyses, and so we may have overlooked a method that

provides a better approach than lagged coherence. However, without more guidance from the referee, we are lost – we are unclear what analysis is being referred to, and we are unclear what specific hypothesis or question, relevant to the current manuscript, the referee is proposing that we address with the spike-LFP phase relationship analysis.

1.9.1. The methodology of the local field potential would benefit from some clarifications: How LFPs were recorded/preprocessed? There is not enough information to understand some details of the methodology. First, were LFPs in the two hemispheres recorded using common ground/reference? Were the means of the two LFPs subtracted from each channel, to perform a sort of re-referencing, or the analysis was performed on raw LFPs?

LFPs were recorded using a common ground and analyses were performed on raw LFPs. We have expanded our description of our recording procedures and LFP analysis to include this information. Perhaps the referee is concerned that a common ground could result in artifactual LFP coherence. We recognize that we cannot rule this out. However, a strength of our study is that *none of our conclusions depend on absolute coherence. Instead, we draw conclusions based on differences in coherence obtained during performance of different (interleaved) tasks.*

Our revised description of our recording procedures appears on page 21, beginning line 810:

Extracellular recordings were made using glass-coated tungsten electrodes (Alpha Omega, Alpharetta, GA; electrode impedance 0.5-3.0 M ohms at 1kHz). Neuronal activity was referenced to the signal recorded from a steel guide tube in the same recording well. Neural signals were acquired using the Plexon MAP system (Plexon, Inc.). The continuous signals were passed through a preamplifier and then separated into two signal paths. The LFP channel was band-pass filtered between 0.7 to 300 Hz and digitized at 1 kHz. The spike channel was by band-pass filtered between 100 Hz and 8 kHz and digitized at 25 kHz. Single units were isolated online via manually-set waveform triggers.

Our revised description of our LFP analyses appears in the thoroughly revised LFP methods section beginning on page 23, line 926. Please see manuscript.

*1.9.2. Second, the relationship between time windows, sliding window steps, and frequency resolution is not clear. In particular, the reported frequency resolution is very low for the LFP analysis (+/-6.25, +/-12.5 and +/-5Hz, derived from the relationship between window length and the time-half-bandwidth: time-half-bandwidth $TW = NF/2$). But it seems that the plots in Figure 2 and 3 show much higher frequency resolution? For the spike-field coherence, $TW = 0.8 * 15 = 12$, but how this results in only 4 Slepian tapers, given number of tapers $L = 2TW - 1$? I am sure I am missing something.*

We regret the error in the reporting of the number of tapers; 4 tapers were used for the LFP analysis, not for the spike-LFP analysis. This has been corrected in the Methods text (Page 24, line 932).

We elected to use a time-half-bandwidth product of 12 which afforded the use of 23 Slepian tapers and yielded a frequency resolution of ± 15 Hz.

For LFP power and LFP-LFP coherence, we chose the commonly used a time-half-bandwidth of 2.5. Our desire for high temporal resolution (e.g., 200 ms) drove the low frequency resolutions that the referee notes (e.g., $2.5 / 0.2$ s or ± 12.5 Hz). Our aim was not to describe frequency effects with high frequency resolution, rather, our aim was to distinguish effects of different task types. For this purpose, high frequency resolution was not critical.

Graphing conventions: We showed data points at 2 Hz intervals (Fig. 2), but this does not imply that each data point is an independent sample. To use an analogy from the temporal domain, if we lowpass (smooth) time data, then adjacent data points are no longer independent observations. Data filtered by convolving with a 20 ms Gaussian (equivalent to a 6.8 Hz lowpass filter) are often plotted at 1 ms intervals, despite the fact that samples closer than 100 ms to one another are not independent. Returning to the frequency domain, one might consider what other authors have done. In the paper recommended by the referee for its methodological approach (point 1.8, above: Hawellek et al. 2017), spike-field coherence data is shown as a function of frequency in Fig. 2a and Fig. S2. The data are plotted continuously with what appears to be greater than 1 Hz resolution, despite a frequency resolution of 10 Hz (time and bandwidth parameters of 500 ms and 5 Hz).

1.9.3. Incidentally, has Chronux also been used for the LFP power and coherence, or custom code, or native MATLAB functions (e.g. pmtm.m), as opposed to the Chronux usage for the spike-LFP coherence?

Our power and coherence analyses use a multitaper discrete fourier transform. The method is very similar to Matlab's native pmtm method, except that our in-house code allows us to use a GPU coprocessor, and we can specify the frequencies to make measures. We replicated all of our analyses using Chronux and confirmed that the results are essentially identical. Please see revised methods.

1.9. 4. What were the reasons for using the multitaper approach for the low frequencies? It has been suggested that while multitaper works well for gamma frequencies, it introduces too much smoothing into the lower frequency (beta) range: (cf. <http://www.fieldtriptoolbox.org/tutorial/timefrequencyanalysis/>)

The main reason was that our analysis began by analyzing all frequencies, not just gamma and beta. Drawing a hard line – one taper below frequency X and multiple tapers above that frequency – would require a principled way to select that border

frequency. Alternatively, the number of tapers could be adjusted continuously. Either method seems to inject unnecessary complexity into the analysis. Since we are not making claims about narrow frequency bands, it is not clear that over-smoothing the low frequency data is particularly problematic.

1.9.5. Third, the patterns for gamma power and spiking are largely very similar (besides the bimanual apart condition). It might be expected, but given that the separation into the two streams -- spiking and LFP -- was done using the Plexon online filtering (and not via recorded broadband data that provides more flexibility for offline analyses, e.g. noncausal median filtering and removal of spiking waveforms in time domain, and/or and high frequency power from LFPs), there are potential concerns that spikes might have leaked into the high frequency LFP. Since the gamma frequency is not the main point of the study, this is not a critical issue, but perhaps the authors could comment on this.

We do not think it is necessary to comment on this. As the referee notes, gamma frequency is not the main point of this study. In fact, we draw no conclusions from the gamma LFP. The one place where this might be a factor would be in the spike-LFP coherence measures. However, we draw no conclusions based on spike-LFP in the gamma range, and in any case, spike-LFP coherence was always computed across separate electrodes, so contamination is not an issue.

1.10. Discussion is very mechanistic-oriented and lacks the big picture for a wider audience, and some important points for experts in the field. For example, I expected the discussion of the interhemispheric competition / rivalry / push-pull models, and the construal of the increased coherence for bimanual-together condition as facilitatory exchange of information between hemispheres, as compared to potentially inhibitory effects during the bimanual-apart condition with decreased coherence (as one potential, perhaps overly simplistic, interpretation). At the very least, the authors should attempt to relate their findings to a lively debate on the compensatory/beneficial vs. maladaptive recruitment of the intact hemisphere in visuospatial and motor functions during unilateral brain damage, in neurological literature (Murase et al., 2004; Umarova et al., 2011; Bartolomeo and Thiebaut de Schotten, 2016), in noninvasive brain stimulation studies (Johansen-Berg, 2007; O'Shea et al., 2007; Sparing et al., 2009; Koch et al., 2011), and in reversible parietal lesion experiments in monkeys (Wilke et al., 2012).

On a more detailed circuit-oriented level, however, the transcallosal connectivity is mainly mediated via axonal projections to inhibitory interneurons (Bloom and Hynd, 2005; Palmer et al., 2012). How does the known interhemispheric circuitry constrain the schematic model of information transfer the authors have in mind?

Another aspect that needs to be considered is the translational value of these experiments to humans, and its limitations. For example, effects of training are important. Previous work demonstrated differences in motor performance in over-trained monkeys as compared to naïve humans: while in humans bimanual movements

took longer than unimanual ones, movement times of bimanual movements in monkeys were shorter than unimanual ones (Gribova et al., 2002).

We have revised the discussion, keeping in mind the suggestion of Reviewer 3 to shorten its overall length. We quote the relevant portion here:

It appears that the amount of information exchanged between hemispheres increases with bimanual reaches in the same direction (bimanual-together) and decreases with bimanual reaches in opposite directions (bimanual-apart). Increases or decreases in information exchange are likely to facilitate yoked or independent movement of the two arms, respectively. Similar results have been found in motor cortex during the movement period itself ⁴⁷. Temporal coordination is required in both our bimanual tasks, but our results indicate that it is the spatial aspects of the task that are responsible for differences in the interhemispheric exchange of information. If temporal coordination were the key aspect then we would expect increased exchange in both bimanual-together and bimanual-apart tasks ^{48,49}.

The increased coherence for bimanual-together movements can be construed as a facilitatory exchange of information between hemispheres, as compared to the decreased coherence for bimanual-apart movements which can be construed as inhibitory effect of one hemisphere on the other. Competing theories of callosal function focus on either its excitatory ⁵⁰ or inhibitory ⁵¹ role in interhemispheric processing and there is evidence for both in the human literature ⁵². Our data indicate that interhemispheric connections can be functionally either inhibitory or excitatory, depending on the particular task being planned. These data do not speak to whether interhemispheric transmitters themselves are excitatory or inhibitory ⁵³.

1.11. “Temporary inactivation of PRR selectively interferes with contralateral arm movements, and at a population level, unit activity codes only contralateral arm movements.” – I am not sure how well this statement agrees with previous work from the same lab that shows a substantial number of ipsilateral and bilateral arm neurons (Chang et al., 2008).

We have clarified this point (Page 2, beginning line 57):

Temporary inactivation of PRR selectively interferes with contralateral arm movements ¹⁶, and the population average unit activity codes only contralateral arm movements. Chang et al. (2008) showed that some individual cells in PRR reflect plans for ipsilateral limb movements, but Mooshagian et al. ² showed that this response could more parsimoniously be attributed to the presence of a behaviorally relevant stimulus in the cells’ response field, and not to an ipsilateral limb movement plan *per se*.

1.12. “We found that local field potentials (LFP) are coherent across hemispheres, with the extent of the coherence depending on the type of reach being planned. This is consistent with information about reach plans being exchanged across the hemispheres. In fact, beta-band LFP carries task-specific plans for ipsilateral arm

movements.” – The last sentence does not appear the logical continuation of the previous two – how ipsilateral arm encoding is mapped onto interhemispheric exchange?

We have revised the last sentence (Page 2, beginning line 68; the emphasis marking the changed sentence does not appear in the manuscript):

We found that local field potentials (LFP) are coherent across hemispheres, with the extent of the coherence depending on the type of reach being planned. This is consistent with information about reach plans being exchanged across the hemispheres. **Since spiking activity does not contain ipsimanual arm information, but beta-band LFP power does, we suggest that this information originates from PRR in the opposite hemisphere.** This is supported by our finding of interhemispheric spike-LFP coherence in the beta-band that, like LFP-LFP coherence, is modulated by the type of reach being planned. Interhemispheric spike-LFP coherence is maximized when the LFP is lagged compared to spikes by about 15 ms, consistent with the input to one hemisphere (LFP) being driven by the output (spikes) from the opposite hemisphere. Altogether, our results suggest that bimanual reach planning is achieved in part by interhemispheric transfer of information at the level of the parietal cortex.

1.13. “Reaches were made with just the arm ipsilateral to the recording site (ipsimanual), with just the arm contralateral to the recording site (contramanual), or with both arms (bimanual).” – This terminology does not make much sense for simultaneous bihemispheric recordings. I guess the idea is to specify conditions for LFP power and spiking activity from each hemisphere, e.g. in the Figure 3. Please consider rephrasing.

We have modified this passage as follows (page 2, beginning line 83):

Unimanual reaches were made with either the right or left arm, and electrophysiological responses were sorted based on whether the arm was ipsilateral or contralateral to the recording site. With bimanual reaches, both arms reached to the same target (bimanual-together) or each arm reached to a different target (bimanual-apart) (Fig. 1a, b; Supplementary Fig. 1).

Figure 2 legend – what are the “30 frequency bands”? What is the frequency resolution in this plot? It seems that it’s 2Hz, but please see the question about LFP methodology above.

Please see response to 1.9.2 above for our explanation that continuous data are often plotted using symbols at discrete intervals, which do not necessarily correspond to the resolution of that processed data. We plotted values at every 2 Hz, from 18-50 and 76-100 Hz, for a total of 30 frequency bands. We tested for significance at each point. Although we tested at 30 different points, each point is not statistically independent, so a Bonferroni correction of 30 is clearly overkill. However, (1) the correct value for this correction is difficult to compute; (2) a correction factor smaller than 30 might be difficult to justify to the average reader, who is not as savvy as this reviewer; and (3) dropping the number of corrections to 6 (an approximation to the correct number) has little effect

on the results. Therefore, we decided to be conservative and simply use the total number of data points. We have changed the text to read,

($P < 0.05$ after Bonferroni correction for testing at **each of the** 30 different frequency values **plotted in the figure**)

“To isolate the effects of movement type independent of frequency, we normalized by dividing by baseline power at each frequency and then plotted the result as a function of time.” – It seems that %signal change from the baseline is plotted, not a purely divisive normalization. Please clarify.

We omitted the fact that we subtracted one after the division; this omission is now rectified (Page 5, line 178):

To isolate the effects of movement type independent of frequency, we normalized by dividing by baseline power at each frequency, **subtracting 1**, and then plotted the result as a function of time.

Supplementary Figure 5 – there are no “Out of RF” traces here.

The graphical legend has been corrected.

1.14. Lines 450-453: “With this in mind, delay period power was greatest for saccades, followed in order by ipsilateral arm movements, contralateral arm movements, movements of both arms together to the same target, and movements of both arms to two different targets. The peak frequency of these effects varied in the same order, from ~21 Hz for saccades up to ~35 Hz for bimanual-apart movements (Fig. 4).” – same problem as in Results, the positive peaks for saccade and ipsimanual are equated with rebound from negative peaks in the other conditions (and in bimanual-apart, there is really no peak in this range – the power just continues to raise with the frequency). I believe that this is misleading description and it does not lead anywhere.

We have extensively revised the discussion and the referenced passage has been removed.

1.15. Lines 543-545: “The substantial baseline level of beta-band LFP-LFP coherence suggests that information is shared between left and right PRR even when the arms do not move, e.g., in a saccade task or during an intertrial interval.” – Or it is there just because there is shared noise, volume conduction, etc. What kind of information the authors are referring to – postural control, awareness of limbs, small spontaneous movements? In general, I would refrain from equating the apparent LFP-LFP coherence

(functionally significant or not) with the “information”.

We have removed this passage from the discussion.

1.16. *It is indeed commendable that the authors managed to train the monkeys to do bimanual-apart reaches. Was it always that the left hand went to the left target, and the right to the right, or they were also doing even more demanding “crossed” reaches (left hand to right space and vice versa) or only “uncrossed” ones?*

The animals made both crossed and uncrossed reaches. We have clarified this important aspect of the task design in the Methods and Results and added panel Fig. 1b that explicitly shows all the arms configurations for one pair of target positions.

Methods (Page 20, beginning line 781):

When two targets appeared (red and green, “bimanual-apart”), they were separated by 180° relative to the central fixation point, i.e., on the left and right, top and bottom, or at opposed diagonal locations. For bimanual-apart reaches, the arms could be uncrossed or crossed (**Fig. 1b**).

Results (Page 2, beginning line 85):

On bimanual reach trials, both arms reached to the same target (bimanual-together) or each arm reached to a different target at diametrically opposed positions (bimanual-apart) (**Fig. 1a, b; Supplementary Fig. 1**) For bimanual-apart reaches, the arms could be uncrossed or crossed (**Fig. 1b**).

Figure 1b:

b. Unimanual left or right arm reaches were instructed with a single green or red peripheral target, respectively. Reaches with both arms to a single target (bimanual-together) were instructed with a blue target. Reaches with each arm to a different target (bimanual-apart) were instructed with two stimuli separated by 180 degrees across the central fixation. Unimanual reaches were either to targets on the same side of the body (upper row) or crossed to the opposite side of the body (lower row). Bimanual-apart reaches were made with the arms either uncrossed (upper row) and or crossed (lower row). Note that only one of the 4 possible target pairs is illustrated.

1.17. *At any rate, the bimanual-apart reaches are important because this condition can be used to argue that the main finding cannot be explained by the task difficulty/effort. The saccade is the easiest condition, the bimanual-apart is the most difficult, the bimanual-together is arguably easier than apart, but still, the highest LFP-LFP coherence corresponds to the bimanual-together condition. If the authors agree with this reasoning, they might want to mention in the text.*

We have added this important point to the Discussion (Page 14, beginning line 488; bolding does not appear in the manuscript):

The coherence results support the idea that interhemispheric interactions are responsible for the pattern of beta LFP results. Interhemispheric beta LFP-LFP coherence (Fig. 2) and spike-LFP coherence (Fig. 5) are movement-specific. A plan to move both arms to the same target increases beta-band coherence, while planning to move each arm to a different target results in a decrease in beta-band coherence, compared to baseline and unimanual reach levels. There were no modulations of LFP-LFP coherence outside the beta-band. Similarly, a plan to move both arms together increased spike-LFP coherence compared to a plan to move each arm to a different target. Again, this was most prominent in the beta-band. **These results cannot be explained by task difficulty, since coherence did not track the presumed effort required to make each movement. The saccade-only condition is likely easy to perform yet has intermediate coherence, while the bimanual-apart condition is difficult yet has the lowest coherence.**

1.18. *“(<http://chronux.org/>; {Mitra:qIMLon5c}.” –proper reference?*

The reference now appears correctly as:

Mitra P, Bokil H. 2009. Observed Brain Dynamics, Observed Brain Dynamics. Oxford University Press.

1.19. *“Mean values were subtracted from the spike signals before the FFT to remove DC components.” – This is not clear. What are mean values of the “spike signals”? Mean firing rate across all trials? Please explain.*

We have clarified this point in the spike-LFP synchronization methods (Page 24, line 919):

For each trial, Fourier transforms were computed separately for spike and LFP signals using a multitaper method. For each taper, the LFP and spike signals were windowed by the taper and a Fourier transform was computed using a fast Fourier transform (FFT) algorithm. Before computing the FFT of the spikes, the mean spike rate of the trial was subtracted away to remove the DC component.

1.20. *“Percival, D. B., Multitaper, A. W. S. A. F. P. A. 1993. Spectral analysis for physical*

applications: Multitaper and conventional univariate techniques. Cambridge Univ. Press, New York. -- the second author is not multitaper, it's Walden :)

Percival, D. B., and A. T. Walden. Spectral Analysis for Physical Applications. Cambridge, UK: Cambridge University Press, 1993.

Multithanks.

Reviewer 2:

2.1. One of my great frustrations over the years is how exactly PRR is defined and to what degree it does or does not correspond to cortical regions identified by other criteria and nomenclatures. In fact, the source of my frustration is that it has never been clearly defined in any study that I have read about "PRR". It just seems to be wherever they record arm-related activity across a large swath of cortex rostral and medial to the intraparietal sulcus. For instance, Fig 1C shows that many neural data were collected in MIP and in PO. Are both of these cortical regions lumped into "PRR" and are all of those data pooled into the "PRR" data set in this study? If they lump them together, that implies that there are no functionally meaningful differences in activity of reach-related neurons in those two cortical areas, at least with respect to this study, but other studies have treated them as functionally different. Please try to be a bit more specific about what "PRR" means in this study.

As noted by the referee, the parietal reach region (PRR) is defined primarily by functional properties rather than by anatomy. The senior author on the paper coined the term in 1997, precisely because the set of functional properties observed did not cleanly map onto a single anatomically-defined area, and also because there exist multiple anatomical schema in the literature, each using slightly different anatomical boundaries. For example, here we include a reviewer figure that shows the MIP/PO border of Lewis and Van Essen (2000) and the MIP/V6A border of Markov et al. (2014) on the same transverse slice through the posterior parietal cortex. The dashed black oval indicates the approximate extent over which we typically record PRR cells. Notice that there is an approximately 1-2 mm difference of the posterior most aspect of MIP between the two schemes. We also label the LIP/LOP and LIP/PIP on the lateral side of the intraparietal sulcus to illustrate that this sort of variation is not exclusive to MIP.

We include a new section in the Methods wherein we describe our definition of PRR and our cell inclusion criteria (Page 21, beginning line 822):

Definition of PRR

PRR does not fit neatly into any single anatomical area, but instead lies at the boundary of MIP and PO/V6A, though it also extends slightly towards the lateral bank, towards LOP ^{2 9,59}. We therefore functionally identify PRR as a region containing many neurons with a transient response to the presentation of a visual stimulus, especially one instructing a reach; sustained activity during a memory or delayed reach task that is greater than the sustained activity during a memory or delayed saccade task; and another (often small) transient at the time of reach onset. (Not all neurons within the region show all of these properties.)

With this functional definition in hand, we have confirmed in study after study, using either post-mortem track reconstruction or in vivo anatomical MRI with MR-lucent marking lesions, that PRR does not fit neatly into any single anatomical area, but instead lies at the boundary of MIP and PO/V6A, and even extends slightly towards the lateral bank, towards LOP (Snyder et al., 1997; Calton et al., 2002; Mooshagian et al. 2018). (Other authors have since used the term “PRR” for areas that are more anterior than the region we record from. Still other authors recorded responses in regions that we consider PRR but have called them MIP.)

If they lump them together, that implies that there are no functionally meaningful differences in activity of reach-related neurons in those two cortical areas, at least with respect to this study,

That is correct. To illustrate this point, we have computed the population unit response separately for cells that fell within anatomical areas MIP, PO and LOP and show that there are no clear differences (new Supplementary Fig. 2, reproduced below). We are not arguing that the cells in these areas are identical in every way, only that, as the referee notes, we cannot see differences with respect to this study.

Results (Page 3, beginning line 92):

There were no clear differences in the population average single unit activity of the cells recorded in each anatomical area, with respect to the hypotheses or conclusions of this study (**Supplementary Fig. 2**). Similar results were obtained in an independent set of recordings that were performed and published several years earlier from the same animals 2.

Supplementary Figure 2:

Supplementary Figure 2. PRR population activity for all ten conditions by anatomical area. **a.** lateral intraparietal area (LOP). **b.** medial intraparietal area (MIP). **c.** parietal-occipital area (PO). Format same as in Fig. 3c.

2.2.1. *There are a number of technical details about the configuration of reach targets and the nature of the unimanual and bimanual reaches that confuse me, because the*

description of the task is too cryptic. A more detailed figure of the configuration of targets on the monitor would be very helpful:

We have revised Figure 1a to clarify the configuration of targets on the monitor. We also revised Figure 1b to show all possible arm configurations for one pair of diametrically opposed target positions.

Figure 1a:

Revised caption for Figure 1a:

Figure 1. Delayed movement tasks and recording sites. **a.** On each trial, after an initial fixation, a peripheral target appears and instructs the spatial location and effector to be used (eyes or arm[s]) for the subsequent movement(s). The stimulus remains visible during a variable delay period. Throughout saccade and unimanual reach trials, the hand(s) that were not instructed to move must remain on the home pad(s). In response to the go cue (fixation offset), animals execute the instructed movement(s). On single target reach trials, eye movements to the target are also required. Movements are either into or 180 degrees out of the RF. On bimanual-apart reach trials, one arm moves into the RF and the other arm moves out of the RF; eye movements are unconstrained after the go cue. Movement directions (8) and movement types (5) are randomly interleaved. A saccade-only trial (white stimulus) is depicted.

Revised Figure 1b:

b. Unimanual left or right arm reaches were instructed with a single green or red peripheral target, respectively. Reaches with both arms to a single target (bimanual-together) were instructed with a blue stimulus. Reaches with each arm to a different target (bimanual-apart) were instructed with two stimuli separated by 180 degrees across the central fixation. Unimanual reaches were either to targets on the same side of the body (upper row) or crossed to the opposite side of the body (lower row). Bimanual-apart reaches were made with the arms either uncrossed (upper row) and or crossed (lower row). Note that only one of the 4 possible target pairs is illustrated.

2.2.2. Ln 831-832: *“Touch positions on the screen were organized in a virtual 3 X 3 grid centered on the fixation point.” Does that mean that there were two 3 X 3 grids of potential target locations on either side of the fixation point and so on either side of the body, or was there only one 3 X 3 grid centered on the fixation point and body midline, or something else? If the latter, does that mean that the monkeys sometimes reached to targets aligned to their body midline?*

The virtual 3 x 3 grid was centered on the body midline. We have modified Fig.1a to illustrate the grid structure and thereby the potential target locations.

2.2.3. Ln 833-834: *“Plexiglass dividers were mounted on the front of the screen at the middle of each target location.” The first 5 times I read that paragraph, they sounded like barriers that would prevent the monkeys from actually touching each target. I think I finally worked it out but a figure would likely make this much easier to understand.*

We have reworded the referenced passage (Page 20, beginning line 758):

Supplementary Figure 1 shows the touch screen in schematic form. The eight target positions were organized in a rectangle centered on the fixation point, with each target ~8 cm (11 deg of visual angle) from the center fixation point. A small piece of plexiglass (2 in x 3/8 in) oriented in the sagittal plane was mounted on the front of the projection screen to bisect the touching surface at each target location (one plexiglass piece per target). Touches were monitored every 2 ms using sixteen capacitive sensors, mounted on the back of the screen, with one sensor on each side

of each of the 8 possible target locations (to sense reach endpoints) and one sensor at each of the two home pads (to sense reach starting positions). The animals were trained to reach with the left hand to the left side of the plexiglass divider and with the right hand to the right side of the divider so that each hand activated a unique capacitive sensor, even when both hands reached to the same target (Supplementary Fig. 1).

We also include new Supplementary Figure 1 that illustrates the touch screen configuration:

Supplementary Figure 1 (related to Figure 1). Schematic of the touch screen configuration. Targets (gray squares) were arranged in a rectangle around the central fixation point. Touches were detected by capacitive switches mounted behind the Plexiglas touch screen (dashed black circles). To ensure that touches could be correctly attributed to the left or right hand, animals were trained to reach, with the left and right hands, respectively, to the left and right side of a small plexiglass divider mounted to the front of the screen at each touch location (vertical black lines). See text for additional details.

2.2.4. Ln 845-847: "When two targets appeared....at opposed diagonal locations." Since I'm still not sure if there were two 3X3 target grids on either side of the midline or only one 3X3 grid centered on the midline, I don't understand what this means, especially "opposed diagonal locations". I'm going to assume that this means that the two targets were not chosen randomly, but were always yoked by some simple rule. I'm going to assume further that there were 2 grids and that the two targets were always mirror-symmetrically located in the two grids. I hope I was right. Please clarify. Again, a figure showing how target locations might be yoked would be useful.

We believe revised Fig. 1 clarifies the experimental setup. See also our responses to comments 2.2.2, 2.2.3, and 2.2.5.

2.2.5. IF there were TWO grids, did the monkeys only reach to the grid on the same side of the body during the unimanual reaches or did they also reach across their body to the grid on the other side of the body? If they did not do the latter, then the only time they made reaches across their body would have been during bimanual-together trials. Is that correct, or not? If it is correct, then they have no unimanual control for the coherence or power associated with reaches of the arm across the body midline into the contralateral hemi-field during bimanual-together movements.

Reaches were made to both sides of midline. The modified Fig. 1a clarifies the target positions. We have added panel b to illustrate that movements were made to both

sides of space, using both crossed and uncrossed movements. In addition to these changes and those described above, we also added the following text to the Methods (Page 20, line 783):

When two targets appeared (red and green, “bimanual-apart”), they were separated by 180° relative to the central fixation point, i.e., on the left and right, top and bottom, or at opposed diagonal locations. For bimanual-apart reaches, the arms could be uncrossed or crossed (Fig. 1b).

2.3. Ln 120-140, Fig 2: *The prediction that LFP-LFP coherence should increase for the bimanual-together condition compared to the unimanual condition seems reasonable, but why should the coherence decrease for the bimanual-apart condition when the movements are aimed at different targets rather than converging on the same one? They are still temporally coordinated reaches (but see the next comment), and if anything seem to be more demanding since the monkeys must split their attention between the two target locations to ensure accurate reaching for both arms. Why do spatially convergent reaches necessarily mean higher spatial coordination than spatially divergent reaches? They are both spatially-directed actions (c.f., “where versus when”, In 129) that require similar degrees of precision, just not to the same target. At one level, the latter is more spatially coordinated because the biomechanics of the two reaching movements are probably nearly mirror-symmetrically similar, unlike the reaches that converge on the same target. Why should this require less coherence than the unimanual reaches when the other arm is not moving at all? I think that the point that they are trying to make is that when the two arm movements are spatially dissociated, it is useful to functionally decouple the activity in the two hemispheres even beyond that observed during unimanual tasks and one way to do that is to reduce the coherence level further. This would provide a stronger argument than making some vague statements about spatial coordination. It sounds too ad hoc and post hoc.*

You accurately capture the argument we wish to make: When the two arms are spatially dissociated, as in the bimanual-apart condition, activity in the two hemispheres is decoupled. We have revised the relevant passage using your teleological language (Page 4, beginning line 135):

We measured interhemispheric LFP-LFP coherence as a function of frequency and task during the planning/preparatory period 650-1150 ms after target onset. We made the following predictions. (1) When only one arm will move, then bimanual interactions do not occur, and no exchange of information is necessary. Therefore, coherence should remain at or near baseline.

(2) When both arms move towards the same target and must reach that target at a similar time (bimanual-together task), both spatial and temporal coordination of the two limbs are required. This bimanual interaction will require an exchange of information across hemispheres. If this exchange involves PRR, then we expect that interhemispheric coherence will increase, relative to a unimanual reach. (3) When each arm moves toward a different target, that is, if the reaches are spatially dissociated (bimanual-apart task), it is useful to functionally decouple the activity of the activity of the two hemispheres. One way that could happen, or one marker of that happening, would be a decrease in interhemispheric coherence. We therefore predict that the bimanual-apart task will not result in increased coherence relative to a unimanual reach, and the requirement for independent spatial control of each arm might even lead to a decrease in interhemispheric coherence.

2.4. Ln 859-860: "On bimanual trials, the left and right hands were required to hit their target(s) within 500ms of one another." 500ms is nearly forever for monkeys in this kind of task. That could give them enough time to make the two reaches completely sequentially, not simultaneously at all. That statement points to critical data that are missing from this report. This is a study of bimanual coordination and yet there are no data about the degree of coordination of the two arm movements. Did they start simultaneously or did one arm tend to lead the other? Likewise, did the two hands touch the targets simultaneously or in sequence? Did this differ for bimanual-together and bimanual-apart actions? I know that this study focusses on preparatory activity, but it would be important to see these execution-related data to give some insight into how the monkeys viewed these task conditions, e.g., coordinated bimanual actions or sequential unimanual actions?

As you note, the present study focuses on the preparatory period. A full analysis of the behavior during these tasks was were presented in an earlier study (Mooshagian et al., 2014). Rather than repeat all of the details here, we now summarize the important points flagged by the referee, and refer interested readers to that earlier paper:

Results (Page 3, beginning line 95):

Overall behavioral performance was good (Supplementary Table 1, Supplementary Table 2). The median movement times were 170 and 211 ms for M1 and M2, respectively. The timing was consistent from trial to trial; the median standard deviations were 22 and 32 ms, or 13 and 15% of the mean. For bimanual movements, the start and end times of reaches with the two arms were similar. In 80% of bimanual-together trials, the two arms began moving within 66 and 67 ms of each other (M1 and M2, respectively) and ended within 78 and 75 ms of one another. For bimanual-apart trials, these values were 81 and 91 ms, and 90 and 156 ms. For a more complete treatment of behavioral performance in these tasks, see 17.

Supplementary Table 1

Supplementary Table 1. Overall performance by movement condition during neuronal recording.

Movement Condition	Mean Success Rate (SD)	20 th percentile	80 th percentile
Saccade	81 (16)	74	93
Left arm	73 (19)	61	89
Right arm	79 (20)	73	94
Bimanual-together	73 (23)	61	91
Bimanual-apart	75 (16)	58	89

Legend: Mean success rates are computed based on the behavioral data collected for each of the reported neurons ($n = 113$). Percentiles indicate the percentage of individual behavior sets with at least that success rate. SD = standard deviation.

Supplementary Table 2

Supplementary Table 2. Movement duration \pm SD, in ms, by movement condition and direction.

Animal	Movement Condition	Movement Direction							
		0°	45°	90°	180°	135°	225°	270°	315°
M1	Left arm	174 \pm 14	168 \pm 14	161 \pm 20	134 \pm 14	145 \pm 17	115 \pm 11	147 \pm 20	159 \pm 24
M1	Right arm	178 \pm 30	187 \pm 17	206 \pm 28	196 \pm 21	203 \pm 22	198 \pm 32	145 \pm 18	131 \pm 32
M1	Bimanual-together	176 \pm 21	201 \pm 26	193 \pm 19	179 \pm 19	192 \pm 26	170 \pm 41	144 \pm 19	157 \pm 35
M1	Bimanual-apart	169 \pm 26	163 \pm 29	159 \pm 21	195 \pm 26	171 \pm 31	180 \pm 53	158 \pm 19	156 \pm 28
M2	Left arm	263 \pm 40	278 \pm 42	258 \pm 45	176 \pm 35	204 \pm 27	222 \pm 86	177 \pm 26	222 \pm 34
M2	Right arm	180 \pm 23	211 \pm 24	229 \pm 27	223 \pm 30	256 \pm 42	211 \pm 27	161 \pm 23	176 \pm 32
M2	Bimanual-together	209 \pm 32	223 \pm 31	223 \pm 33	194 \pm 27	220 \pm 36	200 \pm 48	153 \pm 20	198 \pm 40
M2	Bimanual-apart	176 \pm 25	203 \pm 47	177 \pm 28	229 \pm 43	247 \pm 55	250 \pm 54	186 \pm 32	204 \pm 43

Legend: Movement durations (time between reach onset and endpoint) in milliseconds (ms), with standard deviation. "Movement direction" refers to the endpoint position, coded as the angle in degrees relative to the central fixation. For bimanual-apart movements, this indicates the direction of the right arm; the left arm moved toward a target positioned 180° across the fixation point from the target of the right arm. M1 = monkey 1; M2 = monkey 2.

2.5. With that last two comments in mind, maybe it might be better to use "bimanual-convergent" and bimanual-divergent" rather than "bimanual-together" and bimanual-apart". I find the former more explicit (and maybe even a little bit more esthetically pleasing!) than the latter. Just a suggestion.

We recognize the appeal of the suggested terminology. However, we hope that our clarifications in response to points 2.2.1 – 2.2.5, above, now make our existing terminology clear. Moreover, we prefer to maintain continuity with the nomenclature we

have used in previously published papers on this line of research (Mooshagian et al., 2014; Mooshagian and Snyder, 2018; Mooshagian et al., 2018).

2.6. A major finding is that the LFP coherence reflects the different uni/bimanual reach conditions, but the spiking activity only reflects the action of the contralateral arm. Is there a disconnect in those findings? If the coherence reflects synaptic inputs from the contralateral hemisphere and elsewhere and plays an important role in bimanual coordination, and if spiking activity represents the outputs that generate the movements, how do changes in bilateral LFP coherence influence bimanual control without having a measurable effect on spiking activity? I'm not suggesting that this disproves their hypothesis, but I would like to know how that can accommodate those two findings.

We failed to address this clearly. The *population* spiking activity only reflects the action of the contralateral arm. Yet nearly half of individual neurons show significant differences in firing rate among the contralateral, bimanual-together, and bimanual-apart reach conditions (Mooshagian et al., 2018), indicating that an effect of movement type is indeed present in the spiking output, but is washed out from the population average firing rate. (Page 13, line 451):

Spiking activity in individual cells can be sensitive to particular trial types, but this sensitivity is not systematic across cells and is largely lost in the population average 2.

2.7. This article focuses on PRR, and implicates PRR in bimanual coordination. But is it not possible that these same findings could also generalize to other cortical motor areas that receive callosal projections from their homologous regions in the other hemisphere? The article almost implies that this function is unique to PRR.

We agree, and now include this thought in the discussion (Page 14, beginning line 552):

We only recorded from PRR. Other pairs of homotopic cortical motor areas in the two hemispheres may exhibit task-specific modulations in their interhemispheric coherence, and these may facilitate yoked or independent movements of the two limbs. While it is by no means clear that interhemispheric circuit mechanisms should be common across brain regions, our proposal that local beta power in PRR reflects a weighted sum of local (ipsilateral) and distal (contralateral) spike inputs is consistent with recent findings from SMA and pre-SMA. In these areas, gamma power was observed to be stronger before a contralateral compared to an ipsilateral arm movement, whereas beta power showed the reverse effect 55.

2.8. Ln 877-878: "...an electrode was placed in PRR in one or both hemispheres." How many sessions involved simultaneous recordings in both hemispheres and how many were only unilateral? Please state such information clearly in one place, such as in the Methods. I can see some numbers buried in figure legends, but it is very frustrating to

have to look around for such basic information at random locations throughout the article.

We have corrected this error in the Methods (Page 24, beginning line 948):

During each recording session, one electrode was placed in PRR in the right hemisphere and one in the left hemisphere. ... We recorded LFP from 312 sites (133 from M1 and 179 from M2) and single units from 113 of those sites (43 from M1 and 70 from M2). In each case, we recorded LFP from both hemispheres simultaneously, along with a single unit from either one or both hemispheres. We obtained data either for targets in all 8 directions (29 neurons, 7, in M1, and 22 in M2) or for the preferred and null directions only (84 neurons, 36 in M1, and 48 in M2). We obtained an average of 15 repetitions for each of the 10 trial types for each of the 113 neurons.

2.9. Ln 369-388, 1115-1178, Fig 7: This material purports to show the results of an “algebraic model” that they present to support their proposal that the beta-band LFP modulations across task conditions could be explained by a simple weighted linear sum of mean spike rates in the two hemispheres across task conditions. They show some spike-rate numbers in the Supplementary Table 1 and the results of the “model” in Fig 7, but they don’t show the “model”. My guess is that it is nothing more than a simple two-term algebraic equation like:

*LFP change = b_1 *Contra spike rate change + b_2 *Ipsi spike rate change,*

where b_1 and b_2 are the relative weights for the two spike rates (e.g., 100:0 or 80:20 from the table). Is that all it is? Please provide the equation. This seems to assume a simple linear relationship between spike rates and LFP power, despite all the disclaimers they listed in Ln 1127-1138, including that “there is no reason to assume that the degree of suppression will be linear with the rate of spiking input.” This whole exercise is too simplistic to be very convincing.

It is correct that the model is indeed a two-term algebraic equation, similar to what the referee describes. We are using the model to test an important point: can the spikes from the two hemispheres explain the task-related modulation of LFP power? Certainly, we could make a very impressive and complex model to test this hypothesis. However, as von Neumann famously said, “With four parameters I can fit an elephant, and with five I can make him wiggle his trunk.” We reasoned that, if a very simple model could explain a substantial amount of variance, then this would be much stronger evidence that the hypothesis was correct, compared to fitting incrementally more variance using a complex model with many more parameters. In response to this comment and comment 3.6 from reviewer 3, we now provide a more detailed description of the model but have moved the bulk of the treatment to Supplementary Material, including our explanation of why we have used such a simple (linear) model when the reality is likely to be more nuanced:

We asked if the LFP power modulation during the planning period is a linear sum of the average modulation of firing rates in the preferred and null directions for the ipsimanual and contramanual arms. The population data were fit to a two-term algebraic model,

$$\text{Mean LFP power modulation} = - \left(b1 * \left(\frac{Mod_p + Mod_n}{2} \right)_C + b2 * \left(\frac{Mod_p + Mod_n}{2} \right)_I \right),$$

where *Mod* is the mean modulation of the population average firing rate, the subscripts *p* and *n* indicate the preferred and null directions, the subscripts *C* and *I* are the contramanual and ipsimanual arm movements, respectively, and *b1* and *b2* are the relative weights for the firing rates for the inputs from the contralateral and ipsilateral hemispheres, respectively, and *b1* and *b2* sum to 1. Finally, the value is negated to reflect the sign reversal between spikes and LFP power. We used the mean preparatory firing rate from 650 to 1150 ms after the target onset as input values (**Fig. 3c**).

... We did not attempt to fit the data to an optimal local to distal input ratio. One could do so, but there are numerous factors that would need to be considered in such an attempt. For example, there is no reason to assume that the degree of suppression will be linear with the rate of the spiking input. In fact, since suppression is bounded in one direction (power cannot drop below zero), there may be a compressive relationship between spike rate and suppression as is seen with shunting inhibition γ_6 , such that progressively greater spike rates exert progressively less of a suppressive effect on LFP power. The choice of this relationship, which could be $1/x$, logarithmic or one of many other functions, is likely to have an outsized influence on the fit. In addition, there is a lag between when a spike is produced and when it affects the LFP, and this lag may be different for spikes from the local versus distal hemispheres. Accounting for these additional factors in the model (ratio of local to distal input, shape of the compressive function, lag times, etc.) would improve the fit, but the fact that those variables are poorly constrained would limit the value of such a model.

Reviewer 3:

3.1.1. The authors have long-standing experience with these experiments, but I'd like them to show raw data and "raw results," not only grand averages. For example, 1) How consistent are movement kinematics? (important for trial-averaging);

We have now included much more "raw data" in the manuscript, primarily in the supplemental material given the limits on the main text. (1) We now summarize the consistency of movement kinematics in the main text, and include many details in the supplementary material (see below). We note in passing that, while we agree that consistent kinematics are important for trial averaging, our focus on the delay period is intended to minimize many of the potential confounds that could be introduced by differences in kinematics. Here are our new additions, starting on page 3, line 95:

Overall behavioral performance was good (Supplementary Table 1, Supplementary Table 2). The median movement times were 170 and 211 ms for M1 and M2, respectively. The timing was consistent from trial to trial; the median standard deviations were 22 and 32 ms, or 13 and 15% of the mean. For bimanual movements, the start and end times of reaches with the two arms were similar. In 80% of bimanual-together trials, the two arms began moving within 66 and 67 ms of each other (M1 and M2, respectively) and ended within 78 and 75 ms of one another. For

bimanual-apart trials, these values were 81 and 91 ms, and 90 and 156 ms. For a more complete treatment of behavioral performance in these tasks, see 17.

Supplementary Table 1

Supplementary Table 1. Overall performance by movement condition during neuronal recording.

Movement Condition	Mean Success Rate (SD)	20 th percentile	80 th percentile
Saccade	81 (16)	74	93
Left arm	73 (19)	61	89
Right arm	79 (20)	73	94
Bimanual-together	73 (23)	61	91
Bimanual-apart	75 (16)	58	89

Legend: Mean success rates are computed based on the behavioral data collected for each of the reported neurons (n = 113). Percentiles indicate the percentage of individual behavior sets with at least that success rate. SD = standard deviation.

Supplementary Table 2

Supplementary Table 2. Movement duration \pm SD, in ms, by movement condition and direction.

Animal	Movement Condition	Movement Direction							
		0°	45°	90°	180°	135°	225°	270°	315°
M1	Left arm	174 \pm 14	168 \pm 14	161 \pm 20	134 \pm 14	145 \pm 17	115 \pm 11	147 \pm 20	159 \pm 24
M1	Right arm	178 \pm 30	187 \pm 17	206 \pm 28	196 \pm 21	203 \pm 22	198 \pm 32	145 \pm 18	131 \pm 32
M1	Bimanual-together	176 \pm 21	201 \pm 26	193 \pm 19	179 \pm 19	192 \pm 26	170 \pm 41	144 \pm 19	157 \pm 35
M1	Bimanual-apart	169 \pm 26	163 \pm 29	159 \pm 21	195 \pm 26	171 \pm 31	180 \pm 53	158 \pm 19	156 \pm 28
M2	Left arm	263 \pm 40	278 \pm 42	258 \pm 45	176 \pm 35	204 \pm 27	222 \pm 86	177 \pm 26	222 \pm 34
M2	Right arm	180 \pm 23	211 \pm 24	229 \pm 27	223 \pm 30	256 \pm 42	211 \pm 27	161 \pm 23	176 \pm 32
M2	Bimanual-together	209 \pm 32	223 \pm 31	223 \pm 33	194 \pm 27	220 \pm 36	200 \pm 48	153 \pm 20	198 \pm 40
M2	Bimanual-apart	176 \pm 25	203 \pm 47	177 \pm 28	229 \pm 43	247 \pm 55	250 \pm 54	186 \pm 32	204 \pm 43

Legend: Movement durations (time between reach onset and endpoint) in milliseconds (ms), with standard deviation. "Movement direction" refers to the endpoint position, coded as the angle in degrees relative to the central fixation. For bimanual-apart movements, this indicates the direction of the right arm; the left arm moved toward a target positioned 180° across the fixation point from the target of the right arm. M1 = monkey 1; M2 = monkey 2.

3.1.2 Are the LFPs in PRR modulated by trial type and target location? (e.g., show PSTHs of the beta and gamma powers, similar to their Suppl Fig 4); 3) Show LFP-LFP coherence traces between a few pairs of example channels for the different task conditions (like Fig. 2); 4) Similar to #2, show PSTHs of a few example single neurons (across trial types and target); 5) Show the averaged results by monkey instead of pooling both monkeys together.

With respect to the effect of trial type on LFP power, Supplementary Fig. 8 shows paired comparisons of LFP power in different movement types for the individual site data. Supplementary Fig. 11 shows population spectrograms of the LFP power as a function of movement type and frequency. There is little effect of target direction on LFP power (Supplementary Fig. 9). We now additionally show individual animal data for LFP-LFP coherence (new Supplementary Fig. 3), beta-power, gamma-power, and unit activity (Supplementary Fig. 7a-c), as well as PSTHs of a few example single neurons (new Supplementary Fig. 6).

Supplementary Figure 3 (related to Figure 2). Beta-band cross-hemisphere LFP-LFP coherence shows the same pattern in each individual animal as in the average. Coherence from ~20 to 30 Hz is elevated for bimanual-together reaches (blue) and decreased for bimanual-apart (purple) compared to unimanual reach (yellow), saccades (black) or in the baseline period (gray). Data are averaged from 55 sites in M1 and 64 in M2, with coherence measured 650 - 1150 ms after target onset. Format as in Figure 2.

Supplementary Figure 6 (related to Figure 3). Neural signals from typical PRR neurons in M1 (a) and M2 (b).

We present population single-unit firing rates separately for each animal in Supplementary Figure 7):

Supplementary Figure 7 (Related to Fig. 3). PRR population activity for all ten conditions in monkey 1 (left) and monkey 2 (right). **a.** Beta-band LFP power (20 – 30 Hz). **b.** Gamma-band LFP power (70 – 120 Hz). **c.** Single unit activity. Format same as in Fig. 3c.

Supplementary Figure 11. Population spectrograms of the LFP power as a function of movement type and frequency. (a, b) aligned to the go cue. (c, d) Same data aligned to movement onset. Same format as a and b. Color code indicates the percent LFP power change relative to the baseline period.

3.2. I couldn't find whether the authors used trial-averaged data or single trials for their analyses, and how they did the pooling across days, sessions and monkeys. This should be clarified, and the authors should provide evidence of consistent behavior (and more details about the monkeys' behavior).

We now state this directly (Methods, page 25, line 964):

Statistical analyses used trial-averaged data. Unless specified otherwise, data were pooled across sessions and monkeys.

We also now summarize the overall behavioral performance (Page 3, beginning line 95) and point readers to our previous behavioral report that provides a complete description of the behavioral performance using this paradigm (in the same animals) (Mooshagian et al., 2014). (See 3.1 for the text and tables.)

*3.3. I'm probably missing something, but why does beta power start decreasing *before* target presentation? (lines 189-190, Suppl Fig 3). Could the monkeys predict that a trial was about to start? --I couldn't find in the Methods if the inter-trial interval was variable or not.*

Each trial began with a fixed 500 ms of fixation. We have clarified this aspect of the task design in the Methods (Page 20, beginning line 773):

The task design and the movement conditions are shown in Fig. 1b. The animals performed delayed saccades or reaches with the left, right, or both arms¹⁷. Animals first fixated on a circular white stimulus ($1.5^\circ \times 1.5^\circ$) centered on the screen in front of them. Left and right hands touched 'home' pads situated at waist height and 20 cm in front of each shoulder. After holding fixation ($\pm 3^\circ$) and initial arm positions, for a fixed duration of 500 ms, either one or two peripheral target(s) ($5^\circ \times 5^\circ$) appeared on the screen.

3.4. Lines 225-226: I'm puzzled by the following statement: "Thus, like unit activity but unlike beta LFP power, gamma LFP power carries relatively little information about ipsilateral arm movement". Do the authors simply mean that gamma LFP power and firing rates aren't more modulated by arm movement than saccades are? Is that their "baseline" for this statement? I am not saying that I disagree about there being a difference between, but I do see some modulation. Since this result is the basis for the model, it should be clarified (and perhaps quantified). The same for Lines 272-273, Lines 505-506, and Lines 645-647.

In general, we mean that there is little difference, at the population level, between the activity (single unit or gamma power) for (1) a contramanual reach to a target, (2) a contramanual reach to a target combined with an ipsimanual reach to that same target, and (3) a contramanual reach to a target combined with an ipsimanual reach to a different target. In other words, at a population level, there is essentially no difference between the ipsilateral arm not moving, moving to the same target as the contralateral arm, or moving to a different target, so long as the contralateral arm moves in the preferred direction. If the contralateral arm does not move, then it is true that there is activity evoked by the ipsimanual reach, but that activity is no different than the activity evoked by a saccade to the same target, suggesting that what drives the activity is not a plan to move the ipsilateral arm, but rather the mere appearance of a relevant target in the response field. (For the single unit activity, this point was central to an earlier publication, Mooshagian et al. 2018).

The entire passage including lines 225-226 has been rewritten to make our meaning more clear (Page 7, beginning line 241). We quote here the most relevant portions:

Adding an ipsilateral arm movement to a contralateral arm movement has little effect on gamma power. That is, the bimanual-apart and bimanual-together responses resemble the contramanual only response. A parsimonious explanation of the activity on ipsimanual trials is that it is a response to a target appearing in the response field. This explanation is consistent with the fact that activity on ipsimanual trials is very similar to that on saccade trials, and mirrors the conclusions drawn concerning single unit responses 2. Thus, while beta power carries information about the movements of each arm, gamma LFP power, like unit activity, carries relatively little information about ipsilateral arm movement.

We have reworded lines 645-647 (now Page 15, beginning line 574):

Population average spiking and gamma-band LFP power encode only contralateral arm movements, while beta-band LFP power contains a rich representation of the pattern of bimanual coordination.

3.5. Fig. 6: the chance coherence values may be this low because of how they were generated. The authors should take firing rates from trials from other task conditions and compute their coherence with LFPs from the condition being analyzed; this shuffling should give a more informative lower bound as it will preserve some key features of the neural data. I'd also like to see the same analysis performed "within hemisphere" (i.e., intra-hemisphere LFP-firing rate coherence). It'd be especially interesting to see whether the LFP-firing rate lag that gives maximum coherence is shorter than for the inter-hemispheric analysis.

We re-computed the spike-LFP coherence using an alternate shuffle-by-trial procedure. The results match those previously computed using our other measure. We now show spike-LFP coherence computed in this manner (Figure 5c):

We agree that computing the intra-hemisphere spike-LFP coherence lag would be informative. However, in this study we recorded in only one PRR site per hemisphere per session, so within-hemisphere comparisons are not possible.

3.6. Fig. 7 and “Algebraic model”: As it stands, this section is not very useful. I’d lean towards expanding the explanation (like they do at the beginning of the Discussion in Lines 407-408, and perhaps add a schematic) and leaving it in the paper, or alternatively leave it all in the Supplementary material. Personally, I’d be more convinced if the comparison between inter-hemispheric and intra-hemispheric LFP-firing rate coherence described above is consistent with the authors’ hypothesis than by this model, although I can see its merit.

We have moved the model completely to the Supplementary material. (As described in the previous response, within-hemisphere comparisons are not possible with our dataset.)

3.7. In the Discussion the authors propose “that reach plans are first produced in PRR contralateral to the limb that will move. We propose further that plans for movement of the ipsilateral limb, which are prominently encoded by beta LFP power and appear only sporadically in spiking activity, likely originate from distal areas that include PRR in the opposite hemisphere” (Lines 494–>). I think it’s a very interesting idea, and part of it could be tested with the current data. For example, the authors could test whether spikes or at least LFPs in the contralateral hemisphere lead those in the ipsilateral hemisphere, to complement their Fig. 6. I’m not a huge fan of asking for lots of additional analyses when reviewing a paper, but this one would be nice.

This suggestion is a nice idea, but unfortunately will not work as described. Our model suggests that cells respond to behaviorally relevant targets in their response field, as well as a plan to move the contralateral arm. Consider two PRR cells, one in the left hemisphere and one in the right, which have overlapping response fields. When a target appears in the response field instructing reach with the left paw, our model suggests that the right PRR cell is driven by a plan to reach with the contralateral arm while the left PRR cell is driven, after a lag or 15 ms or more, by information from the right PRR. However, and this is the critical point, **both** cells are first driven by the presence of a behaviorally relevant target in their (overlapping) response fields. As a result, the earliest spikes in both cells are driven by the presence of a target in the field. The relative timing of this early response relates to crossed and uncrossed visual pathway latencies, which will differ by only a tiny amount. Timing differences due to the reach plans will affect firing only after substantial delays, as can be seen in Fig. 3c, and they will be confounded by the differences in response magnitude and the fact that they will be riding on top of the visual responses. Thus, the referee’s intriguing idea of comparing initial spike timing will not work.

We also cannot implement the reviewer’s suggestion for the low frequency LFP, albeit for a different reason. With low frequency LFP, the responses to movement plans are varying degrees of release from suppression. We see release from suppression first

on saccade trials (black trace, Fig. 3a). This presumably reflects the rapid loss of signals that prolong the suppression. With an ipsilateral reach plan, these suppressive signals are lost slightly later, and with a contralateral reach plan, the suppression weakens only very slowly, and at a still later time. Because these responses reflect the *absence* of a suppressive signal and not the imposition of an excitatory signal, it does not make sense to compare the timing of the changes in the signals in the way that is suggested.

3.8. What happens at the higher gamma frequencies? Some groups have shown that there's task-relevant information even at 200-400 Hz in M1 (Bansai et al J Neurophysiol 107, 2012). Is this also true for PPR?

This seems perfectly plausible, though reviewer 1's concerns about bleed-through of spikes contaminating the gamma power would be a large concern in this case. Since we focus on effects in the beta frequency range here, we believe this is somewhat off-topic for the current manuscript and would be better addressed in a separate manuscript.

3.9.1. Ames & Churchland (bioRxiv 2018) have recently studied M1 activity during ipsilateral and contralateral movements. Similar to previous studies (e.g., Cisek et al J Neurophysiol 89, 2003), many single motor cortical neurons were active during both ipsilateral and contralateral movements. But Ames & Churchland present a very interesting novel result: neurons covaried among them in a fully condition-dependent manner (contralateral vs. ipsilateral), meaning that they occupy orthogonal subspaces or "neural manifolds." I think that applying this method to the authors' data would complement nicely their LFP-firing rate coherence analysis at the neural population level, making the paper stronger. For example, I'd expect neural populations to plan "bimanual-apart" and "bimanual-together" reaches in orthogonal subspaces.

The analysis of Ames & Churchland requires recording from a large number of neurons simultaneously. We record only two neurons in PRR per session, one on the right and one on the left hemisphere.

Minor comments:

3.9.2. The authors should report how many trials for each condition and monkey they've averaged. Also, did they use concatenated single trials or trial-averaged data for the different analyses? In my opinion, this information should be upfront in the Results --I also couldn't find it on the Methods. This also ties up to my request to see more raw data.

Our analyses were conducted on trial-averaged data; this has been added to the text (see 3.2). We now state the number of trials per condition, first in the Methods (Page 21, beginning line 816):

Data were then collected for all trial types (**Fig. 1b**). We recorded LFP from 312 sites (133 from M1 and 179 from M2) and single units from 113 of those sites (43 from M1 and 70 from M2). In

each case, we recorded LFP from both hemispheres simultaneously, along with a single unit from either one or both hemispheres. We obtained data either for targets in all 8 directions (29 neurons, 7, in M1, and 22 in M2) or for the preferred and null directions only (84 neurons, 36 in M1, and 48 in M2). We obtained an average of 15 repetitions for each of the 10 trial types for each of the 113 neurons.

We also give the number of cells recorded from each animal at the beginning of the Results (Page 3, beginning line 90):

We recorded LFP from 312 sites (133 from M1 and 179 from M2) and recorded single units from 113 of those sites (43 from M1 and 70 from M2) (Fig. 1c).

3.9.3. The authors repeatedly say “spikes” but they are analyzing “neural firing rates”; perhaps the results would be completely different if they assumed a “precise spike timing” perspective. I’m not suggesting the authors redo the whole paper, but rather that they use a more precise language.

In Figure 3C, we are showing how many spikes occur in each consecutive small slice of time (temporal resolution 1 ms, lowpass filtered). Here we see the reviewer’s point – we are not considering individual spikes, but rather binned spikes or firing rate. However, the important consideration of spikes comes in the spike-LFP coherence analyses (Fig. 6). Here we are in fact considering the precise timing of each individual spike, in relation to the LFP recorded at the same moment. Therefore, we feel that it is indeed appropriate for us to refer to “spikes” rather than “neural firing rates”.

3.9.4. I don’t know what I’m supposed to take out of Suppl Fig 2 on top of the known log-scale of the LFP power. Perhaps highlight some features of interest for your tasks?

This figure (now Supplementary Fig. 5) shows relatively unprocessed LFP power data, rather than the highly processed data shown in the other figures. This is fully in line with this referee’s request for more “raw data”. This figure is intended to help readers understand how we go from the raw recordings of absolute power across the entire frequency spectrum, to the more processed formats of Figures 3a and b, which show relative change in power for restricted portions of the spectrum. The main feature to be seen in the raw data is that even in this unprocessed form, it is clear that the largest task-specific modulation occurs in the 12-30 Hz range. The text has been edited to highlight this point (Page 5, beginning line 177):

LFP power (amplitude squared), computed during the preparatory period as a function of frequency on each trial and averaged across trials, varies with planned movement type and with LFP frequency (**Supplementary Fig. 5**). **Task-specific modulation is most prominent in the frequency range from 12-30 Hz.** To isolate the effects of movement type independent of frequency, we normalized by dividing by baseline power at each frequency, subtracting 1, and then plotted the result as a function of time.

3.9.5. Lines 166-168: What does the following statement mean? “This contrasts with

single units, whose firing rates distinguish only target location and the pattern of contralateral arm movement at the population level, and even at the single unit level show only sporadic and non-systematic effects of bimanual movements.” Given the increasing emphasis on population level analyses (see Cunningham & Yu Nature Neurosci 2014) the authors should clarify what they mean by “population level”. Also, I’d be surprised if they couldn’t find specific neurons that had different responses across task conditions. Again, it would be nice to see more raw data.

By “population level”, we mean simply the average firing rate across the population. We have amended the text to make this clear (below). It is true that at the level of specific neurons we can find neurons with different levels of firing across task conditions. We provide some examples of that in Supplementary Figure 5, as well as reference our previous publication that described exactly these responses in detail (Page 5, beginning line 184):

This analysis revealed that LFP power in the beta range (20-30 Hz) contains substantial information about bimanual and ipsilateral arm movements (**Fig. 3a**). Different movement plans were associated with differences in power of 10-20% (all $P < 0.05$ even after correction for the 10 possible comparisons). This contrasts with single units, whose firing rates distinguish only target location and the pattern of contralateral arm movement at the population level, and even at the single unit level show only sporadic and non-systematic effects of bimanual movements (**Fig. 3c; Supplementary Fig.6**; see **Supplementary Fig. 7a** for individual animals; see also 2 for a detailed analysis of these effects).

We’ve made similar adjustments throughout the text.

3.9.6. Lines 196-197: “The fact that spikes lead LFP power leads to the surprising conclusion that modulations of firing rate by task are not driven by modulations in beta power.” -> refer to where this is shown in the paper –I understand it’s Fig. 3?– and quantify more rigorously.

We have reworded this and the preceding two sentences to make the reference clear (Page 7, beginning line 214):

Interestingly, task-specific effects appear only ~250 ms after target onset in the LFP data, ~100 ms later than in the spiking data (compare divergences in Fig. 3, panels a and c). This observation implies that modulations of firing rate by task are not driven by modulations in beta power.

3.9.7. Lines 208-210: Quantify the similarity of gamma power and neural firing rate to substantiate the statement.

We are planning a separate publication to look in to this more carefully. The field currently treats gamma power as a good proxy for firing rate – we now provide a citation

for this (reff 19). This is generally true for our data as well. In particular, our point is that the gamma power looks much more like the neural firing rate than the beta power. The beta power shows 5 levels, one for each trial type, while gamma shows only 3 levels, like the firing rates. Gamma and neural activity show the same ordering of response amplitudes, while beta power shows the reverse order.

Despite these similarities, there are some important differences. We feel that the most important difference is in the timing of the peak response, which for neuronal firing occurs ~50 ms before reach onset but for gamma power occurs ~50 ms after onset. If one is using the gamma power as a proxy for neuronal activity in a brain-machine interface, this difference could be very consequential. Our early drafts of this paper included substantial detail about this and other differences, but we realized that the additional figures, analyses and discussion of these points detracted from the main point of the paper. Hence our decision to remove those analyses and put them instead in a separate paper that would focus on those effects.

3.9.8. The authors mention in several places that beta LFP power lags changes in neural firing rate. In the Methods they state that they calculate LFP PSDs using 200 or 400 ms windows? Did these windows end at the represented time t , or were they centered around it? Specifically for Fig. 3, did they use 200 or 400 ms windows? I want to understand how reliable this lag estimate is.

We had originally used 400 ms (± 200 ms) windows for Fig. 3a, but decided to use ± 100 ms windows for better temporal resolution. We did not update all of the text, leading to the confusion – thank you for catching that. In the actual figure, the windows are ± 100 ms, centered on the data points, for both panels a and b of Figure 3. The legend and Methods are now corrected (Page 23, line 876):

Power time signals were estimated by stepping the time window by either 100 ms (400 ms windows) or 50 ms (200 ms windows) and estimating band-limited power centered at each time step.

Finally, to directly address the issue of reliability, we include figures for the referee showing effects of different sized windows. Window size does affect the lag, but larger windows shift the beta power lag **earlier** in time. Thus, the observation that the change in beta power lags the change in neuronal firing is **reduced** by strong temporal filtering, and therefore cannot be a **product** (artifact) of that filtering:

3.9.9. I like the simple model that the authors propose. I'd recommend them to rework Fig 5. so it has a schematic of the two alternative models they are considering: the interhemispheric connectivity model and the shared common input model. Adding a cartoon of the expected coherence and coherence vs. lag results for each model would also help lay out the logic for Fig 6.

We have revised Fig. 5 as suggested and combined it with Fig. 6 so that the model, predictions, and data appear together in one figure:

Revised caption for Figure 5:

Figure 5. Interhemispheric beta-band spike-LFP coherence distinguishes between bimanual-together and bimanual-apart movements. **a.** Schematic model depicting local and distal inputs and outputs to and from PRR. Mass input and output of PRR is shown as arrows. The axon terminals of most neurons contact other neurons locally (light gray arrows), but some portion project distally, including to the homotopic area in the opposite hemisphere (dark gray arrows). Connections with non-homotopic areas are omitted for clarity. **b.** Identical spike-LFP coherence predictions under interhemispheric communication scenario (black curve) and common input scenario (gray curve). **c.** From 20-50 Hz, spike-LFP coherence is consistently high for bimanual-together movements (solid blue), intermediate for unimanual movements (solid green), and low for bimanual-apart movements (solid purple). The distributions of coherence expected by chance were computed by shuffling interspike intervals. The medians of these distributions are shown as dashed traces. The gray shaded region covers 95% of the values expected by chance; values that exceed this are marked by thickened lines ($P < 0.05$). Values exceeding the light gray line are significant at $P < 0.01$. Note that there is no effect of movement type on the shuffled coherences. For this reason, data are pooled across movement types for the $P < 0.05$ and $P < 0.01$ thresholds. Coherences that are significantly larger for bimanual-together compared to bimanual-apart are indicated by asterisks (two-tailed t-test, $P < 0.05$). Dashed vertical lines indicate the beta range

(20 – 30 Hz). Coherence was measured during the 800 ms before the go cue. Data are averaged from 42 pairs of sites (24 from M1, 18 from M2) recorded simultaneously in the two hemispheres. Only sites with at least 500 spikes are shown. **d.** Schematic model depicting common input to PRR in each hemisphere. Spike-LFP coherence predictions for common input model. **e.** Lagged spike-LFP coherence predictions for direct communication and common input models, respectively. **f.** Extremes of spike-LFP coherence (24-38 Hz) occur when spikes lead LFP by 10-15 ms (gray lines). Positive and negative x-axis values indicate the relative temporal relationship between spike times and the LFP in the original data.

3.9.10. The authors based many of their claims and analyses on LFP oscillations at the beta frequency largely reflecting inputs to the neural population, as widely-accepted by the community. They should substantiate this with the appropriate references, and perhaps discuss an alternative interpretation if there's an appropriate one.

We now include the appropriate references (Page 9, beginning line 302):

LFP is thought to be generated primarily by dendritic currents ²¹⁻²⁴. Therefore it reflects not only local recurrent signals but also distal input, while spiking activity carries local recurrent signals plus local output ^{25,26}.

New References:

Mitzdorf U. 1985. Current source-density method and application in cat cerebral cortex: investigation of evoked potentials and EEG phenomena. *Physiol Rev.* 65:37–100.

Mitzdorf U. 1987. Properties of the evoked potential generators: current source-density analysis of visually evoked potentials in the cat cortex. *Int J Neurosci.* 33:33–59.

Einevoll GT, Pettersen KH, Devor A, Ulbert I, Halgren E, Dale AM. 2007. Laminar population analysis: estimating firing rates and evoked synaptic activity from multielectrode recordings in rat barrel cortex. *J Neurophysiol.* 97:2174–2190.

Pettersen KH, Hagen E, Einevoll GT. 2008. Estimation of population firing rates and current source densities from laminar electrode recordings. *J Comput Neurosci.* 24:291–313.

3.9.11. I found the first paragraph of the Discussion a bit hard to follow. It may be me, but I advise the authors to revise it. Also, the Discussion is a bit too long; it should be more focused.

We have revised the first paragraph of the Discussion (Page 11, beginning line 374):

This study reveals mechanisms of bimanual coordination in the posterior parietal cortex. Our findings support the hypothesis that the beta-band LFP (PRR input) is driven by spikes (PRR

output) from the opposite hemisphere. First, interhemispheric spike-LFP coherence in the beta band rises while planning bimanual-together movements and falls while planning bimanual-apart movements. Second, there is a reliable relationship between spikes in one hemisphere and LFP in the other (**Fig. 5c**), consistent with direct communication between left and right PRR. Third, this relationship is strongest when spikes are compared with LFP that occurs 10-15 ms later in time (**Fig. 5h**). Our findings also demonstrate that beta-band LFP power contains information about planned movements of either arm that is not present in the population average spiking output (**Fig. 3**). Taken together, the data indicate that signals describing how the contralateral arm will move are sent from one PRR to the other, and that these signals are manifest in the beta-band LFP of the receiving hemisphere. These signals are likely to be used by PRR to adjust planned movements of the contralateral limb in the service of bimanual coordination.

More generally, the Discussion has been streamlined. We do not include the entire Discussion here. Please see manuscript.

Minor clarifications/suggestions:

3.10. Also consider clarifying how the baseline LFP spectra were computed; it's on the methods but a quick clarification will make the paper easier to follow.

We have rephrased the pertinent portion of the Methods as follows (Page 23, beginning line 883):

In most cases, we present power time signals and power spectral density as percentage of baseline power or power spectral density, respectively. Baseline values are estimated as the average value over the 500 ms before target presentation.

3.11. Line 133: clarify that LFP-LFP coherence is across hemispheres

We now specify that we are measuring interhemispheric LFP-LFP coherence (Page 4, line 150):

Prior to the delivery of a task instruction, **interhemispheric (across hemispheres)** LFP-LFP coherence is frequency dependent, ranging from 0.17 to 0.24 on a scale from completely independent (0.00) to completely coherent (1.00) (**Fig. 2**, black trace).

3.12. The authors use a pretty consistent color scheme across the figures for the bimanual tasks; I'd suggest they try not to alternate between black meaning saccade or baseline.

We now use black only to refer to saccades. Baseline traces are dark gray. Combined ipsimanual and contramanual traces (unimanual) are yellow. These changes have been applied to Fig. 2 and Supplementary Figs. 3 and 4.

Reviewers' Comments:

Reviewer #1:

Remarks to the Author:

The authors made extensive revisions in response to the three very detailed previous reviews, and significantly improved many aspects of the manuscript. I still have few relatively minor points that should be addressed in a minor revision, as listed below. [My initial comments are in square brackets]. Congratulations on an important and well-done study!

Major comments

1)

[One obvious question is why the saccades were allowed (or forced in unimanual conditions) during reaching. I believe that most work in this lab (and many other labs) used reach-only or saccade-only conditions, unless the goal was to study explicitly the eye-hand coordination. I fully understand that it is more difficult for the monkeys to perform unimanual and especially bimanual reaches without accompanying saccades, and that saccade-reach is a more natural behavior. But many PRR cells are responsive to visual stimuli denoting upcoming saccade targets, and are active during the delay period prior to saccades (even if less so than for reaches). Therefore it is not entirely clear how the saccade preparation, which might take place in both hemispheres -- in part because PRR is typically not very contralaterally tuned for space -- might affect the results. It would be beneficial for the narrative if the authors explain why they believe this does not represent a confound, or suggest ways to isolate potentially confounding effects of saccade preparation in their analysis. They should also explain how very likely divergent patterns of saccadic behavior -- e.g. saccading first to one target, then to another in the bimanual-apart reaches, as compared to reaches to the same target -- might have influenced the findings (e.g. the decreased LFP-LFP coherence for the bimanual-apart condition).]

This issue has been addressed.

2)

[The results on spatial tuning of beta power are somewhat puzzling. Beta power in the parietal cortex has been reported to be tuned not only for the task/effector, but also spatially, with inRF more suppressed than outRF, relative to the fixation baseline (e.g. Dean et al., 2012). One notable exception is the study of Scherberger and colleagues (Scherberger et al., 2005), but these results are perplexing (as the authors of the present study note, Scherberger et al. also reported higher beta for reaches than for saccades). The present study shows that the delay beta power is highest for saccades and lowest for contramanual and bimanual movements. This makes total sense, but the reversed effect in inRF (higher beta) vs. outRF (lower beta) does not. To re-phrase it in a simpler way, the first finding is that the "most relevant" (or preferred) for the PRR condition (i.e. contramanual reaches) show less beta. But the second finding is that the "most relevant" (i.e. inRF) direction shows more beta. It might be that the apparent discrepancy is due to the sorting of in/out (preferred direction) in accordance to the spatial tuning of single unit spiking. Since PRR neurons could have both contralateral and ipsilateral tuning, and the tuning of a single neuron might not be representative of the overall population reflected in the LFP, my suggestion would be to sort the data in Figure S4 to contralateral/ipsilateral space, rather than to preferred/anti-preferred direction in respect to the associated unit.]

I partially disagree with the argumentation of the authors here:

"This criticism seems to misconstrue both our results and the results of Dean et al. 2012. We found no significance difference for inRF versus outRF power in the range from 20-30 Hz. (There was a small effect for contramanual reaches, inRF greater than outRF, but this did not survive multiple comparisons correction.) We describe this result in the text:..."

The direction of the effect (inRF > outRF), clearly visible in Suppl. Figure 9, is opposite to the expectations set by the previous literature and the ordering of the beta power in this study: namely,

saccade > ipsimanual > contramanual -- even if the difference is not getting significant after correcting in a specific way for multiple comparisons. There are many ways to correct for multiple comparisons, and I would prefer not to go into the discussions about the stats here; this is not the point. The point is the direction of the effect is puzzling, if you focus on "beta" as the 20-30 Hz range. Why specifically 20-30 Hz range (only high beta)? It is not readily clear from the Suppl. Figure 5 (where the power distribution is nearly linear on a log-log plot, without any noticeable bump in beta range).

"This result is almost exactly the same as that of Dean et al. (2012). They show that for all frequencies above 20 Hz, the response to inRF is greater than the response to outRF (Figure 3E of Dean et al. 2012). "

Figure 3E is for saccades in LIP. This is what Dean et al. say about PRR: "During the memory period before movement, directional selectivity of LFP activity in PRR is most pronounced in activity around 15 Hz. In addition, similar to area LIP, selectivity of memory period activity around 15 Hz depends on whether a reach is made with a saccade ($p < 0.05$, rank-sum test, Fig S4a,b, see Fig 8c). As in area LIP, directional selectivity associated with a reach is more often associated with a reduction in power before movements to the preferred direction (Fig S4c)."

"At 20 Hz there is no significant difference, and from 20-30 Hz (the same interval that we report on) there is a small difference with the same polarity as what we report (inRF greater than outRF; in our data the significance disappears with multiple comparison correction). It is only below 20 Hz that Dean et al. see outRF greater than inRF. "

I agree with the authors that a careful inspection of Dean et al. figures shows the difference in directional tuning between the lower beta (which they nearly exclusively focus on) and the higher beta in 20-30 Hz range. But most readers might not go into such details and would be confused why "beta" (however it is defined) behaves differently in these studies.

"We now highlight the congruence between our study and Dean et al. (2012) in the Discussion (Page 13, beginning line 461):..."

I believe that phrased this way, it will cause further confusion about the role and the tuning of the beta. Dean et al., and many other studies from Pesaran lab call 15-17 or ~20 Hz "beta", while in this study "beta" refers to 20-30 Hz. Dean et al. focus on "their" beta range because they find most power in this narrow band (see their Figure 3D), and they clearly intend to say that (their) beta is lower for preferred than for null direction (and the lower beta is in fact associated with faster RTs). My suggestion is to highlight the lack of the expected spatial tuning in the lower beta range in the current data/analysis, because this is where several previous studies found it, and to mention the distinction between different ranges of beta.

"(In fact, we also see higher power for outRF compared to inRF for contramanual reaches in the 10 to 16 Hz range, but the effect is not significant even before multiple comparisons testing.)".

This is important information, is there a figure that shows this?

By the way, there is a typo in Suppl. Figure 7A ("20-20 Hz").

3)

[Lines 200-206: it is surprising that the authors talk about pre-movement increase in beta power (for the contramanual reaches). The common finding is that beta is disrupted by the transient events such as visual cue and saccade/reach in frontoparietal areas (e.g. Dean et al., 2012; Sendhilnathan et al., 2017). Please show population power time-frequency plots for all conditions, aligned to cue and movement onset.]

"Supplementary Figure 9 shows LFP power as a function of time aligned to saccade onset, reach onset and the go cue. The increase in beta power (20 – 30 Hz) is most prominent when the data are aligned on the onset of the reach rather than on the cue or on the saccade onset, and the increase is stronger in the bimanual compared to unimanual reach condition (Supplementary Figure 9). Figure 3 in Dean et al. (2012) is saccade-aligned and shows only a unimanual reaching movement, and therefore lacks the power that we have with reach-aligned data and bimanual movements."

Here the authors meant Suppl. Figure 10, not 9, I assume. I see only (a positive) increase in the bimanual-apart condition, and only after the movement onset. Do authors mean a positive slope (rebound) after the trough, rather than the increase (relative to the baseline)?

I have to admit that the Suppl. Figure 11 did not impress me too much. First, I find the asymmetric log scale with this very gradual colormap really hard to read and interpret (perhaps my subjective perception is not "uniform", but I highly recommend changing the scale to jet, or red-white-blue, where decreases and increases can easily be distinguished). The few yellowish cells in the highlighted bimanual conditions seem to be leaking from the lower frequencies outside of the beta range. For the contramanual condition, I see pre- and peri-movement decrease in what the authors define as beta range (20-30 Hz), very similar to the Dean et al.

All in all, I agree with the authors that there seems to be a difference in the timecourse of beta power for reaches involving contramanual hand as compared to saccades and ipsimanual hand, but I would refrain from the statements such as "Subsequently, the power for trial types that include a contralateral arm movement (contramanual, bimanual-together and bimanual-apart) increases, reaching a maximum around the time of reach onset." – I do not think this is correct, in all three conditions the power continues to rebound to the pre-reach level and for bimanual-apart – beyond -- cf. Figure 10.

"Sendhilnathan et al. (2017) is a study of LFP responses to saccades in the frontal eye fields, not parietal cortex; perhaps the reviewer had a different study in mind?" – no, I had this specific study in mind; I was talking about the frontoparietal circuitry, not just PPC.

4)

[Line 236: "Finally, movement plans affect beta power only ~100 ms after they affect gamma power and single unit activity." – I would be cautious with interpreting temporal differences in this range given the considerable temporal smoothing. Please comment on this.]

This issue has been addressed.

5)

[Lines 260-264: "A second regime occurs at lower frequencies, from about 16 to 32 Hz. Here LFP power depends strongly on both movement type and frequency, but only weakly on direction. Responses are not clustered but instead are ordered by movement type. Peak power occurs at a low frequency for saccades (~24 Hz), at an intermediate frequency for ipsimanual reaches (~30 Hz), and at still higher frequencies for contramanual, bimanual-together and bimanual-apart reaches." – The reasoning behind these statements is not clear. Why the authors refer here to the positive peak of the power if their consistent message is that it is the decrease of beta power (and hence, the negative peak) relative to the baseline that is relevant for contramanual reach conditions?]

This issue has been addressed.

6)

[Lines 303-306: "Spike-LFP coherence for bimanual-together movements was significantly higher than for bimanual-apart movements over most of the beta range (two-tailed t test; asterisks in Fig. 6a). This difference could not be attributed solely to strongly oscillating LFP signals during bimanual-together compared to bimanual-apart trials, since the elevation also occurs in coherence measures that are unaffected by LFP magnitude (spike-LFP synchrony)." – I admit to not following the reasoning here. What's "spike-field synchrony" – isn't this what is being estimated by the spike-LFP coherence (but it sounds like those are considered two different things, from the excerpt above)? I am not entirely certain at the moment if there is a full consensus on whether the power of a noisy LFP signal in a certain band is affecting spike-field coherence level in this band – I understand that coherence is normalized to the LFP power, but if there is very little power in a certain frequency then the noise will probably dominate. But let's assume that the authors meant here that the coherence is not affected by the LFP power.]

What I am a bit more concerned about is the usage of the coherence, rather than less biased pairwise-phase consistency (PPC) measure for assessing the spike-LFP coupling (Vinck et al., 2012). The authors however addressed this issue by fixing the amount of spikes to 500, and the coherence has been extensively used before. Nevertheless, given that PPC = gains more and more popularity, it would be very interesting to see if the coherence findings also hold for the PPC (but it is a lot of work so I leave it at the discretion of the authors whether to add this analysis).]

This issue has been addressed.

7)

[What is however majorly missing from the analysis is the further separation of the data into two conditions – spikes one hemisphere – LFP another hemisphere, and vice versa. For unimanual reaches, this separation can be done in respect to the hemisphere contralateral to the acting hand. For bimanual-together reaches, in respect to the hemisphere contralateral to the target (for bimanual-apart reaches, the meaningful separation is not possible, unless both "crossed" and "uncrossed" conditions were present, in which case I would separate to crossed vs uncrossed conditions – see the question on Methods below in Minor comments). The important question here is whether there is a "directionality" (or asymmetry) of the interactions depending on specific task conditions – for instance, is there more coherence when spikes from the hemisphere that is contralateral to the target are considered, as compared to the spikes from the ipsilateral hemisphere?]

"We agree that this is an interesting question, but a complex one and one that goes beyond the scope of the current manuscript. Since we have data from both crossed and uncrossed conditions, the questions become relatively complex: moving arm on the same or opposite side as the hemisphere in which the spikes are recorded, crossed or uncrossed reach, target in or out of the response field, etc. Splitting up the data in these many ways may require a larger data set and will certainly require many additional analyses and descriptions that are only peripherally related to the main point of the current manuscript."

I respectfully disagree that this is only peripherally related to the main point of the current manuscript – I think the directionality is the core question of inter-hemispheric interactions. But if authors choose to avoid addressing this question, I suggest they acknowledge this in the limitations of their study.

8)

[Figure 6B: Can this finding be also expressed in terms of spike-LFP phase relationship analysis (cf. Hawellek et al., 2016), rather than, or in addition to the lag?]

This issue has been addressed.

9)

[The methodology of the local field potential would benefit from some clarifications:

How LFPs were recorded/preprocessed? There is not enough information to understand some details of the methodology. First, were LFPs in the two hemispheres recorded using common ground/reference? Were the means of the two LFPs subtracted from each channel, to perform a sort of re-referencing, or the analysis was performed on raw LFPs?

Second, the relationship between time windows, sliding window steps, and frequency resolution is not clear. In particular, the reported frequency resolution is very low for the LFP analysis (± 6.25 , ± 12.5 and ± 5 Hz, derived from the relationship between window length and the time-half-bandwidth: $\text{time-half-bandwidth } TW = NF/2$). But it seems that the plots in Figure 2 and 3 show much higher frequency resolution? For the spike-field coherence, $TW = 0.8 * 15 = 12$, but how this results in only 4 Slepian tapers, given number of tapers $L = 2TW - 1$? I am sure I am missing something. Incidentally, has Chronux also been used for the LFP power and coherence, or custom code, or native MATLAB functions (e.g. `pmtm.m`), as opposed to the Chronux usage for the spike-LFP coherence?

What were the reasons for using the multitaper approach for the low frequencies? It has been suggested that while multitaper works well for gamma frequencies, it introduces too much smoothing into the lower frequency (beta) range:

(cf. <http://www.fieldtriptoolbox.org/tutorial/timefrequencyanalysis/>)

Third, the patterns for gamma power and spiking are largely very similar (besides the bimanual apart condition). It might be expected, but given that the separation into the two streams -- spiking and LFP -- was done using the Plexon online filtering (and not via recorded broadband data that provides more flexibility for offline analyses, e.g. noncausal median filtering and removal of spiking waveforms in time domain, and/or and high frequency power from LFPs), there are potential concerns that spikes might have leaked into the high frequency LFP. Since the gamma frequency is not the main point of the study, this is not a critical issue, but perhaps the authors could comment on this.]

This issue has been addressed.

10)

[Discussion is very mechanistic-oriented and lacks the big picture for a wider audience, and some important points for experts in the field. For example, I expected the discussion of the interhemispheric competition / rivalry / push-pull models, and the construal of the increased coherence for bimanual-together condition as facilitatory exchange of information between hemispheres, as compared to potentially inhibitory effects during the bimanual-apart condition with decreased coherence (as one potential, perhaps overly simplistic, interpretation). At the very least, the authors should attempt to relate their findings to a lively debate on the compensatory/beneficial vs. maladaptive recruitment of the intact hemisphere in visuospatial and motor functions during unilateral brain damage, in neurological literature (Murase et al., 2004; Umarova et al., 2011; Bartolomeo and Thiebaut de Schotten, 2016), in noninvasive brain stimulation studies (Johansen-Berg, 2007; O'Shea et al., 2007; Sparing et al., 2009; Koch et al., 2011), and in reversible parietal lesion experiments in monkeys (Wilke et al., 2012).

On a more detailed circuit-oriented level, however, the transcallosal connectivity is mainly mediated via axonal projections to inhibitory interneurons (Bloom and Hynd, 2005; Palmer et al., 2012). How the known interhemispheric circuitry constrains the schematic model of information transfer the authors have in mind?

Another aspect that needs to be considered is the translational value of these experiments to humans, and its limitations. For example, effects of training are important. Previous work demonstrated

differences in motor performance in over-trained monkeys as compared to naïve humans: while in humans bimanual movements took longer than unimanual ones, movement times of bimanual movements in monkeys were shorter than unimanual ones (Gribova et al., 2002).]

“Our findings support the hypothesis that the beta-band LFP (PRR input) is driven by spikes (PRR output) from the opposite hemisphere.” – This sounds as if beta-band LFP is always caused by the opposite hemisphere. Even when there is no bimanual coordination? Even during initial fixation, when beta band power is strongest? Please clarify that you (probably) refer to an enhanced level of beta coherence for specific task conditions.

“It appears that the amount of information exchanged between hemispheres increases with bimanual reaches in the same direction (bimanual-together) and decreases with bimanual reaches in opposite directions (bimanual-apart).”

I don't see how increase or decrease of coherence between LFPs can be equated with “amount of information”.

“whether we describe beta power as enhanced or suppressed in any particular task --depends on the choice of a baseline time interval (Fig. 3a).” - Please fix the sentence.

Lines 399-414 – Please add here the discussion of Dean et al. results (which are compatible to your results in terms of the relationship between beta power for saccades vs reaches, cf. their Figure S4, for example).

I think the discussion is still lacking in terms of relating the finding to translational aspects (e.g. compensatory/beneficial vs. maladaptive recruitment of the intact hemisphere after unilateral lesions).

Generally, I find the discussion still somewhat disjoint and unpolished, and suggest to make another pass to attain a more coherent narrative.

Minor comments

These issues have been largely addressed.

Please make sure Supplementary figures are named consequently as they appear in the main text.

Reviewer #3:

Remarks to the Author:

I appreciate the authors' responses to our comments, however I still have a number of comments about the manuscript. Note that by “Previous comment” I mean my comment on the previous review phase.

Comments:

=====

R2.1. Most of the figures show grand averages of the variables of interest (e.g., LFP power in specific bands) and their s.e.m. I understand that the authors are averaging across many channels and trials, so even if the s.e.m. is very small, there could be quite a large spread in the single trial data. I'd like to see example single trials for all these signals, or all the single trials overlaid on top of each other – moreover, Nature Comm requires authors to include all raw data or at least show the entire distributions or box plots (error bars / bar plots are also not allowed)

R2.2. Suppl. Fig 3, which complements Fig. 2, shows that for each monkey there are significant differences across trial types between 20-30 Hz (although these differences are much less clear for M2). Would the authors see a similar effect looking at single channels on a single trial basis (this should be tested using a classifier)? Or if they averaged over enough trials? —This analyses are important given the large variability of LFP channel power —at least in my experience in other cortical areas. As in my previous review, I'd like to see more single-trial data figures and analyses.

R2.3. In a few places in the paper the authors say "population activity" to refer to all channels from all the monkeys. I know it was a common nomenclature in classic papers, but I think it may be misleading with the adoption of many neural population-based analyses (e.g., see the reviews in Shenoy, Sahani, Churchland, Ann Rev Neurosci 2013; Cunningham & Yu Nature Neurosci 2014; Gallego et al Neuron 2017; etc) that do indeed look at how populations of neurons work in a coordinated fashion. I was especially confused when this term was used for LFPs. Please rewrite.

R2.4. When the authors say that "PPR encodes contralateral arm movements but not ipsilateral arm movements," they mean that they encode that such movement has occurred, correct? In my view, encoding a movement would mean encoding the properties of such movement (e.g., end-point trajectory of the hand, the kinematics of single joints, or the activity of individual muscles). I appreciate that the authors may not have recorded this data, but they could build cross-validated models to predict movement onset from the neural activity on single-trials to test how "accurate" this encoding is, for example. Perform the analysis and rephrase

R2.5. Re: Previous comment 3.7. I appreciate the authors' detailed response, but I'd like to mention one more potentially interesting analysis to tackle this question. I now realise that the planning-related change in firing rate is smaller than the visual response. However, the authors cleverly designed the experiment to have not only ipsilateral and contralateral reach trials, but also saccade trials. Therefore they could ask which "components" of the neural activity relate to a visual response, which to an ipsilateral reach, and which to a contralateral reach. If I understood correctly, their prediction is that the neural data should be dominated by visual response activity (common to all trial types), and movement-planning activity that is first present in the contralateral trials, and later on in the ipsilateral trials. Demixed Principal Component Analysis (dPCA) is a fantastic method for this type of investigation; I refer the authors to the original paper by Kobak, Brendel et al (eLife 2016) [in that article the authors outline how to pool single neuron recordings across sessions to perform population analyses; they also provide a link to download the code]

R2.6. Re: Previous comment 3.9.1. It is true that neural population-based analyses are easier to implement on multi electrode array recordings, but they can also be carried out on datasets comprising single neuron recordings if one is interested on trial-averaged responses, as in the present study: if the behaviour is very consistent across sessions —as is the case here—, one can build an oft-called "pseudo-population" by assuming that all the neurons have been recording simultaneously. This method was used in the Kobak, Brendel et al paper mentioned above, as well as many studies by others (Churchland, Cunningham et al., Nature 2012; Kaufman et al Nature Neurosci 2014; Machens et al J Neurosci 2010; ...).

R2.7. Re: Previous comment 3.9.4. Thank you for the clarification. I think this figure would be much more convincing if the authors had six panels (one per condition) each including the raw LFP panel for each channel (and overlaid on top of it the mean +/- s.e.m.). This would indeed help to highlight the "stability" of their recordings

Minor comments:

=====

- Suppl Fig 7a —> out of RF condition is missing. The band label is incorrect (should be 20 - 30 Hz).

- Re: Previous comment 3.9.3. I'd advise the authors to use both terms (firing rates and spiking)

- Re: Previous comment 3.9.6: Do they mean that modulations in firing rate are not driven by modulations in beta in LFP at that instant? There could be complex interactions that would need to be captured by more complex models —I'm not asking the authors to do that

- Re: Previous comment 3.9.9. In their (nice) schematic the authors have an arrow that seems to indicate that LFPs cause spikes, however, their results indicate that changes in LFP power lag changes in neural firing rate. Is this just because LFPs are thought to represent synaptic inputs? Please, clarify. Also, add a legend to panels (b), (e) and clarify that (c) and (f) show actual data

- Some comments about the discussion (which I think is greatly improved):
 - * Lines 432-2: "while spikes reflect local processing and distal outputs" -> spikes could also represent distal inputs, couldn't they?
 - * The authors mention that some neurons are sensitive to trial type although the average across the entire population is not, and compare the LFPs to the latter. I'd consider the opposite: that averaging is washing away subtle effects that are behaviourally relevant —after all increasing evidence suggests that the brain does not average but rather takes weighted combinations of single neuron activity (see the reviews I mentioned above and the references therein)
 - * Lines 577-8: What is the mechanism the authors refer to?
 - * The authors should discuss the Russo & Churchland paper that I mentioned in my previous review when discussing bimanual movements in M1.

We thank the reviewers for the constructive comments and suggestions on our previous submission (MS# NCOMMS-18-30170A-Z). Our new submission addresses all of the reviewers' concerns. The reviewer comments are indicated by number and *italicized font*. Our responses are in sans serif font and quoted manuscript text is in serif font.

Reviewer #1:

1.1. *Previously resolved.*

1.2.1. *I partially disagree with the argumentation of the authors here:*

“This criticism seems to misconstrue both our results and the results of Dean et al. 2012. We found no significance difference for inRF versus outRF power in the range from 20-30 Hz. (There was a small effect for contramanual reaches, inRF greater than outRF, but this did not survive multiple comparisons correction.) We describe this result in the text:...”

The direction of the effect (inRF > outRF), clearly visible in Suppl. Figure 9, is opposite to the expectations set by the previous literature and the ordering of the beta power in this study: namely, saccade > ipsimanual > contramanual -- even if the difference is not getting significant after correcting in a specific way for multiple comparisons. There are many ways to correct for multiple comparisons, and I would prefer not to go into the discussions about the stats here; this is not the point. The point is the direction of the effect is puzzling, if you focus on “beta” as the 20-30 Hz range. Why specifically 20-30 Hz range (only high beta)? It is not readily clear from the Suppl. Figure 5 (where the power distribution is nearly linear on a log-log plot, without any noticeable bump in beta range).

“This result is almost exactly the same as that of Dean et al. (2012). They show that for all frequencies above 20 Hz, the response to inRF is greater than the response to outRF (Figure 3E of Dean et al. 2012).”

Figure 3E is for saccades in LIP. This is what Dean et al. say about PRR: “During the memory period before movement, directional selectivity of LFP activity in PRR is most pronounced in activity around 15 Hz. In addition, similar to area LIP, selectivity of memory period activity around 15 Hz depends on whether a reach is made with a saccade ($p < 0.05$, rank-sum test, Fig S4a,b, see Fig 8c). As in area LIP, directional selectivity associated with a reach is more often associated with a reduction in power before movements to the preferred direction (Fig S4c).”

I agree with the authors that a careful inspection of Dean et al. figures shows the difference in directional tuning between the lower beta (which they nearly exclusively focus on) and the higher beta in 20-30 Hz range. But most readers might not go into such details and would be confused why “beta” (however it is defined) behaves differently in these studies.

“We now highlight the congruence between our study and Dean et al. (2012) in the Discussion (Page 13, beginning line 461):...”

I believe that phrased this way, it will cause further confusion about the role and the tuning of the beta. Dean et al., and many other studies from Pesaran lab call 15-17 or ~20 Hz “beta”, while in this study “beta” refers to 20-30 Hz. Dean et al. focus on “their” beta range because

they find most power in this narrow band (see their Figure 3D), and they clearly intend to say that (their) beta is lower for preferred than for null direction (and the lower beta is in fact associated with faster RTs). My suggestion is to highlight the lack of the expected spatial tuning in the lower beta range in the current data/analysis, because this is where several previous studies found it, and to mention the distinction between different ranges of beta.

The most critical issue is why we focus on the 20-30 Hz band. This was driven by our observation, highlighted in the very first data figure, that interhemispheric LFP coherence is specifically modulated only within this band. This can be seen in Fig. 2, showing combined data from both animals, and in Supplemental Fig. 3 for each animal separately. We report in the text that significant effects occur from 22-32 Hz (lines 153-155):

During bimanual-together trials, LFP-LFP coherence was significantly elevated compared to unimanual trials at 28-32 Hz ... (blue). During bimanual-apart trials, LFP-LFP coherence was significantly depressed compared to bimanual-together trials at 22-28 Hz (purple).

As for the specific details raised by the reviewer, we have revised our Discussion to succinctly address the major specific points, e.g., clarifying that our results are consistent with Dean et al. (thank you for correcting our figure reference) (lines 464-471):

Unlike spikes, beta power was not tuned to the directional preferences of nearby spikes (beta power: **Supplementary Fig. 6**; spikes: **Fig. 3c**; compare this with gamma band power, which shows clear spatial tuning). These results confirm and extend Dean et al. (2012) who showed little or no spatial tuning between 20 and 30 Hz and strong tuning at 35 Hz and above for contramanual reaching. The lack of spatial tuning from 20 – 30 Hz could reflect widespread pooling of signals from many individual cells with diverse directional preferences. Dean et al. also saw a significant reversal of spatial tuning below ~18 Hz. We also see reversed spatial tuning in the 10 to 16 Hz range, but the effect is not significant even before multiple comparisons testing.

There are many additional specifics regarding spatial tuning that we could present. Inspired by our ongoing exchange with the reviewer, we looked even more carefully at directional tuning of LFP in LIP and PRR as a function of frequency. We have found a number of complex and unexpected patterns. We could include these fascinating and important points in this publication. However, in our enthusiasm to run these issues to the ground during the review process, we fear that both we and our reviewer are losing focus. The focus of this paper is the exchange of information between PRRs across the two hemispheres, and how this exchange may relate to bimanual coordination. Directional tuning of LFP is something of a tangent. While one can imagine that the topics might converge at some time in the future, currently they are quite separate issues, and we agree with the reviewer's astute observations that "most readers might not go into such details", and that if we persist doing so, we will only "cause further confusion", obscuring our main points regarding bimanual coordination. Therefore, we are instead preparing a separate paper that focuses specifically on directional tuning. We hope that our reviewer will have an opportunity to weigh in on that paper in the very near future!

1.2.2. “(In fact, we also see higher power for outRF compared to inRF for contramanual reaches in the 10 to 16 Hz range, but the effect is not significant even before multiple comparisons testing.)”

This is important information, is there a figure that shows this?

We have revised Figure 4 to show the preferred direction and null direction traces on the same plot:

Figure 4. LFP power as a function of movement type and frequency. LFP power is computed over the delay period for movements in the preferred and null directions. In the 16 – 32 Hz range power depends on task and frequency. In the 70 – 170 Hz range power depends on task and is relatively stable across frequency. Data are for the 113 sites from which a tuned unit was simultaneously obtained.

1.2.3. *By the way, there is a typo in Suppl. Figure 7A (“20-20 Hz”).*

Corrected.

1.3.1 [Lines 200-206: *it is surprising that the authors talk about pre-movement increase in beta power (for the contramanual reaches). The common finding is that beta is disrupted by the transient events such as visual cue and saccade/reach in frontoparietal areas (e.g. Dean et al., 2012; Sendhilnathan et al., 2017). Please show population power time-frequency plots for all conditions, aligned to cue and movement onset.*]

“Supplementary Figure 9 shows LFP power as a function of time aligned to saccade onset, reach onset and the go cue. The increase in beta power (20 – 30 Hz) is most prominent when the data are aligned on the onset of the reach rather than on the cue or on the saccade onset, and the increase is stronger in the bimanual compared to unimanual reach condition (Supplementary Figure 9). Figure 3 in Dean et al. (2012) is saccade-aligned and shows only a unimanual reaching movement, and therefore lacks the power that we have with reach-aligned data and bimanual movements.”

Here the authors meant Suppl. Figure 10, not 9, I assume. I see only (a positive) increase in the bimanual-apart condition, and only after the movement onset. Do authors mean a positive slope (rebound) after the trough, rather than the increase (relative to the baseline)?

Indeed, we meant Supplementary Figure 10. We have revised the passage to make the description more precise:

Beta power drops after the go cue, reaching a nadir ~100 ms prior to reach onset (**Supplementary Fig. 10, middle**). Subsequently, the power for trial types that include a contralateral arm movement (contramanual, bimanual-together and bimanual-apart) **rebounds and has a positive slope, up to** the time of reach onset. The pre-movement increase in LFP beta power is less clear when the traces are aligned on either the go cue or saccade onset (**Supplementary Fig. 10, left and right**). This suggests that LFP power is closely related to contralateral limb reach initiation. Indeed, there is no late increase in LFP power with ipsilateral reaches or saccades.

1.3.2. I have to admit that the Suppl. Figure 11 did not impress me too much. First, I find the asymmetric log scale with this very gradual colormap really hard to read and interpret (perhaps my subjective perception is not “uniform”, but I highly recommend changing the scale to jet, or red-white-blue, where decreases and increases can easily be distinguished). The few yellowish cells in the highlighted bimanual conditions seem to be leaking from the lower frequencies outside of the beta range. For the contramanual condition, I see pre- and peri-movement decrease in what the authors define as beta range (20-30 Hz), very similar to the Dean et al.

We generally, avoid the jet color scale due to its lack of perceptual uniformity. Nevertheless, we recognize that it is the scheme that most people are accustomed to and so we have modified Supplemental Figure 11 to use the jet color scale:

1.3.3. *All in all, I agree with the authors that there seems to be a difference in the timecourse of beta power for reaches involving contramanual hand as compared to saccades and ipsimanual hand, but I would refrain from the statements such as “Subsequently, the power for trial types that include a contralateral arm movement (contramanual, bimanual-together and bimanual-apart) increases, reaching a maximum around the time of reach onset.” – I do not think this is correct, in all three conditions the power continues to rebound to the pre-reach level and for bimanual-apart – beyond -- cf. Figure 10.*

We have removed that line from the passage (Page 7, beginning line 221; see also 1.3.1):

Beta power drops after the go cue, reaching a nadir ~100 ms prior to reach onset (**Supplementary Fig. 10, middle**). Subsequently, the power for trial types that include a contralateral arm movement (contramanual, bimanual-together and bimanual-apart) **rebounds and has a positive slope, up to** the time of reach onset. The pre-movement increase in LFP beta power is less clear when the traces are aligned on either the go cue or saccade onset (**Supplementary Fig. 10, left and right**). This suggests that LFP power is closely related to contralateral limb reach initiation. Indeed, there is no late increase in LFP power with ipsilateral reaches or saccades.

1.3.4. *“Sendhilnathan et al. (2017) is a study of LFP responses to saccades in the frontal eye fields, not parietal cortex; perhaps the reviewer had a different study in mind?” – no, I had this specific study in mind; I was talking about the frontoparietal circuitry, not just PPC.*

We now include a reference Sendhilnathan et al., in this regard.

The initial drop in beta power, prior to the first stimulus, could reflect the fact that the animal is about to receive a task instruction. The drop may be more pronounced than that seen in previous studies because of the larger number of possible tasks that could be instructed. One could argue that this is consistent with beta power reflecting the maintenance of a motor plan: when a new plan is expected, beta power drops (Dean et al., 2012; Sendhilnathan et al., 2017). Note, however, that beta power also drops shortly after the go cue and rebounds as movement is initiated (**Supplementary Fig. 10**). The first drop in power, at the time of target instruction, occurs while the movement plan is changing and motor output is held constant. The second drop in power, at the time of the go cue, occurs while the movement plan is held constant but the motor output is changing. We conclude that elevated beta power has no single correlate that applies across all areas, conditions and all times.

Additional references:

Dean, H. L., Hagan, M. A. & Pesaran, B. Only coherent spiking in posterior parietal cortex coordinates looking and reaching. *Neuron* **73**, 829–841 (2012).

Sendhilnathan, N., Basu, D. & Murthy, A. Simultaneous analysis of the LFP and spiking activity reveals essential components of a visuomotor transformation in the frontal eye field. *Proceedings of the National Academy of Sciences* **114**, 6370–6375 (2017).

1.4 *Previously resolved.*

1.5 *Previously resolved.*

1.6 *Previously resolved.*

1.7. *[What is however majorly missing from the analysis is the further separation of the data into two conditions – spikes one hemisphere – LFP another hemisphere, and vice versa. For unimanual reaches, this separation can be done in respect to the hemisphere contralateral to the acting hand. For bimanual-together reaches, in respect to the hemisphere contralateral to the target (for bimanual-apart reaches, the meaningful separation is not possible, unless both “crossed” and “uncrossed” conditions were present, in which case I would separate to crossed vs uncrossed conditions – see the question on Methods below in Minor comments). The important question here is whether there is a “directionality” (or asymmetry) of the interactions depending on specific task conditions – for instance, is there more coherence when spikes from the hemisphere that is contralateral to the target are considered, as compared to the spikes from the ipsilateral hemisphere?]*

“We agree that this is an interesting question, but a complex one and one that goes beyond the scope of the current manuscript. Since we have data from both crossed and uncrossed conditions, the questions become relatively complex: moving arm on the same or opposite side as the hemisphere in which the spikes are recorded, crossed or uncrossed reach, target in or out of the response field, etc. Splitting up the data in these many ways may require a larger data set and will certainly require many additional analyses and descriptions that are only peripherally related to the main point of the current manuscript.”

I respectfully disagree that this is only peripherally related to the main point of the current manuscript – I think the directionality is the core question of inter-hemispheric interactions. But if authors choose to avoid addressing this question, I suggest they acknowledge this in the limitations of their study.

We are planning another manuscript to address the specific issue of directionality in cross area communication. We acknowledge this limitation of the current study and note that it is an important topic of future inquiry (Discussion, page 15, line 575, addition in bold):

Additional factors may contribute to the observed coherence effects, including visual effects of one versus two targets, spatial effects of one versus two movement goals, or the effects of having motor plans that are congruent or incongruent. **In our analyses we do not consider the absolute side of the spike-field pairs. There could be an asymmetry in interhemispheric coherence depending on the specific task conditions (e.g., a left arm reach toward a right visual field target versus a left arm reach to a left visual field target).** As for an influence of the associated eye movements, we showed previously that the direction of the initial saccade on bimanual-apart trials does not affect PRR unit activity either during planning or movement⁵⁴. Saccade direction also does not affect LFP power in PRR (data not shown). Finally, while the lagged spike-LFP results support the hypothesis that there is direct communication between PRR in each hemisphere, we cannot rule out either common input or indirect communication driving the coherence effects. This will require an interventional experiment.

1.8. *Previously resolved.*

1.9. *Previously resolved.*

1.10.1 *[Discussion is very mechanistic-oriented and lacks the big picture for a wider audience, and some important points for experts in the field. For example, I expected the discussion of the interhemispheric competition / rivalry / push-pull models, and the construal of the increased coherence for bimanual-together condition as facilitatory exchange of information between hemispheres, as compared to potentially inhibitory effects during the bimanual-apart condition*

with decreased coherence (as one potential, perhaps overly simplistic, interpretation). At the very least, the authors should attempt to relate their findings to a lively debate on the compensatory/beneficial vs. maladaptive recruitment of the intact hemisphere in visuospatial and motor functions during unilateral brain damage, in neurological literature (Murase et al., 2004; Umarova et al., 2011; Bartolomeo and Thiebaut de Schotten, 2016), in noninvasive brain stimulation studies (Johansen-Berg, 2007; O'Shea et al., 2007; Sparing et al., 2009; Koch et al., 2011), and in reversible parietal lesion experiments in monkeys (Wilke et al., 2012).

On a more detailed circuit-oriented level, however, the transcallosal connectivity is mainly mediated via axonal projections to inhibitory interneurons (Bloom and Hynd, 2005; Palmer et al., 2012). How the known interhemispheric circuitry constrains the schematic model of information transfer the authors have in mind?

Another aspect that needs to be considered is the translational value of these experiments to humans, and its limitations. For example, effects of training are important. Previous work demonstrated differences in motor performance in over-trained monkeys as compared to naïve humans: while in humans bimanual movements took longer than unimanual ones, movement times of bimanual movements in monkeys were shorter than unimanual ones (Gribova et al., 2002).]

“Our findings support the hypothesis that the beta-band LFP (PRR input) is driven by spikes (PRR output) from the opposite hemisphere.” – This sounds as if beta-band LFP is always caused by the opposite hemisphere. Even when there is no bimanual coordination? Even during initial fixation, when beta band power is strongest? Please clarify that you (probably) refer to an enhanced level of beta coherence for specific task conditions.

We have revised the sentence (Page 11, line 375):

Our findings support the hypothesis that the task-specific changes observed in beta-band LFP (PRR input) are driven, in part, by spikes (PRR output) from the opposite hemisphere

1.10.2. *“It appears that the amount of information exchanged between hemispheres increases with bimanual reaches in the same direction (bimanual-together) and decreases with bimanual reaches in opposite directions (bimanual-apart).”*

I don't see how increase or decrease of coherence between LFPs can be equated with “amount of information”.

We have revised the sentence to avoid equating changes in coherence to changes in amount of information exchange:

It appears that information exchange between the hemispheres is effectively facilitated with bimanual reaches in the same direction (bimanual-together) and effectively inhibited with bimanual reaches in opposite directions (bimanual-apart).

1.10.3. “whether we describe beta power as enhanced or suppressed in any particular task -- depends on the choice of a baseline time interval (Fig. 3a).” - Please fix the sentence.

We have reworded the sentence thusly (Page 11, beginning line 404):

Whether beta power is described as enhanced or suppressed in any particular task depends on the choice of a baseline time interval (Fig. 3a).

1.10.4. Lines 399-414 – Please add here the discussion of Dean et al. results (which are compatible to your results in terms of the relationship between beta power for saccades vs reaches, cf. their Figure S4, for example).

We include discussion of the Dean et al. results (Page 11, beginning line 401):

Like these studies, we find that LFP power is strongly and differentially modulated by task and by frequency, especially at frequencies in and around the beta range (**Fig. 4**). Beta power initially decreases, falling nearly in half over the period from 350 ms before to 200 ms after target onset, and then rises to a plateau whose level depends on the task that has been instructed (**Fig. 3a**). Whether beta power is described as enhanced or suppressed in any particular task depends on the choice of a baseline time interval (**Fig. 3a**). This choice may have had a similar influence in previous studies^{36,39}. Therefore, rather than characterize the observed responses as enhancements or suppressions relative to baseline, we focus instead on their rank order. Delay period power was most suppressed for movements of both arms to two different targets, followed in order by movements of both arms together to the same target, contralateral arm movements, ipsilateral arm movements, and saccades (Fig. 4). Dean et al. (2012) also observed beta power that was suppressed more for reaches plus saccades than for saccades alone. In contrast, Scherberger et al. (2005) observed the reverse effect: beta power was greater with contralateral reaches than saccades. This difference may reflect differences in task designs. We used a random-delay visually-guided task, whereas Scherberger et al. used a fixed-delay memory task. Power in visual versus memory tasks has been observed to vary by task and even by monkey⁴⁰. In addition, we allowed coordinated saccades with our reaches while Scherberger et al. did not, and we used 5 interleaved trial types while Scherberger et al. used 2.

1.10.5. I think the discussion is still lacking in terms of relating the finding to translational aspects (e.g. compensatory/beneficial vs. maladaptive recruitment of the intact hemisphere after unilateral lesions). Generally, I find the discussion still somewhat disjoint and unpolished, and suggest to make another pass to attain a more coherent narrative.

We have expanded the section on laterality in the discussion to address how our results speak to the compensatory/beneficial vs maladaptive recruitment of the intact hemisphere after unilateral lesions (quoted below). We have made additional changes throughout the discussion to increase the polish, which we do not enumerate here.

Competing theories of callosal function focus on either its excitatory⁵⁰ or inhibitory⁵¹ role in interhemispheric processing; there is evidence for both in the human literature⁵². Our data indicate that interhemispheric connections can be functionally either inhibitory or excitatory, depending on the particular task being planned. These data do not speak to whether interhemispheric transmitters themselves are excitatory or inhibitory⁵³.

Chronic hemiparesis after unilateral stroke has been attributed to an imbalance of interhemispheric inhibitory interactions (Ward and Cohen 2004). The hemispheric competition hypothesis posits that tonic inhibition between the left and right cortex is disrupted by unilateral lesions (Kinsbourne 1977). The hypothesis is supported by tonic premovement interhemispheric inhibition (IHI) from the intact to the lesioned hemisphere in chronic stroke as measured by transcranial magnetic stimulation (Murase et al. 2004). As a result, downregulation of excitability of the intact hemisphere to restore interhemispheric balance has become a target of stroke rehabilitation (Di Pino et al., 2014). However, our results do not support this hypothesis. We find that interhemispheric interactions in intact subjects do not change shortly before

movement onset – in particular, LFP-LFP coherence was unchanged prior to a unimanual reach compared to baseline (**Fig. 2; Supplementary Fig. 3**). Maladaptive recruitment of the intact hemisphere after unilateral lesions has been recently challenged, however, on the basis that release of IHI prior to unimanual movement onset is normal in acute stroke and only becomes abnormal in chronic stroke (Stinear et al., 2015; Xi et al., 2019). Such findings raise questions about the efficacy of targeting IHI to restore function in the weakened limb. If the goal of rehabilitation is to improve the function of the paretic limb, our results suggest coordination training as another potential target for stroke rehabilitation (Kantak et al., 2017; Rizzo et al., 2019; Rose & Winstein, 2015).

Additional references:

Ward, N. S. & Cohen, L. G. Mechanisms underlying recovery of motor function after stroke. **61**, 1844–1848 (2004).

Kinsbourne, M. Hemi-neglect and hemisphere rivalry. *Advances in neurology* **18**, 41–49 (1977).

Murase, N., Duque, J., Mazzocchio, R. & Cohen, L. G. Influence of interhemispheric interactions on motor function in chronic stroke. *Annals of neurology* **55**, 400–409 (2004).

Pino, G. D. *et al.* Modulation of brain plasticity in stroke: a novel model for neurorehabilitation. *Nat Rev Neurol* **10**, 597–608 (2014).

Xu, J. *et al.* Rethinking interhemispheric imbalance as a target for stroke neurorehabilitation. *Ann Neurol* **85**, 502–513 (2019).

Stinear, C. M., Petoe, M. A. & Byblow, W. D. Primary Motor Cortex Excitability During Recovery After Stroke: Implications for Neuromodulation. *Brain Stimul* **8**, 1183–1190 (2015).

Kantak, S. S., Zahedi, N. & McGRath, R. L. Task-dependent bimanual coordination after stroke: Relationship with sensorimotor impairments. *Archives of physical medicine and rehabilitation* (2016) doi:10.1016/j.apmr.2016.01.020.

Rizzo, J.-R. *et al.* Eye-hand re-coordination: A pilot investigation of gaze and reach biofeedback in chronic stroke. *Prog Brain Res* **249**, 361–374 (2019).

Rose, D. K. & Winstein, C. J. Bimanual Training After Stroke: Are Two Hands Better Than One? *Top Stroke Rehabil* **11**, 20–30 (2004).

Minor comments

Please make sure Supplementary figures are named consequently as they appear in the main text.

Corrected.

Reviewer #3:

R2.1. Most of the figures show grand averages of the variables of interest (e.g., LFP power in specific bands) and their s.e.m. I understand that the authors are averaging across many channels and trials, so even if the s.e.m. is very small, there could be quite a large spread in the single trial data. I'd like to see example single trials for all these signals, or all the single trials overlaid on top of each other — moreover, Nature Comm requires authors to include all raw data or at least show the entire distributions or box plots (error bars / bar plots are also not allowed).

If we understand the reviewer's objection correctly, the reviewer assumes that we are confounding trial-by-trial variation with site-by-site variation. In particular, the reviewer believes that we are calculating SEM based on the total number of trials we recorded, that is, $N = \text{number of sites} \times \text{number of trials per site}$. We completely agree that to do so would grossly misrepresent our results, confounding two separate sources of variability with very different implications. In fact we compute the N for our SEM solely on the number of recording sites.

Of course, the critical question is consistency of our results across sites. We show this in Supplemental Figure 8, but the error bars that we include in our main text figures also convey this information. To reiterate the critical point, the fact that our error bars use an 'N' that is site number, not site number times trials per site, means that we avoid the pitfall that the reviewer describes. Since this was apparently not clear to our reviewer, we have added a section that walks through the error calculations for Figure 3a, making explicit where the 'N' comes from. We also take the opportunity to emphasize that the effect size is quite substantial. We would have liked to put this in the main text, but as we are already over our allowed space, we are adding it instead to the supplemental section:

Error bars are used throughout to highlight the reproducibility of responses across recording sites. In response to a reviewer query we emphasize that the 'N' used for our error bars and significance calculations is based only on the number of recording sites (or site pairs), not on the product of sites times trials. These error bars therefore provide the astute reader with the means to compute effect size. We use Figure 3a as an example. Standard errors are provided for time course and bar plots. For contramanual reaches (red data), the error bars represent our confidence in the change in power between baseline and the interval from 650 to 1150 ms after target presentation, computed within the band-limited LFP power from 20 to 30 Hz. The standard error is 1.97%, computed across 312 recording sites (see figure legend). The standard deviation is therefore a 34.9% change in power (SEM times square root of [number of recording sites minus 1]). The equivalent values for ipsimanual reaches (green data) are $\pm 2.02\%$ and 35.6%. The standard error of the *difference* is a 1.69% change in power and the standard deviation is 29.8%. (Note that this is somewhat less than the 2.82% and 49.8% that would be predicted for independent data.) The effect size can be quantified as the ratio of the effect size to the standard deviation (Cohen's D). For example, the Cohen's D value for difference in LFP power between for contramanual versus ipsimanual reaching is 21.8% divided by 29.8%, or 0.73 – a large effect size.

We do not include information about the trial-to-trial LFP variability. In the current context, this variability is noise, which we reduce by averaging over multiple trials. LFP fidelity would be important if one wished to argue that LFP is used by the brain to perform computations. Of course, there is no evidence for this and asserting otherwise is, in our view, irrational. LFP is an epiphenomenon, not a signaling mechanism, and it is generated from a wide range of processes, including non-neural processes. Thus, for our purposes, LFP variance is of little or no interest. We acknowledge that LFP variance is critical in other contexts, for example, in LFP-controlled brain machine interfaces. However, any work along those lines would use electrodes optimized for recording LFP rather than single units (much lower impedance and much larger surface area).

The reviewer argues that it is useful to show raw data. But band-limited LFP power is very far from the raw data. The raw data are voltage as a function of time. We separate out frequencies around 600-6000 Hz and use them to identify action potentials. We also separate out frequencies from about 1-100 hz and call that “LFP”. Here is what the LFP filtered from 1-100 hz looks like, averaged over 113 sites (shading shows + or – 1 SEM for two of the traces):

Next, here are averaged data from multiple trials at each of two individual sites:

Next, here are 4 sequential trials for each of 5 trial types, from a single site (the one shown above on the left):

One can see that similar patterns of responses even in just these four traces. The next steps would be to filter these data down still further, to 20-30 Hz (Fig. 3a) or 70-120 Hz (Fig. 3b), and then to compute the power within multiple ± 200 ms windows, spaced 100 ms apart, and then plot those points. We could show all of these plots, but we do not see how this would add to or illuminate the main points of this paper. However, if the reviewer feels strongly that the above plots should be in the supplemental figures, we can certainly add them.

R2.2. Suppl. Fig 3, which complements Fig. 2, shows that for each monkey there are significant differences across trial types between 20-30 Hz (although these differences are much less clear for M2). Would the authors see a similar effect looking at single channels on a single trial basis (this should be tested using a classifier)? Or if they averaged over enough trials? — These analyses are important given the large variability of LFP channel power — at least in my experience in other cortical areas. As in my previous review, I'd like to see more single-trial data figures and analyses.

Again, we are unclear on the reviewer's point. Figure 2 shows that there are significant differences in LFP coherence across the two hemispheres, as a function of trial type, between 20-30 Hz. Supplemental Figure 3 shows that this is true when the data from either one of the two monkeys is considered in isolation. The reviewer asks if we would see a similar effect looking at a single channel on a single trial basis. Again, as in R2.1, we do not see the import of this question. If we were arguing that there is a brain mechanism that reads out and acts upon this information, then it would be relevant to know how much information is available in a single trial. But this is not what this manuscript describes. We record these epiphenomenal signals to learn what shared information is present in the brain, not because we believe we are looking at signaling pathways. The idea that the brain might read out coherence across long distances seems contrived and implausible; why does the reviewer wish us to pursue such an odd direction?

Perhaps the reviewer is thinking that one might use this coherence information in a brain-machine interface to capture information about the subject's intent. Such a readout could use multiple sites, and as previously mentioned, electrodes optimized to capture LFP, not the single unit electrodes that we have used. But more to the point, why rely on second order statistics (coherence) to extract a signal that is encoded quite clearly in LFP power and spike rate data? In sum, we do not see why "these analyses are important".

R2.3. In a few places in the paper the authors say "population activity" to refer to all channels from all the monkeys. I know it was a common nomenclature in classic papers, but I think it may be misleading with the adoption of many neural population-based analyses (e.g., see the reviews in Shenoy, Sahani, Churchland, Ann Rev Neurosci 2013; Cunningham & Yu Nature Neurosci 2014; Gallego et al Neuron 2017; etc) that do indeed look at how populations of neurons work in a coordinated fashion. I was especially confused when this term was used for LFPs. Please rewrite.

We have identified and rephrased the two remaining instances in figures legends where we used this phrasing.

R2.4. When the authors say that "PPR encodes contralateral arm movements but not ipsilateral arm movements," they mean that they encode that such movement has occurred, correct? In my view, encoding a movement would mean encoding the properties of such movement (e.g., end-point trajectory of the hand, the kinematics of single joints, or the activity of individual muscles). I appreciate that the authors may not have recorded this data, but they could build cross-validated models to predict movement onset from the neural activity on single-trials to test how "accurate" this encoding is, for example. Perform the analysis and rephrase.

Our view is that PRR is concerned with planning movements, not executing them. We thought this was clear; for example, the main figures show only delay period activity, obtained prior to the cue to execute the movement. And the reviewer seems to have grasped this point; for example, point R2.5 discusses "movement-planning activity". We make this still more explicit by inserting the word "plan" or "planned" when describing PRR reach-related responses:

Abstract:

The population average unit activity (PRR output) encodes only planned contralateral arm movements while beta-band LFP power (putative PRR input) reflects the pattern of planned bimanual movements.

Results:

We have shown that beta power encodes substantial information about both contralateral and ipsilateral arm movement plans, while spikes primarily encode information about the contralateral arm.

Conclusions:

Population average spiking and gamma-band LFP power encode only contralateral arm movement plans, while beta-band LFP power contains a rich representation of the pattern of bimanual coordination. We conclude that the information about ipsilateral arm movements encoded in the beta-band LFP is driven by spikes from the opposite hemisphere, and that increased information transfer facilitates spatially coordinated movements while decreased transfer facilitates disjunctive movements.

R2.5. Re: Previous comment 3.7. I appreciate the authors' detailed response, but I'd like to mention one more potentially interesting analysis to tackle this question. I now realise that the planning-related change in firing rate is smaller than the visual response. However, the authors cleverly designed the experiment to have not only ipsilateral and contralateral reach trials, but also saccade trials. Therefore they could ask which "components" of the neural activity relate to a visual response, which to an ipsilateral reach, and which to a contralateral reach. If I understood correctly, their prediction is that the neural data should be dominated by visual response activity (common to all trial types), and movement-planning activity that is first present in the contralateral trials, and later on in the ipsilateral trials. Demixed Principal Component Analysis (dPCA) is a fantastic method for this type of investigation; I refer the authors to the original paper by Kobak, Brendel et al (eLife 2016) [in that article the authors outline how to pool single neuron recordings across sessions to perform population analyses; they also provide a link to download the code]

We agree that using demixed PCA to analyze single unit data is absolutely worth doing, and in fact we have already begun work along these lines. However, this analysis would be outside the scope of the current manuscript. Demixed PCA analysis of spikes would have fit well with our previous publication wherein we analyzed single unit activity in detail (Mooshagian et al., 2018). In the current manuscript, we focus instead on LFP responses – note the title of the manuscript, and the fact that unit responses show up in only one panel of one main figure while LFP are shown in every figure. We are hoping to publish the demixed PCA results in a separate manuscript at a later date.

Reference:

Mooshagian, E., Wang, C., Holmes, C., Snyder, L. (2018). Single Units in the Posterior Parietal Cortex Encode Patterns of Bimanual Coordination. *Cerebral Cortex* 28(5), 1549 - 1567.
<https://dx.doi.org/10.1093/cercor/bhx052>

R2.6. Re: Previous comment 3.9.1. It is true that neural population-based analyses are easier to implement on multi electrode array recordings, but they can also be carried out on datasets comprising single neuron recordings if one is interested on trial-averaged responses, as in the present study: if the behaviour is very consistent across sessions —as is the case here—, one can build an oft-called “pseudo-population” by assuming that all the neurons have been recording simultaneously. This method was used in the Kobak, Brendel et al paper mentioned above, as well as many studies by others (Churchland, Cunningham et al., Nature 2012; Kaufman et al Nature Neurosci 2014; Machens et al J Neurosci 2010; ...).

We thank the reviewer for this suggestion. An advanced analysis of the unit data would be out of place in the current paper. The suggested analysis would be appropriate for a manuscript describing the unit responses, e.g., Mooshagian et al. 2018. It would also do nicely in a follow-up paper on those unit responses; we may take this suggestion and add it to the manuscript with the demixed PCA analysis. But as we said for R2.5, a sophisticated analysis of unit responses would be out of place in the current paper, which focuses on LFP.

R2.7. Re: Previous comment 3.9.4. Thank you for the clarification. I think this figure would be much more convincing if the authors had six panels (one per condition) each including the raw LFP panel for each channel (and overlaid on top of it the mean +/- s.e.m.). This would indeed help to highlight the “stability” of their recordings.

We agree that the issue of how similar responses are across channels is important. Showing this in the raw data is not particularly helpful, however. The amplitude of LFP responses, much like the amplitude of spikes, depends on the characteristics of the electrode used for recording and on the distance of the electrode from the various sources. We could, of course, normalize the data such that we show percentage change from baseline rather than absolute voltage. Further, what is particularly important is the relative power, moving from one condition to another. Supplemental Figure 9 shows exactly these data. As requested, multiple panels are shown such that focus on just 2 conditions, and, as requested, the LFP power is shown for each individual channel. From this figure, the astute reader can immediately see just how variable the raw LFP power is across channels. In particular, in most panels, the large majority of data points lie above the diagonal line. This shows the important point that the ordering of LFP power with task is present not just in the population-averaged data, but also in the individual channel data. We believe this is exactly the point that the reviewer wished to understand:

- Suppl Fig 7a → out of RF condition is missing. The band label is incorrect (should be 20 - 30 Hz).

The band label has been corrected. Panel **a** does not include the out of RF conditions because in- and out-of-RF responses are so similar; this comparison is shown in Supplementary Figure 6. (Note that the format of this figure is identical to that of Figure 3).

- Re: Previous comment 3.9.3. I'd advice the authors to use both terms (firing rates and spiking)

We now use both terms, each where appropriate.

- Re: Previous comment 3.9.6: Do they mean that modulations in firing rate are not driven by modulations in beta in LFP at that instant? There could be complex interactions that would need to be captured by more complex models — I'm not asking the authors to do that.

We mean that the earliest appearance of modulations in firing rate cannot be driven by corresponding modulations in beta LFP, either at that instant or any time in the past, since there are no corresponding modulations in beta LFP until a time that is well *after* the modulations appear in the unit data.

- Re: Previous comment 3.9.9. In their (nice) schematic the authors have an arrow that seems to indicate that LFPs cause spikes, however, their results indicate that changes in LFP power lag changes in neural firing rate. Is this just because LFPs are thought to represent synaptic inputs? Please, clarify. Also, add a legend to panels (b), (e) and clarify that (c) and (f) show actual data.

You are correct that the arrow indicates that LFPs are thought to represent synaptic inputs. We have clarified this point (changes in bold, see caption below). In addition to the colored outlines in panels (a) and (d), we have added an explicit legend for panels (b) and (e) and amended the caption to indicate that (c) and (f) show actual data. We agree that it is puzzling that changes in LFP power lag changes in neural firing rate --- this is not what one would expect. We cannot explain why this is.

Revised Figure 5:

Revised legend:

Figure 5. Interhemispheric beta-band spike-LFP coherence distinguishes between bimanual-together and bimanual-apart movements. **a.** Schematic model depicting local and distal inputs and outputs to and from PRR. Arrows indicate mass input and output of PRR. The axon terminals of most neurons contact other neurons locally (light gray arrows), but some portion project distally, including to the homotopic area in the opposite hemisphere (dark gray arrows). Connections with non-homotopic areas are omitted for clarity. **b.** Identical spike-LFP coherence predictions under interhemispheric communication scenario (black curve) and common input scenario (gray curve). **c.** From 20-50 Hz, spike-LFP coherence is consistently high for bimanual-together movements (solid blue), intermediate for unimanual movements (solid green), and low for bimanual-apart movements (solid purple). The distributions of coherence expected by chance were computed by shuffling interspike intervals. The medians of these distributions are shown as dashed traces. The gray shaded region covers 95% of the values expected by chance; values that exceed this are marked by thickened lines ($P < 0.05$). Values exceeding the light gray line are significant at $P < 0.01$. Note that there is no effect of movement type on the shuffled coherences. For this reason, data are pooled across movement types for the $P < 0.05$ and $P < 0.01$ thresholds. Coherences that are significantly larger for bimanual-together compared to bimanual-apart are indicated by asterisks (two-tailed t-test, $P < 0.05$). Dashed vertical lines indicate the beta range (20 – 30 Hz). Coherence was measured during the 800 ms before the go cue. Data are averaged from 42 pairs of sites (24 from M1, 18 from M2) recorded simultaneously in the two hemispheres. Only sites with at least 500 spikes are shown. **d.** Schematic model depicting common input to PRR in each hemisphere. Spike-LFP coherence predictions for common input model. **e.** Lagged spike-LFP coherence predictions for direct communication and common input models, respectively. **f.** Extremes of spike-LFP coherence (24-38 Hz) occur when spikes lead LFP by 10-15 ms (gray lines). Positive and negative x-axis values indicate the relative temporal relationship between spike times and the LFP in the original data.

** Lines 432-2: “while spikes reflect local processing and distal outputs” -> spikes could also represent distal inputs, couldn't they?*

Spikes recorded in an axon terminal would represent distal inputs. However, we note in the sentence preceding the quoted sentence that with our setup (large extracellular recording electrodes), we record spikes from cells bodies and from proximal axon segments. We only very occasionally record spikes that we believe come from an axon (triphasic, short duration, high amplitude, very transient, and most common when using particularly sharp, high impedance electrodes). We do not believe we record spikes from axon terminals. Thus, the spikes that we record almost always arise from a proximal source and therefore do not represent distal inputs. We believe that in context this phrasing is correct:

The dendrites from which beta LFP most likely arise are driven by a mixture of inputs from local and distal sources. In contrast, although we record spikes exclusively from local neurons and their proximal axons, those axons may project to either local or distal targets. Thus, beta LFP reflects local processing plus distal *inputs*, while spikes reflect local processing and distal *outputs*. In addition to this structural constraint, beta rhythms in particular may be biased to reflect distal input.

** The authors mention that some neurons are sensitive to trial type although the average across the entire population is not, and compare the LFPs to the latter. I'd consider the opposite: that averaging is washing away subtle effects that are behaviourally relevant — after all, increasing evidence suggests that the brain does not average but rather takes weighted combinations of single neuron activity (see the reviews I mentioned above and the references therein).*

We completely agree with this perspective. We were not clear enough that we believe that the small changes we see are in fact behaviorally relevant – else there would be no point in sharing information

across hemispheres. We have modified a paragraph in the discussion section on “LFP power versus single unit responses” to emphasize this point:

Although not present in the population average, individual cells do show idiosyncratic effects of ipsilateral arm movement plans. We speculate that aspects of a contralateral movement plan must be modified when an ipsimanual component is added. For example, proximal musculature related to posture may depend on whether the ipsilateral arm will remain at rest or move in the same or opposite direction as the contralateral arm. The contralateral movement trajectory and hand posture might also be subtly altered, depending on whether both arms are approaching the same target⁴⁴. These subtle changes would not be systematic across different patterns of coordination or movement direction, and so would tend to disappear in the population average.

* *Lines 577-8: What is the mechanism the authors refer to?*

We have changed the sentence to read as follows:

More generally, beta LFP power reveals one site at which information about the movements of each arm is shared across the hemispheres.

* *The authors should discuss the Russo & Churchland paper that I mentioned in my previous review when discussing bimanual movements in M1.*

The reviewer presumably means to refer to the Ames and Churchland paper mentioned in their previous critique. We now cite this paper in the Discussion section on laterality. Neither Russo & Churchland nor Ames & Churchland report on bimanual movements and as a result, they do not address the issues of bimanual coordination addressed in the current manuscript. Instead, we cite Ames and Churchland as providing evidence for primary motor cortex activity for movements of either limb (Page 14, beginning line 539).

Dorsal premotor and M1 neurons are active during contralateral and ipsilateral arm movements (Ames and Churchland, 2019; Cisek et al., 2003; Steinberg et al., 2002). Bimanual movements are not coded as the linear sum of the activations of the left and right arm in the SMA⁴, M1^{3,4} or parietal cortex^{2,5}. This arrangement may allow lateralized cortex to command movement plans of the contralateral arm dependent on the state of the ipsilateral arm. Controlling bimanual movements is a particular challenge in the development of brain-machine interfaces for restoration of motor function after paralysis⁶⁰. Simultaneous recording of spikes from both hemispheres shows promise for controlling two limb prostheses at once⁶¹. LFPs offer advantages over spikes in terms of signal degradation^{62,63} and the combination of spikes and low-frequency LFP improves performance beyond spikes alone⁶⁴. Our results indicate that, in order to control both arms from signals from just one hemisphere, it is critical to record beta band LFP power, either with or without spikes.

Additional references:

Ames, K. C. & Churchland, M. M. Motor cortex signals for each arm are mixed across hemispheres and neurons yet partitioned within the population response. *Elife* **8**, e46159 (2019).

Cisek, P., Crammond, D. J. & Kalaska, J. F. Neural activity in primary motor and dorsal premotor cortex in reaching tasks with the contralateral versus ipsilateral arm. *Journal of Neurophysiology* **89**, 922–942 (2003).

Steinberg, O. *et al.* Neuronal populations in primary motor cortex encode bimanual arm movements. *The European journal of neuroscience* **15**, 1371–1380 (2002).

Reviewers' Comments:

Reviewer #1:

Remarks to the Author:

The authors made further revisions in response to the previous reviews. I still have a small number of minor points / clarifications that should be addressed in the final version of the manuscript.

Introduction

"Suggestive evidence also exists for bimanual representations in parietal cortex, but this has not been well studied 2,5,6,8."

- "Not well studied" seems rather dismissive (and ref. 2 is in fact the lab's own extensive study on bimanual coordination, published recently in *Cerebral Cortex* – should it be not taken seriously?) – perhaps specify what was the problem or is missing from these studies?

"The putative human homolog of PRR, though similarly contralaterally biased, also shows some degree of bilateral activation 15,16"

The ref. 15 is outdated and substandard by modern fMRI standards. Furthermore, it does not show contralaterally-biased activation in human PPC, because only the right hand was used to do the pointing movements. The "Limb-specific representation for reach" section in 16 is mostly relevant (beyond several exceptions such as ref. 15 in this regard), but also somewhat outdated. I suggest removing [15] and replacing it with more recent and relevant paper, e.g.

the one below (which is also very relevant for the interhemispheric mechanistic concepts discussed in the current manuscript):

Fitzpatrick AM, Dundon NM, Valyear KF (2019) The neural basis of hand choice: An fMRI investigation of the Posterior Parietal Interhemispheric Competition model. *NeuroImage* 185:208–221.

Results

The part about beta tuning has improved, and I certainly concede the point that the directional tuning properties of LFP and LFP coherence are complex – although it is not a "tangential issue", given that actions are specified and performed to specific spatial positions. I would have loved to see the analysis splitting contralateral and ipsilateral hemifields (as opposed to inRF/outRF of the "neighboring units"), but fine, let's wait for the next paper. I still maintain (despite the sarcastic tinge of the authors' response) that it would be good to avoid causing more confusion by statements that can easily be misinterpreted. Hence, I would clarify this part along the following lines:

Ln 464: "Unlike spikes, beta power was not tuned to the directional preferences of nearby spikes (beta power: Supplementary Fig. 6; spikes: Fig. 3c; compare this with gamma band power, which shows clear spatial tuning)." – remove "Unlike spikes,"?

"These results confirm and extend Dean et al. (2012) who showed little or no spatial tuning between 20 and 30 Hz and strong [congruent to spikes] tuning at 35 Hz and above for contramanual reaching. The lack of spatial tuning from 20 – 30 Hz could reflect widespread pooling of signals from many individual cells with diverse directional preferences. Dean et al. also saw a significant reversal of spatial tuning below ~18 Hz [, in the lower beta range which is the focus of their study]. We also see reversed spatial tuning in the 10 to 16 Hz range, but the effect is not significant even before multiple comparisons testing." – I added suggestions in square brackets, to ensure that the beta range of this study (high beta, >20 Hz) and previous studies (low beta, <20 Hz) are clearly identified.

Previous comment: [I have to admit that the Suppl. Figure 11 did not impress me too much. First, I

find the asymmetric log scale with this very gradual colormap really hard to read and interpret (perhaps my subjective perception is not “uniform”, but I highly recommend changing the scale to jet, or red-white-blue, where decreases and increases can easily be distinguished). The few yellowish cells in the highlighted bimanual conditions seem to be leaking from the lower frequencies outside of the beta range. For the contramanual condition, I see pre- and peri-movement decrease in what the authors define as beta range (20-30 Hz), very similar to the Dean et al.]

“We generally, avoid the jet color scale due to its lack of perceptual uniformity. Nevertheless, we recognize that it is the scheme that most people are accustomed to and so we have modified Supplemental Figure 11 to use the jet color scale:”

Unfortunately, the color bars are different for different panels, leading to zero being a different color in a, b, c and d, which is really difficult to interpret! Please re-plot with the same colormap (min, max, zero) for all panels.

The part “The color map (“viridis” package of R and matplotlib of python) is designed to be perceptually uniform (<http://cran.mtu.edu/web/packages/viridis/viridis.pdf>).” is not valid anymore, I presume.

Discussion

Previous comment: [I respectfully disagree that this is only peripherally related to the main point of the current manuscript – I think the directionality is the core question of inter-hemispheric interactions. But if authors choose to avoid addressing this question, I suggest they acknowledge this in the limitations of their study.]

“We are planning another manuscript to address the specific issue of directionality in cross area communication. We acknowledge this limitation of the current study and note that it is an important topic of future inquiry (Discussion, page 15, line 575, addition in bold):

Additional factors may contribute to the observed coherence effects, including visual effects of one versus two targets, spatial effects of one versus two movement goals, or the effects of having motor plans that are congruent or incongruent. ****In our analyses we do not consider the absolute side of the spike-field pairs [, and hence the directionality of interhemispheric communication**.**” – I think it would be good to be explicit what the absolute side of the spike-field pair might mean – my suggestion is in square brackets.

Reviewer #3:
Remarks to the Author:

I appreciate the authors’ detailed and astute answers to my comments (and those from Reviewer 1); overall I believe they’ve addressed my concerns. I only have a few minor follow up comments.

- Given their philosophical stance on LFPs —a position I largely agree with— I understand their rationale for not including single trial analysis.

- However, I’m a bit confused about reconcile this stance and the phrasing of some parts of the paper: e.g., when they state in the abstract that “beta power encodes substantial information”. The most extended use of the word “encode” assumes that someone (i.e. a downstream area) decodes the encoded information, what the authors call somewhere “signaling”. Therefore, if the LFP is not “used by the brain to perform computations” (quoting the authors again), I think they should use a different

word when talking about LFPs, since those correlations are observed in epiphenomenal signals.

- I quite like Suppl Fig 9 indeed!

- I'd advise the authors to call their monkeys Mk1 and Mk2 (or something like that) since a distracted reader skimming the paper may think M1 represents the primary motor cortex rather than Monkey 1 (and these days people working in rodents some times call premotor cortex M2 "exporting" the term from their species to primates...)

We thank the reviewers for the constructive comments and suggestions on our previous submission (MS# NCOMMS-18-30170C). Our new submission addresses all of the reviewers' concerns. The reviewer comments are indicated by *italicized font*. Our responses are in sans serif font and quoted manuscript text is in serif font.

Reviewer #1 (Remarks to the Author):

The authors made further revisions in response to the previous reviews. I still have a small number of minor points / clarifications that should be addressed in the final version of the manuscript.

Introduction

“Suggestive evidence also exists for bimanual representations in parietal cortex, but this has not been well studied 2,5,6,8.”

- “Not well studied” seems rather dismissive (and ref. 2 is in fact the lab’s own extensive study on bimanual coordination, published recently in Cerebral Cortex – should it be not taken seriously?) – perhaps specify what was the problem or is missing from these studies?

Point taken. We have revised the passage:

Suggestive evidence also exists for bimanual representations in parietal cortex^{2,5,6,8}, but those studies did not address bilateral contributions to movement.

“The putative human homolog of PRR, though similarly contralaterally biased, also shows some degree of bilateral activation 15,16”

The ref. 15 is outdated and substandard by modern fMRI standards. Furthermore, it does not show contralaterally-biased activation in human PPC, because only the right hand was used to do the pointing movements. The “Limb-specific representation for reach” section in 16 is mostly relevant (beyond several exceptions such as ref. 15 in this regard), but also somewhat outdated. I suggest removing [15] and replacing it with more recent and relevant paper, e.g.

the one below (which is also very relevant for the interhemispheric mechanistic concepts discussed in the current manuscript):

Fitzpatrick AM, Dundon NM, Valyear KF (2019) The neural basis of hand choice: An fMRI investigation of the Posterior Parietal Interhemispheric Competition model. NeuroImage 185:208–221.

We have updated the references as suggested.

Results

The part about beta tuning has improved, and I certainly concede the point that the directional tuning properties of LFP and LFP coherence are complex – although it is not a “tangential issue”, given that actions are specified and performed to specific spatial positions. I would have loved to see the analysis splitting contralateral and ipsilateral hemifields (as opposed to inRF/outRF of the “neighboring units”), but fine, let’s wait for the next paper. I still maintain (despite the sarcastic tinge of the authors’ response) that it would be good to avoid causing more confusion by statements that can easily be misinterpreted. Hence, I would clarify this part along the following lines:

Ln 464: “Unlike spikes, beta power was not tuned to the directional preferences of nearby spikes (beta power: Supplementary Fig. 6; spikes: Fig. 3c; compare this with gamma band power, which shows clear spatial tuning).” – remove “Unlike spikes,”?

We have revised the sentence as suggested:

Beta power was not tuned to the directional preferences of nearby spikes (beta power: Supplementary Fig. 6; spikes: Fig. 3c; compare this with gamma band power, which shows clear spatial tuning).

“These results confirm and extend Dean et al. (2012) who showed little or no spatial tuning between 20 and 30 Hz and strong [congruent to spikes] tuning at 35 Hz and above for contramanual reaching. The lack of spatial tuning from 20 – 30 Hz could reflect widespread pooling of signals from many individual cells with diverse directional preferences. Dean et al. also saw a significant reversal of spatial tuning below ~18 Hz [, in the lower beta range which is the focus of their study]. We also see reversed spatial tuning in the 10 to 16 Hz range, but the effect is not significant even before multiple comparisons testing.” – I added suggestions in square brackets, to ensure that the beta range of this study (high beta, >20 Hz) and previous studies (low beta, <20 Hz) are clearly identified.

We have revised the passage as suggested:

These results confirm and extend Dean et al. (2012) who showed little or no spatial tuning between 20 and 30 Hz and strong tuning, congruent to spikes, at 35 Hz and above for contramanual reaching. The lack of spatial tuning from 20 – 30 Hz could reflect widespread pooling of signals from many individual cells with diverse directional preferences. Dean et al. also saw a significant reversal of spatial tuning below ~18 Hz, in the lower beta range which is the focus of their study. We also see reversed spatial tuning in the 10 to 16 Hz range, but the effect is not significant even before multiple comparisons testing.

Previous comment: [I have to admit that the Suppl. Figure 11 did not impress me too much. First, I find the asymmetric log scale with this very gradual colormap really hard to read and interpret (perhaps my subjective perception is not “uniform”, but I highly recommend changing the scale to jet, or red-white-blue, where decreases and

increases can easily be distinguished). The few yellowish cells in the highlighted bimanual conditions seem to be leaking from the lower frequencies outside of the beta range. For the contramanual condition, I see pre- and peri-movement decrease in what the authors define as beta range (20-30 Hz), very similar to the Dean et al.]

“We generally, avoid the jet color scale due to its lack of perceptual uniformity. Nevertheless, we recognize that it is the scheme that most people are accustomed to and so we have modified Supplemental Figure 11 to use the jet color scale.”

Unfortunately, the color bars are different for different panels, leading to zero being a different color in a, b, c and d, which is really difficult to interpret! Please re-plot with the same colormap (min, max, zero) for all panels.

The part “The color map (“viridis” package of R and matplotlib of python) is designed to be perceptually uniform (<http://cran.mtu.edu/web/packages/viridis/viridis.pdf>).” is not valid anymore, I presume.

Thank you for catching this. We have corrected the color bars. The stray reference to the viridis color package has been removed.

Discussion

Previous comment: [I respectfully disagree that this is only peripherally related to the main point of the current manuscript – I think the directionality is the core question of inter-hemispheric interactions. But if authors choose to avoid addressing this question, I suggest they acknowledge this in the limitations of their study.]

“We are planning another manuscript to address the specific issue of directionality in cross area communication. We acknowledge this limitation of the current study and note that it is an important topic of future inquiry (Discussion, page 15, line 575, addition in bold):

*Additional factors may contribute to the observed coherence effects, including visual effects of one versus two targets, spatial effects of one versus two movement goals, or the effects of having motor plans that are congruent or incongruent. **In our analyses we do not consider the absolute side of the spike-field pairs [, and hence the directionality of interhemispheric communication”.]” – I think it would be good to be explicit what the absolute side of the spike-field pair might mean – my suggestion is in square brackets.*

We have revised the sentence as recommended:

Additional factors may contribute to the observed coherence effects, including visual effects of one versus two targets, spatial effects of one versus two movement goals, or the effects of having motor plans that are congruent or incongruent. In our analyses we do not consider the absolute side of the spike-field pairs, and hence the directionality of interhemispheric communication, e.g., the laterality of the target in each trial, or whether or not reaches cross the midline.

Reviewer #3 (Remarks to the Author):

I appreciate the authors' detailed and astute answers to my comments (and those from Reviewer 1); overall I believe they've addressed my concerns. I only have a few minor follow up comments.

- Given their philosophical stance on LFPs —a position I largely agree with— I understand their rationale for not including single trial analysis.

Thank you.

- However, I'm a bit confused about reconcile this stance and the phrasing of some parts of the paper: e.g., when they state in the abstract that “beta power encodes substantial information”. The most extended use of the word “encode” assumes that someone (i.e. a downstream area) decodes the encoded information, what the authors call somewhere “signaling”. Therefore, if the LFP is not “used by the brain to perform computations” (quoting the authors again), I think they should use a different word when talking about LFPs, since those correlations are observed in epiphenomenal signals.

The last sentence of the comment is throwing me. The reference to the brain not using LFPs for computation is from our previous reply. I think he's suggesting we not say “beta power encodes”, right? So we should say beta power

- I quite like Suppl Fig 9 indeed!

Thank you.

- I'd advise the authors to call their monkeys Mk1 and Mk2 (or something like that) since a distracted reader skimming the paper may think M1 represents the primary motor cortex rather than Monkey 1 (and these days people working in rodents some times call premotor cortex M2 "exporting" the term from their species to primates...)

Agreed. We now refer to the monkeys by their initials, MkT and MkZ instead of monkeys 1 and 2, respectively. Abbreviations have been changed throughout the manuscript.